# Adversarial Latent Feature Augmentation for Fairness

**Hoin Jung, Junyi Chai & Xiaoqian Wang** [*]
Elmore Family School of Electrical and Computer Engineering
Purdue University
West Lafayette, IN 47907, USA
`{jung414,chai28,joywang}@purdue.edu`

## Abstract

Achieving fairness in machine learning remains a critical challenge, especially due to the opaque effects of data augmentation on input spaces within nonlinear neural networks. Nevertheless, current approaches that emphasize augmenting latent features, rather than input spaces, offer limited insights into their ability to detect and mitigate bias. In response, we introduce the concept of the "unfair region" in the latent space, a subspace that highlights areas where misclassification rates for certain demographic groups are disproportionately high, leading to unfair prediction results. To address this, we propose Adversarial Latent Feature Augmentation (ALFA), a method that leverages adversarial fairness attacks to perturb latent space features, which are then used as data augmentation for fine-tuning. ALFA intentionally shifts latent features into unfair regions, and the last layer of the network is fine-tuned with these perturbed features, leading to a corrected decision boundary that enhances fairness in classification in a cost-effective manner. We present a theoretical framework demonstrating that our adversarial fairness objective reliably generates biased feature perturbations, and that fine-tuning on samples from these unfair regions ensures fairness improvements. Extensive experiments across diverse datasets, modalities, and backbone networks validate that training with these adversarial features significantly enhances fairness while maintaining predictive accuracy in classification tasks. The code is available on GitHub.

## 1 Introduction

The issue of fairness in machine learning is a well-recognized and multifaceted challenge. Addressing fairness often involves manipulating or augmenting data to address inequalities between demographic groups in the input space, as studied in (Jang et al., 2021; Rajabi & Garibay, 2022). However, the transparency and efficacy of data augmentation in the input space to foster fairness are not always clear due to the challenge of determining how transformations affect the nonlinear decision boundary. This complexity has led to exploring augmentation strategies in the latent space, allowing for a more nuanced analysis of augmentation's impact.

The linearity of the last layer in neural networks' latent space, such as in Multilayer Perceptron (MLP) and Convolutional Neural Networks (CNNs), facilitates the examination of fairness issues at the decision boundary. For example, (Buolamwini & Gebru, 2018) highlights how demographic imbalances, like race, can lead to disproportionately higher misclassification rates, such as higher false positives for individuals with darker skin tones in facial analysis software (Klare et al., 2015). This example illustrates how biased data can result in unfair outcomes. In the latent space, the linear classifier enables a clearer examination of how such imbalances in data distribution impact the decision boundary. Beyond group-level analysis, exploring the latent space helps to identify misclassified segments influenced by the linear classifier.

In short, exploring fairness within the latent space of neural networks is crucial for understanding and mitigating biases. However, existing latent augmentation methods often overlook the fundamental

---

[*]Corresponding author.

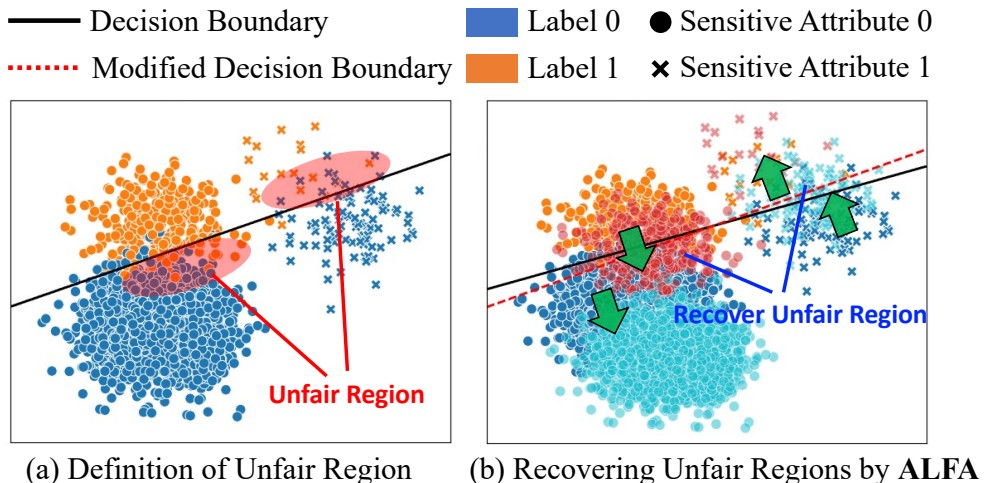

(a) Definition of Unfair Region      (b) Recovering Unfair Regions by **ALFA**

Figure 1: A synthetic data example illustrating fairness issues and the identification of unfair regions within the latent features. Let the demographic group $\{A = 1\}$ privileged to be predicted as $Y = 1$. The misclassification rates of subgroup $\{A = 1, Y = 0\}$ and $\{A = 0, Y = 1\}$ are disproportionately high, indicated as *unfair region* in the left figure. ALFA generates adversarial perturbations in the latent space against the fairness constraint, pushing the features towards a biased direction so that the perturbed features overlap with the unfair region. Fine-tuning the last layer on these perturbed features adjusts the decision boundary, correcting the unfair region and resulting in fairer predictions.

question of where and how fairness issues originate within the latent distribution. For example, Fair-Mixup (Mroueh et al., 2021) operates under the assumption that a manifold exists in the latent space between two demographic groups and advocates for data generation via interpolation on this manifold. However, this assumption may be overly stringent, and Fair-Mixup does not specifically address where fairness issues arise in the latent space.

Similarly, FAAP (Wang et al., 2022) attempts to obfuscate sensitive attributes in the latent representation by projecting features towards the sensitive decision boundary. The challenge, however, arises when perturbed features align along this boundary, potentially distorting the original feature distribution. If these features remain confined to a linear alignment, the resulting decision boundary can vary significantly, reducing the model's generalization capacity.

To address these challenges, we propose the concept of the *unfair region* to analyze the root causes of fairness issues in the latent space. This region is characterized by disproportionate misclassification rates between privileged and underprivileged groups. Figure 1 (a) illustrates this concept, highlighting areas where biased predictions are most prevalent. We further demonstrate the extent of bias within the unfair region using synthetic data, as shown in Figure 2. Details of the synthetic data are introduced in Appendix C.

However, examining the high-dimensional latent distribution is not straightforward. To automate the detection and correction of unfair regions in the latent space, we propose a novel approach called **A**dversarial **L**atent **F**eature **A**ugmentation (**ALFA**). This method employs a counter-intuitive use of adversarial attacks and data augmentation. Specifically, we introduce a fairness attack by perturbing latent features based on a fairness constraint (Zafar et al., 2017). For instance, the perturbation pushes privileged groups toward favorable outcomes while directing underprivileged groups toward unfavorable outcomes, regardless of their true labels. This manipulation strengthens the correlation between sensitive attributes and decision outcomes. To maintain the semantic integrity of perturbed features, ALFA minimizes the Sinkhorn distance (Genevay et al., 2018) between the original and perturbed features.

Consequently, fine-tuning the classifier with these perturbed features helps correct the decision boundary, directly addressing the unfair regions and achieving more balanced misclassification rates across demographic groups. Figure 2 (b) illustrates this concept using synthetic data, showcasing the

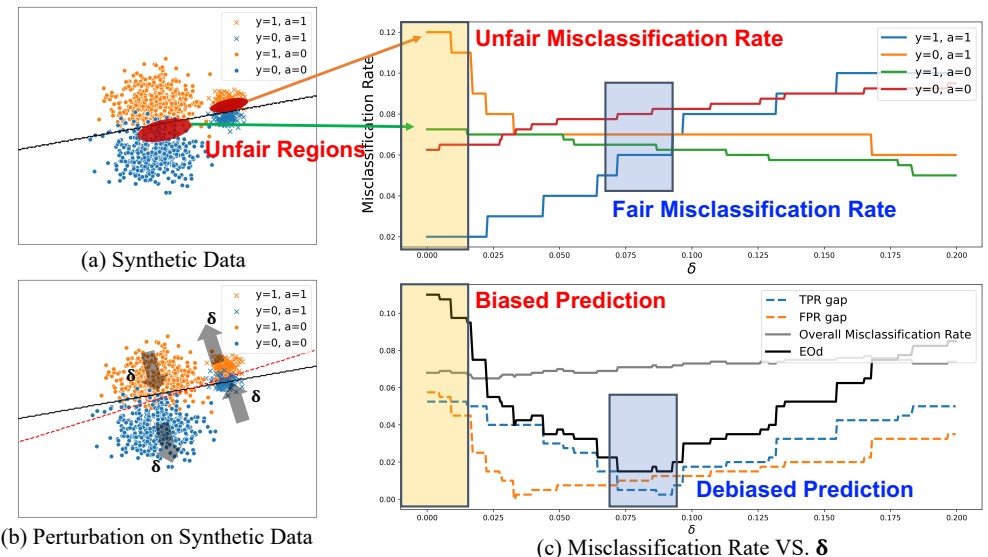

Figure 2: (a) In the synthetic dataset, regions exhibiting disproportionately high misclassification rates are identified, indicating potential unfairness in predictions. (b) A fairness attack introduces perturbations, denoted by $\delta$ towards these unfair regions, attempting to balance the misclassification rate across groups by the corrected decision boundary (red line in (b)). (c) An appropriate $\delta$ can equalize misclassification rates, achieving debiased predictions with a lower $\Delta EOd$ (sum of FPR gap and TPR gap), without significant compromising in accuracy, as shown in the gray line in (c). The improvement in $\Delta EOd$ is theoretically guaranteed as described in Section 3.3

impact of adversarial perturbations on the decision boundary. Furthermore, we provide theoretical proof that training on perturbed datasets improves fairness in Section 3.3.

Our method is validated through extensive experiments on various datasets, including tabular datasets such as Adult, COMPAS, German, and Drug; images from CelebA; and text from Wikipedia, demonstrating its versatility. These experiments confirm that our method preserves accuracy while significantly enhancing group fairness across diverse datasets and backbone networks.

We summarize our contributions as follows:

1. Introduced a novel latent space data augmentation method aimed at identifying and rectifying areas of unfairness in classification models.

2. Provided a theoretical foundation that elucidates the counter-intuitive impact of adversarial perturbations on improving fairness, supported by visual illustrations of the corrected decision boundary.

3. Demonstrated that our method consistently achieves group fairness without compromising accuracy through experiments on tabular, image, and text datasets.

## 2 RELATED WORK

### 2.1 FAIRNESS IN MACHINE LEARNING

Diverse approaches have been proposed to secure fairness in the classification tasks. Chai & Wang (2022) and Li & Liu (2022) proposed data reweighing; allocating weights for all samples according to their importance. Chai & Wang (2022) balanced the gap between demographic groups weighing error-prone samples in an adaptive way. Li & Liu (2022) adopted influence function (Koh & Liang, 2017) to evaluate individual sample's importance in affecting prediction. As an in-processing approach, Zafar et al. (2017) and Wu et al. (2019) developed a fairness constraint adopting covariance between sensitive attribute and classifier, and extending the constraint having convexity, respectively. Jang et al. (2024) integrated both data and model perspectives to improve fairness.

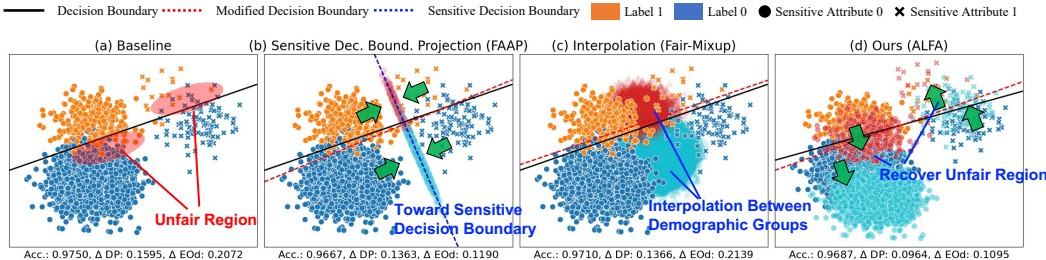

Figure 3: Comparison of each data manipulation in latent space for synthetic unfair data, (a) a naive classifier, (b) FAAP, (c) Fair-Mixup, and (d) ALFA. The solid black line represents the original decision boundary obtained via Logistic Regression, while the red dashed line shows the updated decision boundary after feature manipulation. The manipulated features, indicated in cyan and red, correspond to perturbed features in FAAP and ALFA, and interpolated features in Fair-Mixup. The blue dashed line in FAAP indicates the sensitive decision boundary. The generated features in FAAP and Fair-Mixup do not directly consider the region where fairness issue happens, and might not sufficiently mitigate bias.

Some approaches use data augmentation to improve fairness. Jang et al. (2021) and Rajabi & Garibay (2022) generated new fair data using VAE and GAN, respectively. Hsu et al. (2022) and Zhao et al. (2020) adopted adversarial samples as data augmentation to improve accuracy and robustness, respectively. Similarly, Li et al. (2023) generated antidote data analogous to the original data but containing the opposite sensitive attribute to enhance individual fairness.

Manipulating features in the latent space becomes popular. Mroueh et al. (2021) proposed to generate new data in the latent space by interpolation between latent features from different sensitive groups to optimize fairness constraints. Wang et al. (2022) suggested adversarial perturbation on the latent features towards the sensitive hyperplane which predicts the demographic group. Sun et al. (2023) disentangle the latent feature into the sensitive feature and non-sensitive feature and obfuscate the sensitive feature only. Mao et al. (2023) fine-tune the pre-trained classifier by training the last layer with the balanced latent features under the designated fairness constraint. We demonstrate in Figure 3 how our latent feature manipulation differs from that of Mroueh et al. (2021) and Sun et al. (2023).

In contrast, there exist attempts to attack fairness. Koh et al. (2018) suggested attacking anomaly detectors by blending perturbed data with the natural data and by optimizing influence-based gradient ascent. Mehrabi et al. (2021) extended the idea of (Koh et al., 2018) combining the fairness constraint suggested by Zafar et al. (2017). Similarly, Solans et al. (2020) developed a gradient-based poisoning attack on algorithmic fairness. Chhabra et al. (2022) proposed a fairness attack and defense framework in terms of unsupervised learning and fair clustering.

## 3 PROPOSED METHOD

**Motivation.** We use *Equalized Odds* (EOd) as the criterion for group fairness. Demographic Parity (DP) requires independence between the predicted outcome and the sensitive attribute $A \in \{0, 1\}$, such that $P(\hat{Y}|A = 0) = P(\hat{Y}|A = 1)$, i.e., $\hat{Y} \perp\!\!\!\perp A$. However, DP's usefulness is limited when there is a correlation between $Y$ and $A$, where $Y \not\perp\!\!\!\perp A$. EOd overcomes this limitation by conditioning on the true label $Y$. It requires that $P(\hat{Y}|A = 1, Y = y) = P(\hat{Y}|A = 0, Y = y)$ for all $y \in \{0, 1\}$. In other words, EOd ensures that the misclassification rates between the two demographic groups are equal for each true label. In general, when a classifier is biased, misclassification occurs in specific regions of the latent space. For instance, if individuals in the privileged group are more likely to be predicted as positive, i.e., $P(\hat{Y}|A = 1) \geq P(\hat{Y}|A = 0)$, the false positive rate for group $\{A = 1\}$ and the false negative rate for group $\{A = 0\}$ will be disproportionately higher. Specifically, $P(\hat{Y} = 1|A = 1, Y = 0) \geq P(\hat{Y} = 1|A = 0, Y = 0)$ and $P(\hat{Y} = 0|A = 0, Y = 1) \geq P(\hat{Y} = 0|A = 1, Y = 1)$. To quantify this disparity, we define the evaluation metric as $\Delta EOd = \sum_{y \in \{0,1\}} |P(\hat{Y} = 1|A = 1, Y = y) - P(\hat{Y} = 1|A = 0, Y = y)|$.

**Unfair Region.** As shown in Figure 1 and 2, the *unfair regions* represent areas in the latent space where certain demographic groups are disproportionately misclassified by the biased classifier.

**Definition 3.1.** Consider a linear classifier $g(z) = \text{sign}(w^T z + b)$, where $z = f(x)$ represents the latent space generated by an encoder $f$, and $w$ and $b$ are the classifier's weights and bias. The classifier $g(z)$ is considered biased if its decision boundary $w^T z + b = 0$ results in significantly different outcomes (e.g., higher misclassification rates) for the groups defined by the sensitive attribute $A \in \{0, 1\}$. We define the unfair region $R_{\text{unfair}}$ as the subspace of the latent space where the classifier's decision boundary results in disproportionately high differences in error rates between these groups:

$$R_{\text{unfair}} = \big\{ z \in \mathbb{R}^d : \big| P(g(z) = y | A = 1) \big| - P(g(z) = y | A = 0) \big| > \tau \big\} \,\forall y \in \{0, 1\}, \quad (1)$$

where $\tau$ is a threshold indicating significant bias.

The proposed method automatically identifies the unfair regions $R_{\text{unfair}}$ and generates perturbed samples that directly cover the area by over- or under-representing demographic groups for each label, leveraging a fairness attack. This region highlights where the classifier exhibits significant discrepancies in outcomes across demographic groups. Notably, this identification does not rely on a predefined threshold $\tau$ but instead uses the attack to pinpoint areas where bias is most pronounced. Consequently, training the last layer of the network on these perturbed latent features corrects the decision boundary, reducing the misclassification rates for biased subgroups.

## 3.1 FAIRNESS ATTACK

In this section, we adopt an objective function suggested in (Zafar et al., 2017) for fairness attack, $\mathcal{L}_{\text{fair}}$. Zafar et al. (2017) suggested measurement for disparate impact using a covariance between the sensitive attribute $a$ and the signed distance $d_\theta$ from $x$ to the decision boundary, i.e. $Cov(a, d_\theta) \approx 0$ means fair where the signed distance $d_\theta$ obtained by the logit (inverse sigmoid) function from the predicted probability $\hat{y}$, i.e. $d_\theta = \sigma^{-1}(\hat{y})$. Contrary to (Zafar et al., 2017), we maximize the covariance between the sensitive feature and the signed distance between the perturbed feature and the decision boundary of the pre-trained classifier. Therefore, the fairness constraint $\mathcal{L}_{\text{fair}}$ is defined

$$\mathcal{L}_{\text{fair}} = |Cov(a, \sigma^{-1}(\hat{y}))| = |Cov(a, g(z + \delta))|, \quad (2)$$

where the overall model consists of an encoder $f$ and linear classifier $g$ such that $\hat{y} = g(f(x)) = g(z)$, $x$ is the input, $z \in \mathbb{R}^{N \times d}$ is the latent feature, $\delta \in \mathbb{R}^{N \times d}$ is the perturbation, $N$ is the number of samples, and $d$ is the dimension of latent feature. Let $\tilde{z} = z + \delta$ and $d_i = g(\tilde{z}_i)$, then Eq. 2 becomes

$$\mathcal{L}_{\text{fair}} = |Cov(a, g(\tilde{z}))| = \left| \mathbb{E}\big[ (a - \bar{a})(g(\tilde{z}) - \mathbb{E}[g(\tilde{z})]) \big] \right| \approx \frac{1}{N_p} \left| \sum_{i=1}^{N_p} (a_i - \bar{a})(d_i - \bar{d}) \right|,$$

where $N_p$ is the number of target samples and $\bar{d}$ is the mean of all $d_i$. In the fairness attack, we adopt *upsampling strategy* selecting the same size of samples from each subgroup as an attacking target such that $N_p = 4 \cdot \max(n_{00}, n_{01}, n_{10}, n_{11})$, where $n_{ay}$ denotes the number of samples for each subset such that $n_{ay} = |S_{ay}|$, $S_{ay} = \{i | a_i = a, y_i = y\}$, $a \in \{0, 1\}$ and $y \in \{0, 1\}$.

In fact, any type of fairness constraint can be applied for $L_{\text{fair}}$ during the fairness attack. We demonstrate a convex fairness constraint (Wu et al., 2019) as an alternative to the covariance fairness constraint, also showing significant improvements in fairness as demonstrated in Appendix L.

A positive covariance between two variables indicates that they tend to increase or decrease together, while a negative covariance means an inverse relationship. A fairness attack aims to maximize the covariance to make the sensitive attribute significantly affect the decision of the given classifier. Instead of $|Cov(a, g(\tilde{z}))|$ in $\mathcal{L}_{\text{fair}}$, we follow the sign of covariance $(Cov(a, y))$ of the clean dataset to determine $\mathcal{L}_{\text{fair}}$ for fairness attack to effectively exacerbate the fairness for the given classifier,

$$\mathcal{L}_{\text{fair}} = \begin{cases} Cov(a, g(\tilde{z})) & \text{if } Cov(a, y) \geq 0 \\ -Cov(a, g(\tilde{z})) & \text{if } Cov(a, y) < 0. \end{cases} \quad (3)$$

In this way, we observe in Table 3 in Appendix E that the consequent sign of $Cov(a, \hat{y})$ also follows the sign of covariance in clean dataset.

---

**Algorithm 1** Adversarial Latent Feature Augmentation

---

**Require:** Clean dataset $(\boldsymbol{X}_c, \boldsymbol{Y}_c)$, hyperparameter $\alpha$, the number of epochs $T$, pretrained encoder $f$ and classifier $g$.

**Ensure:** Fair classifier $g_\theta$

    Obtain $(\boldsymbol{X}_p, \boldsymbol{Y}_p)$ by balanced upsampling for $(\boldsymbol{X}_c, \boldsymbol{Y}_c)$.

    Obtain latent feature set $(\boldsymbol{Z}_p, \boldsymbol{Y}_p)$ where $\boldsymbol{Z}_p = f(\boldsymbol{X}_p)$.

    Compute a mean absolute distance $\epsilon$ between latent features and the decision boundary.

    Fairness attack to obtain $\boldsymbol{\delta}^* = \arg\max_{\|\boldsymbol{\delta}\|_2 \leq \epsilon}\big(\mathcal{L}_{\text{fair}} - \alpha D(\boldsymbol{z}, \boldsymbol{z}+\boldsymbol{\delta})\big), \forall \boldsymbol{z} \in \boldsymbol{Z}_p$ .

    **for** $i = 1, \cdots, T$ **do**

        Fine-tune the classifier $g$ with the adversarial latent feature $\tilde{\boldsymbol{z}} = \boldsymbol{z} + \boldsymbol{\delta}^*$.

        $\theta \leftarrow \theta^* = \arg\min_\theta \frac{1}{|\boldsymbol{X}_c|+|\boldsymbol{Z}_p|}\Big(\sum_{\boldsymbol{x}_i \in \boldsymbol{X}_c} \mathcal{L}_{\text{ce}}\big(g(f(\boldsymbol{x}_i)), y_i, \theta\big) + \sum_{\boldsymbol{z}_j \in \boldsymbol{Z}_p} \mathcal{L}_{\text{ce}}\big(g(\boldsymbol{z}_j + \boldsymbol{\delta}_j^*), y_j, \theta\big)\Big)$.

    **end for**

---

We prove that $\mathcal{L}_{\text{fair}}$ is the lower bound of $\Delta EOd$ through **Proposition B.1 and Theorem B.2 in Appendix B**. Consequently, we can conduct a fairness attack by maximizing $\mathcal{L}_{\text{fair}}$, causing the perturbed latent features to result in unfair predictions with a high $\Delta EOd$ on the given pre-trained classifier.

## 3.2 SINKHORN DISTANCE

The goal of an adversarial fairness attack is to lead a pre-trained classifier to predict biased results on perturbed samples while maintaining the distribution of given data to keep it semantically meaningful. In order to effectively attack the classifier, we adopt the Wasserstein Distance (Arjovsky et al., 2017) to minimize the statistical distance between $\boldsymbol{z}$ and $\tilde{\boldsymbol{z}}$, i.e. $D(\boldsymbol{z}, \tilde{\boldsymbol{z}})$. Wasserstein distance is a powerful tool for measuring the statistical distance between two probability distributions and is sensitive to small perturbations. One drawback of Wasserstein distance is its burden on computational cost. However, a faster and more accurate algorithm is developed to approximate the Wasserstein distance using Sinkhorn iteration, namely Sinkhorn distance (Genevay et al., 2018). Sinkhorn Distance is an approximate entropy regularized Wasserstein distance using the Sinkhorn algorithm measuring the distance between two probability distributions in terms of optimal transport problem. A detailed explanation and cost effectiveness of Sinkhorn distance is in Appendix D.1.

## 3.3 ADVERSARIAL LATENT FEATURE AUGMENTATION

We propose a novel data augmentation technique in the latent space, *Adversarial Latent Feature Augmentation* (ALFA) to mitigate the bias in the binary classification. We pre-train the encoder and classifier by empirical risk minimization with binary cross entropy loss $\mathcal{L}_{\text{ce}}$, $\min_\theta \frac{1}{N}\sum_{i=1}^N \mathcal{L}_{\text{ce}}(g(f(\boldsymbol{x}_i)), y_i)$, where $\boldsymbol{x}_i$ is the input data and $y_i \in \{0, 1\}$ is the class label. The trained classifier is potentially biased to the particular sensitive attribute due to the imbalance in the dataset. As shown in Figure 1, unfair regions are identified for each label caused by a given classifier which we aim to cover by introducing the perturbed latent features having corresponding labels with the over/underestimated demographic group.

The adversarial latent features are generated by the fairness attack while maintaining their distribution by the Sinkhorn distance denoted as $D(\boldsymbol{z}, \boldsymbol{z}+\boldsymbol{\delta})$. During the attacking step, parameters of both encoder $f$ and linear classifier $g$ are frozen. The direction and magnitude of perturbation are determined by the fairness attack introduced in Section 3.1 and 3.2,

$$\max_{\|\boldsymbol{\delta}\|_2 \leq \epsilon} \Big(\mathcal{L}_{\text{fair}} - \alpha D(\boldsymbol{z}, \boldsymbol{z}+\boldsymbol{\delta})\Big), \tag{4}$$

where $\alpha$ is a hyperparatmer, and $\epsilon$ is the mean absolute distance between latent features and pre-trained decision boundary. The Sinkhorn distance term is obtained by batch-wise computation. Finally, the classifier is fine-tuned using both the original and adversarial latent features, with the encoder $f$ remaining frozen, and only the parameters of the linear classifier $g$ being updated. The objective function for the fine-tuning is

$$\min_\theta \frac{1}{|\boldsymbol{X}_c|+|\boldsymbol{Z}_p|}\Big(\sum_{\boldsymbol{x}_i \in \boldsymbol{X}_c} \mathcal{L}_{\text{ce}}\big(g(f(\boldsymbol{x}_i)), y_i, \theta\big) + \sum_{\boldsymbol{z}_j \in \boldsymbol{Z}_p} \mathcal{L}_{\text{ce}}\big(g(\boldsymbol{z}_j + \boldsymbol{\delta}_j^*), y_j, \theta\big)\Big), \tag{5}$$

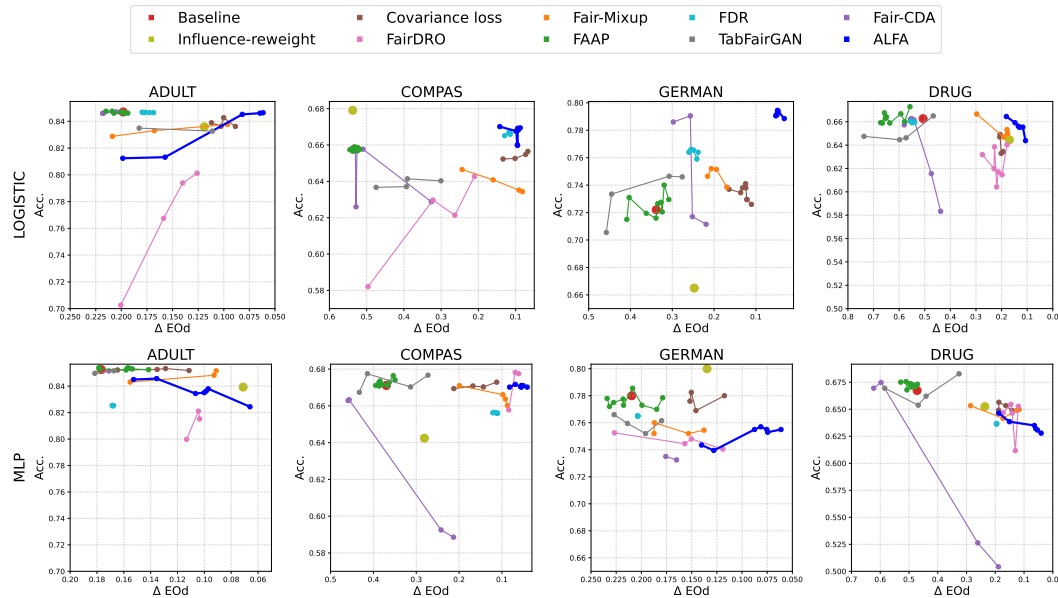

Figure 4: Fairness-Accuracy trade-off for Logistic Regression (top) and MLP (bottom) on tabular datasets. The x-axis shows $\Delta EOd$, where smaller values (to the right) indicate better fairness. Thus, the upper-right region reflects better performance. **ALFA** consistently outperforms other methods, achieving lower $\Delta EOd$ with minimal impact on accuracy.

where $\boldsymbol{X}_c$ is the original dataset and $\boldsymbol{Z}_p$ is the upsampled feature dataset to be attacked, respectively.

**Theorem 3.2.** *Retraining the classifier using Eq. 5 results in a fairer classifier by reducing $\Delta EOd$:*

$$\Delta EOd(\theta_p) \leq \Delta EOd(\theta). \tag{6}$$

*where $\theta$ and $\theta_p$ denote the classifier's parameter when trained on original dataset and combined dataset, respectively. The detailed proof of Theorem 3.2 is provided in Appendix A.*

In the neural networks, the encoder and the last layer are easily defined. However, in the Logistic Regression, there's no encoder is defined. As a special case, in the Logistic Regression, the linear classifier is pre-trained in the same manner to produce adversarial samples and trained again with our data augmentation, while the perturbation is conducted on the input space. The detailed algorithm is introduced in Algorithm 1.

## 4 EXPERIMENTAL DETAIL

### 4.1 DATASET

In this paper, we use four different tabular datasets Adult (Dua et al., 2017), COMPAS (Jeff Larson & Angwin, 2016), German (Dua et al., 2017), and Drug (Dua et al., 2017). Also CelebA (Liu et al., 2018) and Wikipedia Toxicity (Thain et al., 2017) datasets are used for verify the performance of the proposed method in image and text classification, respectively. All datasets are split into 60:20:20 for train, validation, and test subset, respectively. The detailed description of datasets is in Appendix I.

### 4.2 EXPERIMENTAL SETUP

To verify our approach, we apply our method to two base classifiers for tabular datasets, Logistic Regression and MLP with ReLU activation function and two hidden layers of 128 dimensions. For the CelebA dataset, we adopt ResNet-50 (He et al., 2016), ViT (Dosovitskiy, 2020), and Swin Transformer (Liu et al., 2021) as baselines. For the Wiki dataset, we use LSTM (Hochreiter & Schmidhuber, 1997), BERT (Devlin, 2018), and DistillBERT (Sanh, 2019) as baselines. During the pre-training, we choose the best parameter when the validation accuracy is the highest. In the

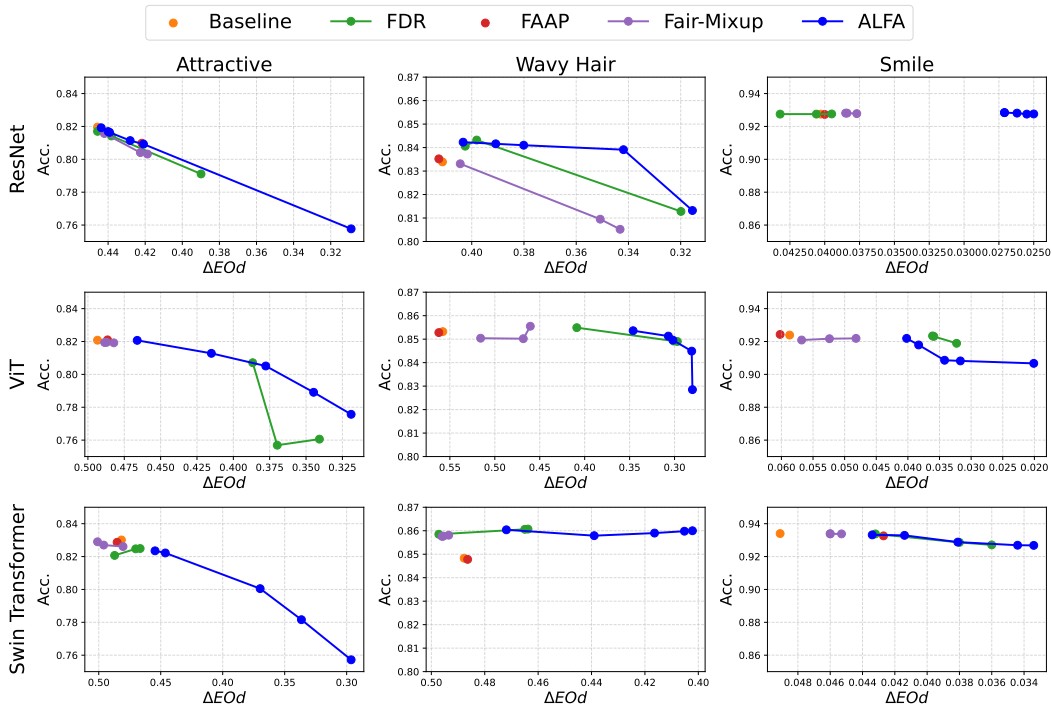

Figure 5: Fairness-Accuracy trade-off for the CelebA dataset. Each row of subfigures represents a different backbone network: ResNet, ViT, and Swin Transformer, while each column corresponds to a different target attribute for classification. Similar to the results on tabular datasets, **ALFA** consistently outperforms other methods in fairness without compromising accuracy.

attacking step, parameters of both the encoder and classifier are fixed, while only the last layer is newly initialized for fine-tuning with the augmented latent features. The different learning rates are used in each step, Adam optimizer with learning rate $1e-3$ in pre-training and fine-tuning, Adam optimizer with learning rate 0.1 in adversarial attack. For each experiment, we take the result when the validation accruacy is the highest. For a fair comparison, we train each case 10 times and report the mean and the standard deviation for tabular datasets and text dataset.

To evaluate the fairness improvement of our method, we compare its performance against other approaches using data augmentation, fairness constraints, data reweighing, or latent space manipulation methods, such as Covariance Loss (Zafar et al., 2017), Fair-Mixup (Mroueh et al., 2021), FDR (Mao et al., 2023), Fair-CDA (Sun et al., 2023), Influence-Reweighing (Li & Liu, 2022), FairDRO (Jung et al., 2023), FAAP (Wang et al., 2022), and TabFairGAN (Rajabi & Garibay, 2022), as shown in Figure 4. For a fair comparison, we follow each method's implementations, and adjust hyperparameters as detailed in Appendix G. For the CelebA and Wiki datasets, we adopt Fair-Mixup, FAAP, and FDR as comparisons which can operate in the latent space.

Furthermore, we acknowledge the issue of fairness in various tasks, including multi-class, multi-label, and handling multi-sensitive attributes scenarios. These tasks can be considered variants of binary classification, making ALFA applicable to them. We present the extension of ALFA and provide experimental results in Appendix J. Moreover, we discuss the applicability of our framework as input perturbation on neural networks in Appendix K.

## 4.3 RESULT ANALYSIS

### 4.3.1 ACCURACY-FAIRNESS TRADE-OFF

Figure 4,5 and 6 illustrate the trade-off between $\Delta EOd$ and accuracy. Since each comparison method, including ALFA, involves multiple hyperparameters, we conduct extensive experiments for each,

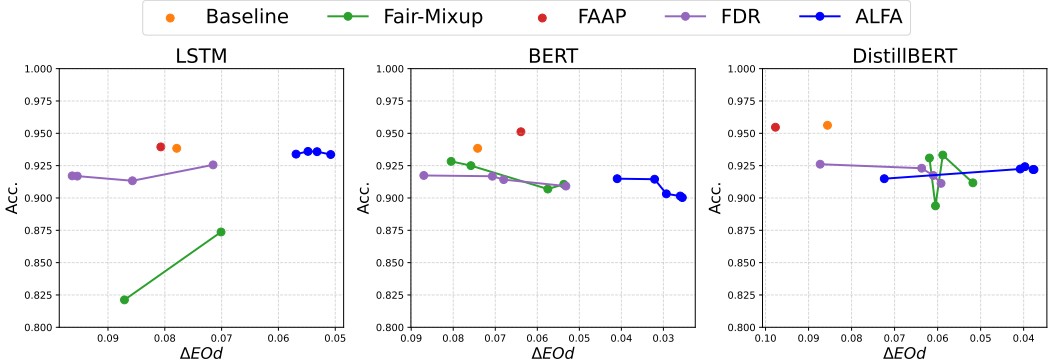

Figure 6: Fairness-Accuracy trade-off for Wikipedia dataset. Each column of subfigure represents different backbone networks, LSTM, BERT and DistillBERT. Similar to the tabular datasets, **ALFA** consistently outperfroms other methods in fairness without compromising accuracy.

displaying the results as line plots. For ALFA, the only hyperparameter used to generate the plot is the weight of the Sinkhorn distance in the fairness attack, denoted by $\alpha$.

Across datasets with different modalities, including tabular, image, and text, and with various backbone networks, ALFA consistently outperforms other methods. While ALFA may not always achieve the top performance, it consistently ranks either first or second across all comparison methods, with no other approach demonstrates such a high level of overall performance. Notably, in the COMPAS, German, Drug, CelebA, and Wikipedia datasets, across all backbone networks, ALFA achieves significant improvement in $\Delta EOd$ with minimal impact on accuracy.

In addition to the trade-off plots, detailed experimental results, including the standard deviation of quantitative outcomes, are presented in Appendix H, further highlighting ALFA's consistent fairness improvements. Moreover, Appendix M presents an in-depth analysis of the comparison methods shown in Figure 3 focusing on the differences in approach between FAAP, Fair-Mixup, and ALFA.

### 4.3.2 ABLATION STUDY

We visualize the impact of the hyperparameter $\alpha$, which controls the weight of the Sinkhorn distance in the fairness attack. Intuitively, a larger $\alpha$ helps preserve the original distribution of the perturbed features, maintaining the accuracy of the fine-tuned classifier, as shown in Figure 7 (a). On the other hand, a small or zero $\alpha$ may alter the semantic meaning of the perturbed features, potentially impacting accuracy, as also shown in Figure 7 (a). While the relationship is not perfectly linear, smaller values of $\alpha$ generally improve fairness, introducing a trade-off between fairness and accuracy. This suggests that the Sinkhorn distance helps balance the two objectives, as shown in both subfigures of Figure 7.

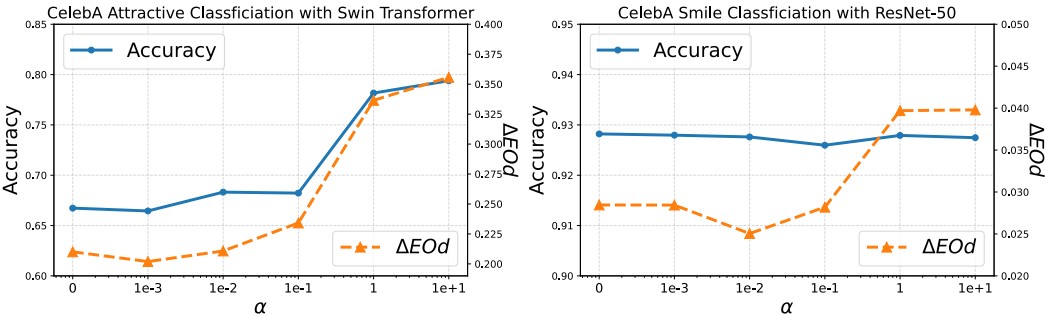

Figure 7: Ablation study varying $\alpha$ from 0 to 10 for the CelebA dataset across different tasks and backbone networks. While not entirely consistent, smaller $\alpha$ generally improves fairness but may negatively impact accuracy compared to larger $\alpha$.

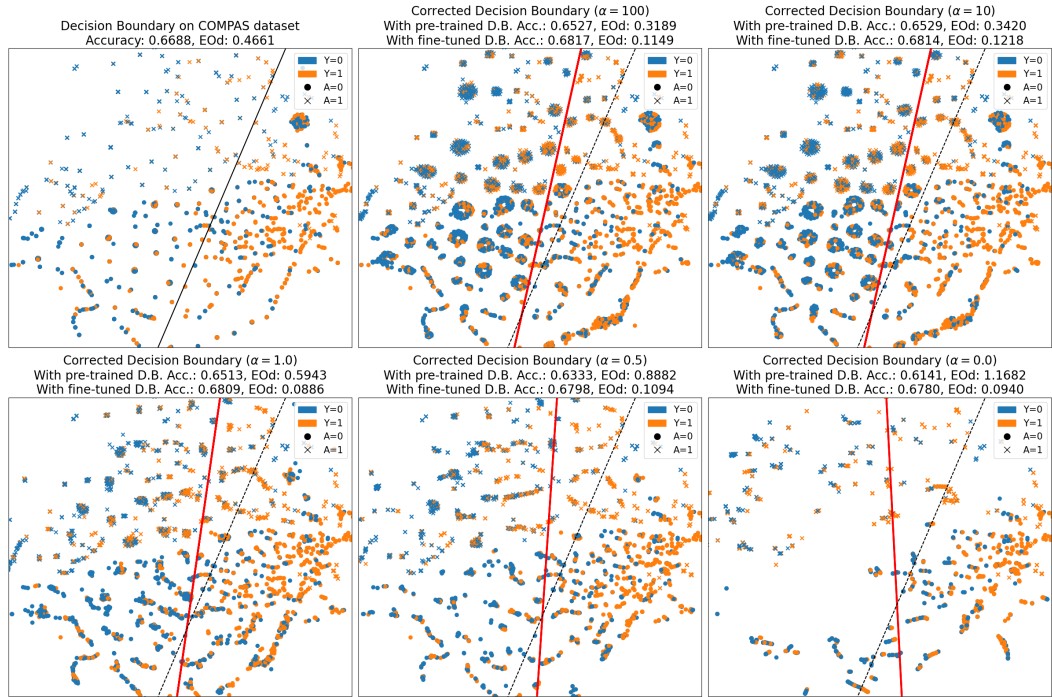

Figure 8: t-SNE plots for the COMPAS dataset. The black line represents the pre-trained decision boundary, while the red line represents the newly trained decision boundary on the combined dataset, where equal weighting is applied to the original and each perturbed dataset.

### 4.3.3 Visualization of Decision Boundary Correction with Real-world Dataset

For deeper analysis, we provide t-SNE plots for the COMPAS dataset, including the original dataset and the perturbed datasets (with $\alpha = 0$ and $\alpha = 1$, respectively). The visualization in Figure 8 reveals that under the pre-trained decision boundary, the perturbed samples exhibit extremely high $\Delta$EOd, indicating the success of our fairness attack. However, fine-tuning on the concatenated dataset results in a corrected decision boundary (represented by the red line) that maintains accuracy while achieving significant improvements in fairness.

Moreover, the effect of $\alpha$ aligns with our intuition. A higher $\alpha$ retains the original distribution more closely, resulting in a less corrected decision boundary and a less pronounced fairness attack. Nevertheless, both cases ($\alpha = 0$ and $\alpha = 1$) demonstrate significant improvements in fairness after fine-tuning.

## 5 Conclusion

In this research, we address the critical issue of fairness in machine learning models, specifically focusing on biases caused by demographic data imbalances. We propose a novel method, Adversarial Latent Feature Augmentation (ALFA), to effectively identify and mitigate unfairness in classification models, promoting more equitable decision-making. ALFA generates biased perturbed features using a fairness attack based on a fairness constraint. Fine-tuning the classifier on these biased samples reduces discrepancies in misclassification rates across different demographic groups. We provide theoretical proof of our claims, and our method is validated through extensive experiments on a wide range of datasets, modalities, and backbone networks. ALFA consistently achieves group fairness without compromising accuracy, demonstrating its effectiveness in promoting unbiased machine learning models. As future work, we aim to explore integrating individual fairness metrics and extending ALFA to larger, more complex datasets to assess its scalability and broader applicability.

ACKNOWLEDGEMENTS

This work was partially supported by the EMBRIO Institute, contract #2120200, a National Science Foundation (NSF) Biology Integration Institute, Purdue's Elmore ECE Emerging Frontiers Center, and NSF IIS #1955890, IIS #2146091, IIS #2345235.

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

## A    PROOF FOR THEOREM 3.2

*Consider a linear classifier $g(z) = sign(w^T z + b)$, where $z = f(x)$ represents the latent space generated by an encoder $f$, and $w$ and $b$ are the classifier's weights and bias. The classifier $g(z)$ is considered biased if its decision boundary $w^T z + b = 0$ results in significantly different outcomes (e.g., higher misclassification rates) for the groups defined by the sensitive attribute $A \in \{0, 1\}$. We define the unfair region $R_{unfair}$ as the subspace of the latent space where the classifier's decision boundary results in disproportionately high differences in error rates between these groups:*

$$R_{\text{unfair}} = \big\{ z \in \mathbb{R}^d : \big|P(g(z) = y|A = 1)| - P(g(z) = y|A = 0)\big| > \tau \big\} \, \forall y \in \{0, 1\}, \quad (7)$$

*where $\tau$ is a threshold indicating significant bias.*

*Retraining the classifier using Eq. 5 results in a fairer classifier by reducing $\Delta EOd$:*

$$\Delta EOd(\theta_p) \leq \Delta EOd(\theta). \quad (8)$$

*where $\theta$ and $\theta_p$ denote the classifier's parameter when trained on original dataset and combined dataset, respectively.*

*Proof.* When the classifier exhibits bias, the decision boundary poorly separates classes within $R_{\text{unfair}}$. To address this, we retrain the classifier by focusing on the unfair region with the generated samples by fairness attack, minimizing a new loss function:

$$L(w, b) = \frac{1}{N'} \Big( \sum_{i \in D} L_{ce}(y_i, g(z_i)) + \sum_{j \in R_{\text{unfair}}} L_{ce}(y_j, g(z_j)) \Big), \quad (9)$$

where $D$ represents the original dataset, $L_{ce}$ is the cross-entropy loss, and $N'$ is the size of the combined dataset. In Eq. 9, the gradient of $L$ is more heavily influenced by the samples in $R_{\text{unfair}}$ than it would be if the model were trained solely on $D$. This induces an upweighting effect in the unfair region during retraining, leading to an adjustment of the decision boundary, which can be specifically approximated by:

$$\Delta w = -\eta \nabla L(w, b)$$

where $\eta$ is the learning rate. The adjustment $\Delta w$, driven by the samples in $R_{\text{unfair}}$, reduces the misclassification rates in this region, particularly benefiting the disadvantaged group.

Let's define $R_{\text{unfair}}^{y=1}$, the unfair region for $y = 1$ such that

$$R_{\text{unfair}}^{y=1} = \big\{ z \in \mathbb{R}^d : \big|P(g(z) = 1|A = 1)| - P(g(z) = 1|A = 0)\big| > \tau \big\}$$

As the decision boundary adjusts to correct misclassifications within $R_{\text{unfair}}^{y=1}$, the impact is most significant for the group with higher error rates. This leads to a reduction in the False Positive Rate (FPR) within $R_{\text{unfair}}^{y=1}$, especially if a sensitive group initially exhibits a disproportionately high FPR when predictions in $R_{\text{unfair}}^{y=1}$ result in $g(z) = 1$. Consequently, this reduces the FPR gap:

$$\big|P(g(z) = 1|A = 1)| - P(g(z) = 1|A = 0)\big|.$$

Similarly, we can derive the reduction in the TPR gap. When considering the unfair region $R_{\text{unfair}}^{y=0}$ in relation to the False Negative Rate (FNR), a sensitive group may exhibit a disproportionately high FNR, resulting in $g(z) = 0$:

$$R_{\text{unfair}}^{y=0} = \big\{ z \in \mathbb{R}^d : \big|P(g(z) = 0|A = 1)| - P(g(z) = 0|A = 0)\big| > \tau \big\}.$$

A high FNR gap is problematic because it implies a higher gap in True Positive Rate (TPR), given that $TPR_a = 1 - FNR_a, |TPR_0 - TPR_1| = |(1 - FNR_0) - (1 - FNR_1)| = |FNR_1 - FNR_0|$. Therefore, the unfair region $R_{\text{unfair}}^{y=0}$ becomes a critical target area to address for fairness. Similar to $R_{\text{unfair}}^{y=1}$ for FPR gap, retraining that focuses on $R_{\text{unfair}}^{y=0}$ also leads to a reduction in the TPR gap.

Therefore, after retraining on samples in $R_{\text{unfair}}^{y=1}$ and $R_{\text{unfair}}^{y=0}$, the following inequalities hold:

$$|FPR_0 - FPR_1|_{retrain} \leq |FPR_0 - FPR_1|_{origin}$$

and

$$|TPR_0 - TPR_1|_{retrain} \leq |TPR_0 - TPR_1|_{origin}.$$

Consequently, retraining on samples within the unfair regions identified by the fairness attack leads to a reduction in both the FPR and TPR disparities across demographic groups.

As the Equalized Odds gap $\Delta EOd$ is defined as the sum of the TPR and FPR gaps:

$$\Delta EOd = |TPR_1 - TPR_0| + |FPR_1 - FPR_0|,$$

retraining on samples from the unfair regions ensures a fairer classifier by minimizing these gaps:

$$\Delta EOd(\theta_p) \leq \Delta EOd(\theta). \tag{10}$$

where $\theta$ and $\theta_p$ denote the classifier's parameter when trained on original dataset and combined dataset, respectively. This strategy not only addresses the existing bias but also actively improves the fairness of the model in a measurable and theoretically grounded manner. $\qquad\square$

## B  PROPERTIES

### B.1  PROPORTIONALITY OF $\mathcal{L}_{\text{FAIR}}$

**Proposition B.1.** $\mathcal{L}_{fair}$ *is proportional to the mean signed distance gap ($\Delta d_{dp}$) between two sensitive attribute groups, and the sum of the mean signed distance gap ($\Delta d_{eod,y}$) between the sensitive groups for each ground truth label $y \in \{0, 1\}$,*

$$\mathcal{L}_{fair} = \frac{1}{4}\Delta d_{dp} = \frac{1}{8}\Big[\Delta d_{eod,1} + \Delta d_{eod,0}\Big],$$

*where $\Delta d_{dp} = \Big|\frac{1}{n_1}\sum_{i \in S_1} d_i - \frac{1}{n_0}\sum_{j \in S_0} d_j\Big|$, $\Delta d_{eod,1} = \Big|\frac{1}{n_{11}}\sum_{i \in S_{11}} d_i - \frac{1}{n_{01}}\sum_{j \in S_{01}} d_j\Big|$, and $\Delta d_{eod,0} = \Big|\frac{1}{n_{10}}\sum_{i \in S_{10}} d_i - \frac{1}{n_{00}}\sum_{j \in S_{00}} d_j\Big|$.*

*Proof.* Let $d_i = g(\tilde{z})$, $\bar{d}$ is the mean of all $d_i$ and

$$\Delta d_{dp} = \Big|\frac{1}{n_1}\sum_{i \in S_1} d_i - \frac{1}{n_0}\sum_{j \in S_0} d_j\Big|$$

$$\Delta d_{eod,1} = \Big|\frac{1}{n_{11}}\sum_{i \in S_{11}} d_i - \frac{1}{n_{01}}\sum_{j \in S_{01}} d_j\Big|$$

$$\Delta d_{eod,0} = \Big|\frac{1}{n_{10}}\sum_{i \in S_{10}} d_i - \frac{1}{n_{00}}\sum_{j \in S_{00}} d_j\Big|$$

where $S_a$ is a subset containing each sensitive attributes $S_a = \{i|a_i = a\}$, $a \in \{0, 1\}$, and $n_{ay}$ means the number of samples for each sensitive subset for given $y$, $S_{ay} = \{i|a_i = a, y_i = y\}$, $a \in \{0, 1\}$ and $y \in \{0, 1\}$. In our experiments, we select samples with the same size such that $\frac{N}{4} = n_{00} = n_{01} = n_{10} = n_{11}$, and $\frac{N}{2} = n_0 = n_1$ where $n_0 = n_{00} + n_{01}$, $n_1 = n_{10} + n_{11}$, and $N = n_{00} + n_{01} + n_{10} + n_{11}$.

The objective function $\mathcal{L}_{\text{fair}} = |Cov(a, g(\tilde{z}))|$ can be rewritten as

$$\begin{aligned}
\mathcal{L}_{\text{fair}} &= \frac{1}{N}\Big|\sum_{i=1}^{N}(a_i - \bar{a})(d_i - \bar{d})\Big| \\
&= \frac{1}{N}\Big|\sum_{i \in S_1}(1 - \bar{a})(d_i - \bar{d}) + \sum_{j \in S_0}(0 - \bar{a})(d_j - \bar{d})\Big| \\
&= \frac{1}{N^2}\Big|n_0\sum_{i \in S_1}(d_i - \bar{d}) - n_1\sum_{j \in S_0}(d_j - \bar{d})\Big|
\end{aligned}$$

$$= \frac{1}{N^2}\left|n_0\sum_{i\in S_1}d_i - n_1\sum_{j\in S_0}d_j - n_0 n_1 \bar{d} + n_0 n_1 \bar{d}\right|$$

$$= \frac{n_0 n_1}{N^2}\left|\frac{1}{n_1}\sum_{i\in S_1}d_i - \frac{1}{n_0}\sum_{j\in S_0}d_j\right|$$

$$= \frac{1}{4}\Delta d_{dp}. \tag{11}$$

Similarly, we can conditionize $\mathcal{L}_{\text{fair}}$ in terms of $y$,

$$\mathcal{L}_{\text{fair}} = \frac{1}{N}\left|\sum_{i=1}^N (a_i - \bar{a})(d_i - \bar{d})\right|$$

$$= \frac{1}{N}\left|\sum_{i\in S_{11}}(1-\bar{a})(d_i-\bar{d}) + \sum_{j\in S_{01}}(0-\bar{a})(d_j-\bar{d}) + \sum_{i\in S_{10}}(1-\bar{a})(d_i-\bar{d}) + \sum_{j\in S_{00}}(0-\bar{a})(d_j-\bar{d})\right|$$

$$= \frac{1}{N}\left|\sum_{i\in S_{11}}\frac{(n_{01}+n_{00})}{N}(d_i-\bar{d}) - \sum_{j\in S_{01}}\frac{(n_{11}+n_{10})}{N}(d_j-\bar{d})\right.$$

$$\left. + \sum_{i\in S_{10}}\frac{(n_{01}+n_{00})}{N}(d_i-\bar{d}) - \sum_{j\in S_{00}}\frac{(n_{11}+n_{10})}{N}(d_j-\bar{d})\right|$$

$$= \left|\frac{n_{11}(n_{01}+n_{00})}{N^2}\frac{1}{n_{11}}\sum_{i\in S_{11}}d_i - \frac{n_{01}(n_{11}+n_{10})}{N^2}\frac{1}{n_{01}}\sum_{j\in S_{01}}d_j\right.$$

$$\left. + \frac{n_{10}(n_{01}+n_{00})}{N^2}\frac{1}{n_{10}}\sum_{i\in S_{10}}d_i - \frac{n_{00}(n_{11}+n_{10})}{N^2}\frac{1}{n_{00}}\sum_{j\in S_{00}}d_j\right|$$

$$= \frac{n_{11}n_0}{N^2}\left[\left|\frac{1}{n_{11}}\sum_{i\in S_{11}}d_i - \frac{1}{n_{01}}\sum_{j\in S_{01}}d_j\right| + \left|\frac{1}{n_{10}}\sum_{i\in S_{10}}d_i - \frac{1}{n_{00}}\sum_{j\in S_{00}}d_j\right|\right]$$

$$= \frac{1}{8}\left[\Delta d_{eod,1} + \Delta d_{eod,0}\right] \tag{12}$$

$\square$

## B.2 BOUNDEDNESS OF $\mathcal{L}_{\text{FAIR}}$

**Theorem B.2.** *If the cardinalities of subgroups $S_{ay} = \{i|a_i = a, y_i = y\}$, $a \in \{0,1\}, y \in \{0,1\}$ are equal, $\mathcal{L}_{fair}$ is the lower bound of $\Delta DP$ and $\Delta EOd$ when we approximate the logit (inverse sigmoid) function as a piecewise linear function with $m$ segments s.t $d = f_k(\hat{y}) = a_k\hat{y} + b_k$ for $k \in \{1, 2, \cdots, m\}$, $m > 1, m \in \mathbb{N}$, and $a_{max} = \max(a_1, \cdots, a_k)$. Then,*

$$\mathcal{L}_{fair} \leq \frac{1}{4}\left(a_{max}\Delta DP + C\right), \tag{13}$$

$$\mathcal{L}_{fair} \leq \frac{1}{8}\left(a_{max}\Delta EOd + C_0 + C_1\right). \tag{14}$$

$C = \frac{2}{N}\sum_{k=1}^m (n_1^{(k)} - n_0^{(k)})b_k$, and $C_a = \frac{4}{N}\sum_{k=1}^m (n_1^{(k)} - n_0^{(k)})b_k$ are constants where $n_a^{(k)}$ is the number of samples in $k$-th segment for $a \in \{0, 1\}$. We set $\beta \leq \hat{y} \leq 1 - \beta$ when we compute the signed distance to avoid $d_\theta = \ln\left(\frac{\hat{y}}{1-\hat{y}}\right) \to (\infty \text{ or } -\infty)$ so that $a_{\max} \leq \frac{m}{1-2\beta}\left[\text{logit}(\beta + \frac{1-2\beta}{m}) - \text{logit}(\beta)\right]$, theoretically. In this work, we set $\beta = 1e{-}7$ and $m = 10$ for all experiments.

*Proof.* As we fix the sign of $\mathcal{L}_{\text{fair}}$ following the sign of $Cov(a, y)$, the sign of $\Delta d_{dp}$ and $\Delta DP$ are particularly defined as

$$\Delta d_{dp} = \left|\frac{1}{n_1}\sum_{i\in S_1}d_i - \frac{1}{n_0}\sum_{j\in S_0}d_j\right|$$

$$= \frac{1}{n_1} \sum_{i \in S_1} d_i - \frac{1}{n_0} \sum_{j \in S_0} d_j$$

$$\Delta DP = \left| \frac{1}{n_1} \sum_{i \in S_1} \hat{y}_i - \frac{1}{n_0} \sum_{j \in S_0} \hat{y}_j \right|$$

$$= \frac{1}{n_1} \sum_{i \in S_1} \hat{y}_i - \frac{1}{n_0} \sum_{j \in S_0} \hat{y}_j$$

when we assume that $Cov(a, y)$ is positive. In the negative case, the sign of $\Delta d_{dp}$ and $\Delta DP$ will be changed simultaneously.

If we assume the logit function as a piecewise linear function with $m$ segments s.t $m > 1, m \in \mathbb{N}$, and recall that $n_{dp} = n_0 = n_1$ and $\frac{N}{4} = n_{00} = n_{01} = n_{10} = n_{11}$. Let each linear function is $d = f_k(\hat{y}) = a_k \hat{y} + b_k$, $k = 1, 2, \cdots, m$. Then the $\Delta d_{dp}$ and $\Delta DP$ becomes

$$\Delta d_{dp} = \frac{1}{n_1} \sum_{i \in S_1} d_i - \frac{1}{n_0} \sum_{j \in S_0} d_j$$

$$= \sum_{k=1}^{m} \left[ \frac{1}{n_1} \sum_{i \in S_1^{(k)}} (a_k \hat{y}_i + b_k) - \frac{1}{n_0} \sum_{j \in S_0^{(k)}} (a_k \hat{y}_j + b_k) \right]$$

$$= \frac{1}{n_{dp}} \sum_{k=1}^{m} a_k \left( \sum_{i \in S_1^{(k)}} \hat{y}_i - \sum_{j \in S_0^{(k)}} \hat{y}_j \right)$$

$$+ \frac{1}{n_{dp}} \sum_{k=1}^{m} (n_1^{(k)} - n_0^{(k)}) b_k$$

$$\leq \frac{a_{\max}}{n_{dp}} \left[ \sum_{k=1}^{m} \left( \sum_{i \in S_1^{(k)}} \hat{y}_i - \sum_{j \in S_0^{(k)}} \hat{y}_j \right) \right] + C$$

$$= \frac{a_{\max}}{n_{dp}} \left[ \sum_{i \in S_1} \hat{y}_i - \sum_{j \in S_0} \hat{y}_j \right] + C$$

$$= a_{\max} \Delta DP + C \tag{15}$$

where $n_a^{(k)}$ means the number of samples in $k$-th segment for $a \in \{0, 1\}$ and $C = \frac{1}{n_{dp}} \sum_{k=1}^{m} (n_1^{(k)} - n_0^{(k)}) b_k$ is a constant. Therefore, maximizing $\mathcal{L}_{\text{fair}}$ maximizes $\Delta DP$ since

$$\mathcal{L}_{\text{fair}} = \frac{1}{4} \Delta d_{dp} \leq \frac{1}{4} \left( a_{\max} \Delta DP + C \right). \tag{16}$$

Similarly, the same proof can be applied to the relationship between $\Delta d_{eod,0}$, $\Delta d_{eod,1}$, and $\Delta EOd$ as explained in Eq. 15, such that

$$\Delta EOd = \left| \frac{1}{n_{11}} \sum_{i \in S_{11}} \hat{y}_i - \frac{1}{n_{01}} \sum_{j \in S_{01}} \hat{y}_j \right| + \left| \frac{1}{n_{10}} \sum_{i \in S_{10}} \hat{y}_i - \frac{1}{n_{00}} \sum_{j \in S_{00}} \hat{y}_j \right|$$

$$= \frac{1}{n_{eod}} \left[ \left( \sum_{i \in S_{11}} \hat{y}_i - \sum_{j \in S_{01}} \hat{y}_j \right) + \left( \sum_{i \in S_{10}} \hat{y}_i - \sum_{j \in S_{00}} \hat{y}_j \right) \right]$$

$$\Delta d_{eod,1} = \left| \frac{1}{n_{11}} \sum_{i \in S_{11}} d_i - \frac{1}{n_{01}} \sum_{j \in S_{01}} d_j \right|$$

$$= \frac{1}{n_{eod}} \left[ \sum_{i \in S_{11}} d_i - \sum_{j \in S_{01}} d_j \right] \leq \frac{a_{\max}}{n_{eod}} \left[ \sum_{i \in S_{11}} \hat{y}_i - \sum_{j \in S_{01}} \hat{y}_j \right] + C_1$$

$$\Delta d_{eod,0} = \left| \frac{1}{n_{10}} \sum_{i \in S_{10}} d_i - \frac{1}{n_{00}} \sum_{j \in S_{00}} d_j \right|$$

$$= \frac{1}{n_{eod}} \Big[ \sum_{i \in S_{10}} d_i - \sum_{j \in S_{00}} d_j \Big] \leq \frac{a_{\max}}{n_{eod}} \Big[ \sum_{i \in S_{10}} \hat{y}_i - \sum_{j \in S_{00}} \hat{y}_j \Big] + C_0$$

Therefore,

$$
\begin{aligned}
\mathcal{L}_{fair} &= \frac{1}{8} \Big[ \Delta d_{eod,1} + \Delta d_{eod,0} \Big] \\
&\leq \frac{1}{8} \Big[ \frac{a_{\max}}{n_{eod}} \Big[ \big( \sum_{i \in S_{11}} \hat{y}_i - \sum_{j \in S_{01}} \hat{y}_j \big) + \big( \sum_{i \in S_{10}} \hat{y}_i - \sum_{j \in S_{00}} \hat{y}_j \big) \Big] + C_0 + C_1 \Big] \\
&= \frac{1}{8} \Big[ a_{\max} \Delta EOd + C_0 + C_1 \Big]
\end{aligned}
\tag{17}
$$

where $C_a = \sum_{k=1}^{m} (n_1^{(k)} - n_0^{(k)}) b_k, a \in \{0, 1\}$ are constants. $\qquad \square$

### B.2.1 USAGE OF PIECEWISE LINEAR APPROXIMATION

We empirically verify that the naive logit function is feasible as well and effectively attacks the fairness in terms of $\Delta DP$ and $\Delta EOd$. However, the upper bound of $\mathcal{L}_{\text{fair}}$ with the naive logit function is not fully supported mathematically, while the piecewise linear logit function can be proved as Appendix B.2. Moreover, there's no significant difference in the fairness performances between the naive logit function and its piecewise linear approximation. We choose the piecewise linear function to ensure the upper bound of $\mathcal{L}_{\text{fair}}$.

### B.2.2 INSIGHTS

We randomly choose an equal number of samples for each subset for effective fairness attack, i.e. $\frac{N_p}{4} = n_{00} = n_{01} = n_{10} = n_{11}$ to satisfy the condition in Theorem B.2. Consequently, since $\mathcal{L}_{\text{fair}}$ is the lower bound of $\Delta DP$ and $\Delta EOd$, we can attack fairness by maximizing $\mathcal{L}_{\text{fair}}$ as the perturbed latent features produce unfair prediction with high $\Delta DP$ and $\Delta EOd$ on given pre-trained classifier.

## C SYNTHETIC DATASET

We provide the details of the synthetic data, illustrating the concept of the unfair region and how the decision boundary is corrected. We simplify the binary classification task with a 2D Gaussian mixture model, as assumed in (Xu et al., 2021), consisting of two classes $y \in \{0, 1\}$ and two sensitive attributes $A \in \{0, 1\}$ (indicating unprivileged and privileged groups).

$$
x \sim
\begin{cases}
group1 : \mathbf{N}\big( \begin{bmatrix} \mu \\ \mu \end{bmatrix}, \sigma^2 \big) & \text{if} : y = 1, a = 1 \\[2mm]
group2 : \mathbf{N}\big( \begin{bmatrix} \mu \\ \mu' \end{bmatrix}, \sigma^2 \big) & \text{if} : y = 0, a = 1 \\[2mm]
group3 : \mathbf{N}\big( \begin{bmatrix} 0 \\ \mu \end{bmatrix}, (K\sigma)^2 \big) & \text{if} : y = 1, a = 0 \\[2mm]
group4 : \mathbf{N}\big( \begin{bmatrix} 0 \\ 0 \end{bmatrix}, (K\sigma)^2 \big) & \text{if} : y = 0, a = 0
\end{cases}
\tag{18}
$$

where $\mu' = r\mu, 0 < r < 1$ and $K > 1$, where the number of samples in each group is $N_1 : N_2 : N_3 : N_4$. We arbitrarily set $K = 3$, $r = 0.7$, $\mu = 1$, $N_1 = N_2 = 100$, and $N_3 = N_4 = 400$. From the synthetic data, we observe a decision boundary like Figure 2 (a) in the paper. Due to dataset imbalance, the subgroup $a = 1, y = 0$ is overestimated as label $y = 1$, and the subgroup $a = 0, y = 1$ is underestimated as label $y = 0$. The disparity in misclassification rates is depicted in Figure 2 (c). We define these disparities as 'unfair regions' where the misclassification rate is disproportionately high.

## D  SINKHORN DISTANCE

### D.1  DEFINITION

Optimal transport with lowest cost is defined as $\mathcal{L}_{\boldsymbol{C}} = \min_{\boldsymbol{P}} \sum_{i,j} \boldsymbol{C}_{ij} \boldsymbol{P}_{ij}$, where $\boldsymbol{C}$ is a *cost matrix* (2-Wasserstein Distance), and $\boldsymbol{P}$ is the *coupling matrix*. Genevay et al. (2018) suggested a regularized optimal transport scheme which includes entropy term to secure stability and smoothness of $\boldsymbol{P}$, $\mathcal{L}_{\boldsymbol{C}} = \min_{\boldsymbol{P}} \sum_{i,j} \boldsymbol{C}_{ij} \boldsymbol{P}_{ij} - \epsilon_s H(\boldsymbol{P})$ where $H(\boldsymbol{P}) = -\sum_{ij} \boldsymbol{P}_{ij} \log \boldsymbol{P}_{ij}$. $\mathcal{L}_{\boldsymbol{C}}$ can be solved by *Sinkhorn iteration*, s.t. $\boldsymbol{P}_{ij} = \mathrm{diag}(\boldsymbol{u}_i)\boldsymbol{K}_{ij}\mathrm{diag}(\boldsymbol{v}_j)$, and updated alternately,

$$\boldsymbol{u}^{(k+1)} = \frac{\boldsymbol{a}}{\boldsymbol{K}\boldsymbol{v}^{(k)}},$$

$$\boldsymbol{v}^{(k+1)} = \frac{\boldsymbol{b}}{\boldsymbol{K}^{\mathsf{T}}\boldsymbol{u}^{(k+1)}},$$

where $\boldsymbol{P}\boldsymbol{1} = \boldsymbol{a}, \boldsymbol{P}^{\mathsf{T}}\boldsymbol{1} = \boldsymbol{b}$, and Gibbs kernel $\boldsymbol{K}_{ij} = e^{-c_{ij}/\epsilon_s}$. Therefore, the distance between clean data $\boldsymbol{x}$ and perturbed data $\tilde{\boldsymbol{x}}$ can be rewritten as follows,

$$D(\boldsymbol{x}, \tilde{\boldsymbol{x}}) = \text{Sinkhorn Distance}(\boldsymbol{x}, \tilde{\boldsymbol{x}})$$

$$= \min_{\boldsymbol{P}(\boldsymbol{x}, \tilde{\boldsymbol{x}})} \sum_{i,j} \boldsymbol{C}_{ij}(\boldsymbol{x}, \tilde{\boldsymbol{x}}) \boldsymbol{P}_{ij}(\boldsymbol{x}, \tilde{\boldsymbol{x}}) - \epsilon_s H(\boldsymbol{P}(\boldsymbol{x}, \tilde{\boldsymbol{x}})).$$

### D.2  COMPUTATIONAL COST OF SINKHORN DISTANCE

The Sinkhorn distance involves solving a regularized optimal transport problem using the Sinkhorn-Knopp algorithm, which benefits from linear convergence rates and efficient GPU execution. This makes the computation of the Sinkhorn distance tractable even for large-scale problems.

This is further supported by the non-significant empirical time cost observed in ALFA. We report the attack times for each dataset. Each result represents the mean and standard deviation across 10 runs. Notably, the attack time is proportional to the dataset size, independent of the data type (e.g., images), as the attack occurs in the latent space rather than the input space. Note that this attack step is conducted only once before fine-tuning, enhancing the effectiveness of our framework.

Table 1: Time cost for one-time attacking step

| Dataset | Number of Samples | Feature Dimension | Attacking Times (s) |
|---|---|---|---|
| Adult | 271,320 | 512 | 15.9613±0.3242 |
| COMPAS | 43,280 | 512 | 2.6102±0.1050 |
| German | 6,000 | 512 | 0.2692±0.1112 |
| Drug | 11,280 | 512 | 0.6949±0.1114 |
| CelebA | 162,748 | 768 | 115.5459±2.4313 |
| Wikipedia | 342,528 | 384 | 377.0901±75.1673 |

## E  COVARIANCE BETWEEN THE LABEL AND THE SENSITIVE ATTRIBUTE

Table 2: The estimated value of $Cov(a, y)$ and $Cov(a, \hat{y})$. We set the sign of $\mathcal{L}_{\text{fair}}$ the same as the covariance.

|        | $Cov(a, y)$ | $Cov(a, \hat{y})$ | $\mathcal{L}_{\text{fair}}$ |
|--------|-------------|-------------------|-----------------------------|
| Adult  | 0.0439      | 0.0441            | $Cov(a, g(\tilde{z}))$      |
| COMPAS | -0.0198     | -0.0194           | $-Cov(a, g(\tilde{z}))$     |
| German | 0.0210      | 0.0188            | $Cov(a, g(\tilde{z}))$      |
| Drug   | 0.0434      | 0.0401            | $Cov(a, g(\tilde{z}))$      |

## F  HYPERPARAMETERS

Table 3: Hyperparameters for the experiments for ALFA

| Hyperparameter       | Search-range                  |
|----------------------|-------------------------------|
| $\alpha$             | [0.0, 0.001,0.01,0.1, 1.0, 10] |
| Total Epoch $T$      | 50                            |
| Attacking Iteration  | 10                            |
| Batch Size           | 128                           |

## G  EXPERIMENTAL SETTINGS FOR FAIR COMPARISON

**Fair-Mixup.** Fair-Mixup is an in-processing data augmentation using interpolation on manifold between two sensitive groups. Smooth regularizers for linear interpolation on DP and EOd are as follows

$$R_{mixup}^{DP} = \int_0^1 \left| \int \langle \nabla_x f(tg(x_0) + (1-t)g(x_1)), g(x_0) - g(x_1) \rangle dP_0(x_0)dP_1(x_1) \right| dt,$$

$$R_{mixup}^{EOd} = \sum_{y \in \{0,1\}} \int_0^1 \left| \int \langle \nabla_x f(tx_0 + (1-t)x_1), x_0 - x_1 \rangle dP_0^y(x_0)dP_1^y(x_1) \right| dt,$$

where $g : \mathcal{X} \to \mathcal{Z}$ is a feature encoder. The final objective function of Fair Mixup is

$$\mathcal{L}_{mixup} = L_{acc} + \lambda R_{mixup}(f).$$

For a fair comparison, we vary the ratio of regularizer adjusting $\lambda \in \{0.1, 0.3, 0.5, 0.7\}$ for tabular datasets and $\lambda = 20$ for CelebA dataset as suggested in the released implementation.

**TabFairGAN.** It aims to produce high-quality tabular data containing the same joint distribution as the original dataset using Wasserstein GAN. The training algorithm in (Rajabi & Garibay, 2022) consists of two phases, training for accuracy (phase 1) and training for both accuracy and fairness (phase 2). In both phases, the loss function for critics $C$ adopts gradient penalty (Gulrajani et al., 2017).

$$V_c = \mathbb{E}_{\hat{x} \sim P_g}[C(\hat{x})] - \mathbb{E}_{x \sim P_r}[C(x)] + \lambda_c \mathbb{E}_{\bar{x} \sim P_g}[(\|\nabla_{\bar{x}} C(\bar{x})\|_2 - 1)^2]$$

The loss function for the generator differs from each phase.

$$V_G = - \mathbb{E}_{\hat{x} \sim P_g}[C(\hat{x})] \tag{phase 1}$$

$$V_G = - \mathbb{E}_{\hat{x}, \hat{y}, \hat{a} \sim P_g}[C(\hat{x}, \hat{y}, \hat{a})] - \lambda_f (\mathbb{E}_{\hat{x}, \hat{y}, \hat{a} \sim P_g}[\hat{y}|\hat{a} = 0] - \mathbb{E}_{\hat{x}, \hat{y}, \hat{a} \sim P_g}[\hat{y}|\hat{a} = 1]) \tag{phase 2}$$

where $\lambda_f$ is hyperparameter. We excute TabFairGAN with various $\lambda_f \in \{0.1, 0.3, 0.5, 0.7\}$ for fair comparison since the implementation uses $\lambda_f = 0.5$.

**FAAP** FAAP aims to generate a perturbation using GANs, while the generator makes perturbation and the discriminator predicts the perturbed features' sensitive attributes. In formula,

$$\mathcal{L}_D = \mathcal{L}_{ce}(D(f(\hat{x}), a))$$
$$\mathcal{L}_G^{fair} = -\mathcal{L}_D - \alpha \mathcal{H}(D(f(\hat{x}), a))$$
$$\mathcal{L}_G^T = \mathcal{L}_{ce}(g(f(\hat{x})), y)$$

where $G$ is a generator, $D$ is a discriminator, $\hat{x}$ is the perturbed samples, $\mathcal{H}$ is the entropy, $g$ is label predictor (classifier), and $f$ is an encoder. The final formulation becomes

$$\arg \max_G \min_D \mathcal{L}_{ce}(D(g(\hat{x})), a) + \alpha \mathcal{H}(D(g(\hat{x})) - \beta \mathcal{L}_G^T$$

where $g(\hat{x}) = g(x + G(x))$. As the architectures for the generator and discriminator are not provided, we set a generator as an MLP model with two hidden layers with 128 nodes, having a ReLU activation function. For the discriminator, we adopt the same network with the label predictor in each tabular dataset and image dataset. For the fair comparison, we grid search $\alpha$ and $\beta$ by $\alpha \in \{0.1, 1.0, 10\}$ and $\beta \in \{0.1, 1.0, 10\}$ since the value is not given in the original paper.

**FDR** FDR is a simple fine-tuning method, including balanced sampling in the latent features, and use fairness constraint as a objective function. In detail, the Equalized-odds-based fairness constraint is

$$fpr = \left| \frac{\sum_i p_i(1 - y_i)a_i}{\sum_i a_i} - \frac{\sum_i p_i(1 - y_i)(1 - a_i)}{\sum_i(1 - a_i)} \right|$$
$$fnr = \left| \frac{\sum_i(1 - p_i)y_ia_i}{\sum_i a_i} - \frac{\sum_i(1 - p_i)y_i(1 - a_i)}{\sum_i(1 - a_i)} \right|$$

where $p_i$ denotes the predicted probability. The final objective function for the fine-tuning is

$$\min_\theta \left[ \mathcal{L}_{ce}(g) + \alpha(fpr + fnr) \right].$$

As suggested in the original paper, we search $\alpha \in \{0.5, 1, 2, 5, 10\}$.

**Fair-CDA** Fair-CDA aims to disentangle latent features into 'sensitive feature' and 'non-sensitive feature', and obfuscate the sensitive features to obtain a fairer classifier. Fair-CDA consists of three extractor, $h$, $h_y$, and $h_a$ as

$$z_i = h(x_i), z_i^y = h_y(z_i), z_i^a = h_a(z_i)$$

$h_y$ should extract features only related to the label predictions, while $h_a$ is related to the sensitive attribute only. The regularization becomes

$$\beta(\mathcal{L}_i^y + \mathcal{L}_i^a + \mathcal{L}_i^\perp)$$

and

$$\mathcal{L}_i^y = \mathcal{L}_{ce}(g_y(z_i^y), y_i)$$
$$\mathcal{L}_i^a = \mathcal{L}_{ce}(g_a(z_i^a), a_i)$$
$$\mathcal{L}_i^\perp = \frac{\langle \nabla_{z_i}\mathcal{L}_i^y, \nabla_{z_i}\mathcal{L}_i^a \rangle^2}{\|\nabla_{z_i}\mathcal{L}_i^y\|^2 \cdot \|\nabla_{z_i}\mathcal{L}_i^a\|^2}.$$

where $g_y$ and $g_a$ are two classifier for $y$ and $a$, respectively. In stage 1 for the first 450 epochs, the objective function is

$$\frac{1}{n}\sum_{i=1}^n \mathcal{L}_i + \beta(\mathcal{L}_i^y + \mathcal{L}_i^a + \mathcal{L}_i^\perp),$$

where $\mathcal{L}_i = \mathcal{L}_{ce}(g([z_i^y, z_i^a]), y_i)$. For stage 2, Fair-CDA conducts semantic augmentation to make the sensitive features along the direction to increase the attribute loss,

$$\tilde{z}_i^a = z_i^a + \alpha_i \frac{\nabla_{z_i^a}\mathcal{L}_{ce}(g_a(z_i^a), a_i)}{\|\nabla_{z_i^a}\mathcal{L}_{ce}(g_a(z_i^a), a_i)\|}$$

Based on the obtained $\tilde{z}_i^a$ and the solution of the task model in stage 1; $\hat{g}$, obtain two loss functions for stage 2 for 50 epochs,

$$\tilde{\mathcal{L}}_i = \mathcal{L}_{ce}(g([z_i^y, \tilde{z}_i^a]), y_i)$$
$$\hat{\mathcal{L}}_i = \mathcal{L}_{ce}(g([z_i^y, \tilde{z}_i^a]), \hat{g}([z_i^y, \tilde{z}_i^a])).$$

Then, the final objective function for stage 2 becomes

$$\frac{1}{n}\sum_{i=1}^{n}\gamma\tilde{\mathcal{L}}_i + (1-\gamma)\hat{\mathcal{L}}_i + \beta(\mathcal{L}_i^y + \mathcal{L}_i^a + \mathcal{L}_i^\perp).$$

Fair-CDA requires five hyperparameters, perturbation size $\alpha_i$ randomly drawn by $U(0, \lambda)$ where $\lambda \in \{0, 1, 10, 100, 1000\}$. $\gamma = 0.9$ as written in the paper, and $\beta$ is the initial loss value. As the learning rate for stages 1 and 2 are not given, we grid search $\eta_1, \eta_2 \in \{0.0001, 0.001, 0.01\}$ as well as $\lambda$.

**LAFTR** LAFTR (Madras et al., 2018) includes a classifier model named adversary, aiming to predict the sensitive attribute, while an encoder wants to deceive the adversary. It is also an adversarial training, but is different from ours as LAFTR does not involve perturbation, data augmentation, or fairness attack. Because of the architecture of LAFTR, it is only applicable to MLP baseline.

## H    DETAILS IN EXPERIMENTAL RESULTS

In this appendix, we quantitatively demonstrate the superiority of ALFA across all datasets and backbone networks. We also include results for a ResNet-like architecture applied to tabular datasets (Gorishniy et al., 2021) to show versability of ALFA. The results compare accuracy, $\Delta DP$, and $\Delta EOd$ with other fairness approaches. For the tabular and text datasets, the mean and standard deviation from 10 experiments are reported. In each table, Blue indicates the best result for each dataset, and Cyan represents the second and third best results. The results for each method are obtained by varying the hyperparameters, and for each method, we report the result corresponding to the point closest to the upper right in Figures 4, 5, and 6. The findings show that ALFA consistently ranks as either the best or second-best in terms of $\Delta EOd$ across all comparison methods, without compromising accuracy—a distinction not achieved by any other approach. This highlights ALFA's superiority over the alternatives.

### H.1    EXPERIMENTAL RESULTS OF TABULAR DATASETS

Table 4: Experimental Results for Adult dataset with Logistic Regression

| Adult | Logistic Regression | | |
|---|---|---|---|
| | Accuracy | $\Delta DP$ | $\Delta EOd$ |
| Baseline | 0.8470±0.0007 | 0.1829±0.0020 | 0.1982±0.0077 |
| Influence-Reweight | 0.8359±0.0000 | 0.1815±0.0000 | 0.1190±0.0000 |
| Covariance-Loss | 0.8427±0.0008 | 0.1174±0.0042 | 0.1001±0.0065 |
| Convex-Concave-Loss | 0.8296±0.0056 | 0.1194±0.0186 | 0.0687±0.0235 |
| Fair-DRO | 0.8012±0.0023 | 0.2123±0.0029 | 0.1261±0.0116 |
| Fair-Mixup | 0.8376±0.0022 | 0.1311±0.0125 | 0.0963±0.0103 |
| FAAP | 0.8461±0.0012 | 0.1822±0.0116 | 0.1937±0.0235 |
| FDR | 0.8465±0.0007 | 0.1797±0.0041 | 0.1685±0.0190 |
| TabFairGAN | 0.8327±0.0007 | 0.1488±0.0035 | 0.1112±0.0117 |
| Fair-CDA | 0.8469±0.0005 | 0.1839±0.0021 | 0.2013±0.0054 |
| Ours (ALFA) | 0.8464±0.0004 | 0.1555±0.0013 | 0.0616±0.0022 |

Table 5: Experimental Results for Adult dataset with MLP

| Adult | MLP | | |
|---|---|---|---|
| | **Accuracy** | $\Delta DP$ | $\Delta EOd$ |
| Baseline | 0.8525±0.0010 | 0.1824±0.0114 | 0.1768±0.0411 |
| LAFTR | 0.8470±0.0020 | 0.1497±0.0191 | 0.1117±0.0443 |
| Influence-Reweight | 0.8470±0.0020 | 0.1497±0.0191 | 0.1117±0.0443 |
| Covariance-Loss | 0.8518±0.0015 | 0.1568±0.0159 | 0.1114±0.0454 |
| Convex-Concave-Loss | 0.8455±0.0024 | 0.1219±0.0272 | 0.1050±0.0366 |
| Fair-DRO | 0.8211±0.0026 | 0.1795±0.0111 | 0.1044±0.0077 |
| Fair-Mixup | 0.8516±0.0018 | 0.1515±0.0153 | 0.0912±0.0399 |
| FAAP | 0.8524±0.0015 | 0.1695±0.0166 | 0.1416±0.0432 |
| FDR | 0.8253±0.0001 | 0.1830±0.0002 | 0.1677±0.0010 |
| TabFairGAN | 0.8516±0.0022 | 0.1752±0.0151 | 0.1674±0.0392 |
| Fair-CDA | 0.8515±0.0012 | 0.1786±0.0055 | 0.1711±0.0161 |
| Ours (ALFA) | 0.8244±0.0150 | 0.1012±0.0283 | 0.0660±0.0434 |

Table 6: Experimental Results for Adult dataset with ResNet-like network

| Adult | ResNet-like | | |
|---|---|---|---|
| | **Accuracy** | $\Delta DP$ | $\Delta EOd$ |
| Baseline | 0.8565±0.0012 | 0.1800±0.0083 | 0.1825±0.0341 |
| Covariance-Loss | 0.8557±0.0015 | 0.1834±0.0127 | 0.1916±0.0483 |
| Convex-Concave-Loss | 0.8305±0.0029 | 0.1064±0.0331 | 0.1155±0.0482 |
| Fair-DRO | 0.6627±0.0048 | 0.3154±0.0166 | 0.2991±0.0228 |
| Fair-Mixup | 0.8528±0.0015 | 0.1889±0.0121 | 0.1888±0.0399 |
| FAAP | 0.8558±0.0022 | 0.1724±0.0131 | 0.1631±0.0377 |
| FDR | 0.8472±0.0001 | 0.1930±0.0002 | 0.1765±0.0005 |
| TabFairGAN | 0.8544±0.0007 | 0.1699±0.0114 | 0.1317±0.0316 |
| Fair-CDA | 0.8529±0.0012 | 0.1921±0.0040 | 0.1714±0.0100 |
| Ours (ALFA) | 0.8262±0.0014 | 0.1956±0.0056 | 0.1304±0.0149 |

Table 7: Experimental Results for COMPAS dataset with Logistic Regression

| COMPAS | Logistic Regression | | |
|---|---|---|---|
| | **Accuracy** | $\Delta DP$ | $\Delta EOd$ |
| Baseline | 0.6578±0.0034 | 0.2732±0.0129 | 0.5319±0.0245 |
| Influence-Reweight | 0.6791±0.0000 | 0.2874±0.0000 | 0.5374±0.0000 |
| Covariance-Loss | 0.6565±0.0036 | 0.0243±0.0105 | 0.0671±0.0210 |
| Convex-Concave-Loss | 0.6462±0.0050 | 0.0194±0.0118 | 0.0825±0.0192 |
| Fair-DRO | 0.6427±0.0155 | 0.0989±0.0576 | 0.2106±0.1308 |
| Fair-Mixup | 0.6352±0.0109 | 0.0536±0.0345 | 0.0911±0.0576 |
| FAAP | 0.6577±0.0033 | 0.2670±0.0225 | 0.5184±0.0464 |
| FDR | 0.6677±0.0039 | 0.0377±0.0464 | 0.1001±0.0683 |
| TabFairGAN | 0.6403±0.0120 | 0.1506±0.0761 | 0.3005±0.1384 |
| Fair-CDA | 0.6288±0.0149 | 0.1629±0.0969 | 0.3265±0.1881 |
| Ours (ALFA) | 0.6694±0.0036 | 0.0193±0.0156 | 0.0876±0.0354 |

Table 8: Experimental Results for COMPAS dataset with MLP

| COMPAS | MLP | | |
|---|---|---|---|
| | Accuracy | $\Delta DP$ | $\Delta EOd$ |
| Baseline | 0.6711±0.0049 | 0.2059±0.0277 | 0.3699±0.0597 |
| LAFTR | 0.6397±0.0284 | 0.1164±0.0183 | 0.2089±0.0252 |
| Influence-Reweight | 0.6424±0.0000 | 0.1513±0.0000 | 0.2810±0.0000 |
| Covariance-Loss | 0.6729±0.0018 | 0.0820±0.0255 | 0.1121±0.0420 |
| Convex-Concave-Loss | 0.6631±0.0054 | 0.0641±0.0261 | 0.1026±0.0566 |
| Fair-DRO | 0.6782±0.0036 | 0.0457±0.0222 | 0.0690±0.0136 |
| Fair-Mixup | 0.6661±0.0033 | 0.0634±0.0331 | 0.0978±0.0537 |
| FAAP | 0.6741±0.0060 | 0.1976±0.0513 | 0.3495±0.1149 |
| FDR | 0.6559±0.0010 | 0.0771±0.0027 | 0.1104±0.0055 |
| TabFairGAN | 0.6767±0.0019 | 0.1616±0.0339 | 0.2726±0.0727 |
| Fair-CDA | 0.5886±0.0155 | 0.1088±0.0291 | 0.2131±0.0582 |
| Ours (ALFA) | 0.6702±0.0021 | 0.0204±0.0151 | 0.0410±0.0188 |

Table 9: Experimental Results for COMPAS dataset with ResNet-like network

| COMPAS | ResNet-like | | |
|---|---|---|---|
| | Accuracy | $\Delta DP$ | $\Delta EOd$ |
| Baseline | 0.6753±0.0037 | 0.2055±0.0307 | 0.3683±0.0700 |
| Covariance-Loss | 0.6793±0.0034 | 0.0957±0.0342 | 0.1349±0.0599 |
| Convex-Concave-Loss | 0.6671±0.0060 | 0.0856±0.0707 | 0.1795±0.1099 |
| Fair-DRO | 0.6699±0.0043 | 0.0376±0.0262 | 0.0546±0.0236 |
| Fair-Mixup | 0.6729±0.0049 | 0.2069±0.0355 | 0.3752±0.0774 |
| FAAP | 0.6718±0.0038 | 0.2020±0.0256 | 0.3626±0.056 |
| FDR | 0.6725±0.0006 | 0.1288±0.0018 | 0.1984±0.0034 |
| TabFairGAN | 0.6769±0.0022 | 0.1751±0.0421 | 0.3046±0.0872 |
| Fair-CDA | 0.6701±0.0016 | 0.1850±0.0139 | 0.3244±0.0306 |
| Ours (ALFA) | 0.6756±0.0032 | 0.0124±0.0137 | 0.0659±0.0316 |

Table 10: Experimental Results for German dataset with Logistic Regression

| German | Logistic Regression | | |
|---|---|---|---|
| | Accuracy | $\Delta DP$ | $\Delta EOd$ |
| Baseline | 0.7220±0.0131 | 0.1186±0.0642 | 0.3382±0.1268 |
| Influence-Reweight | 0.6650±0.0000 | 0.0665±0.0000 | 0.2475±0.0000 |
| Covariance-Loss | 0.7410±0.0218 | 0.0758±0.0459 | 0.1247±0.0792 |
| Convex-Concave-Loss | 0.7625±0.0144 | 0.0590±0.0373 | 0.0979±0.0458 |
| Fair-DRO | 0.6805±0.0223 | 0.0627±0.0401 | 0.1419±0.0610 |
| Fair-Mixup | 0.7385±0.0103 | 0.0778±0.0174 | 0.1694±0.0533 |
| FAAP | 0.7295±0.0250 | 0.1128±0.0810 | 0.3083±0.1518 |
| FDR | 0.7640±0.0049 | 0.0398±0.0178 | 0.2382±0.0523 |
| TabFairGAN | 0.7460±0.0189 | 0.0677±0.0341 | 0.2762±0.0587 |
| Fair-CDA | 0.7115±0.0219 | 0.0662±0.0302 | 0.2191±0.1178 |
| Ours (ALFA) | 0.7940±0.0058 | 0.0470±0.0199 | 0.0469±0.0276 |

Table 11: Experimental Results for German dataset with MLP

| German | MLP | | |
|---|---|---|---|
| | Accuracy | $\Delta DP$ | $\Delta EOd$ |
| Baseline | 0.7800±0.0150 | 0.0454±0.0282 | 0.2096±0.0924 |
| LAFTR | 0.7308±0.0270 | 0.0419±0.0410 | 0.1677±0.1433 |
| Influence-Reweight | 0.8000±0.0000 | 0.0297±0.0000 | 0.1347±0.0000 |
| Covariance-Loss | 0.7800±0.0116 | 0.0588±0.0185 | 0.1175±0.0375 |
| Convex-Concave-Loss | 0.7570±0.0150 | 0.0318±0.0217 | 0.0793±0.0459 |
| Fair-DRO | 0.7405±0.0096 | 0.0661±0.0222 | 0.1190±0.0705 |
| Fair-Mixup | 0.7545±0.0154 | 0.0641±0.0235 | 0.1378±0.0778 |
| FAAP | 0.7785±0.0148 | 0.0191±0.0221 | 0.1788±0.0638 |
| FDR | 0.7650±0.0000 | 0.0284±0.0000 | 0.2036±0.0000 |
| TabFairGAN | 0.7615±0.0123 | 0.0361±0.0227 | 0.1799±0.0889 |
| Fair-CDA | 0.7325±0.0288 | 0.0408±0.0253 | 0.1651±0.0976 |
| Ours (ALFA) | 0.7570±0.0024 | 0.0053±0.0064 | 0.0813±0.0110 |

Table 12: Experimental Results for German dataset with ResNet-like network

| German | ResNet-like | | |
|---|---|---|---|
| | Accuracy | $\Delta DP$ | $\Delta EOd$ |
| Baseline | 0.7595±0.0224 | 0.0524±0.0368 | 0.2193±0.1224 |
| Covariance-Loss | 0.7605±0.0149 | 0.0360±0.0263 | 0.1086±0.0479 |
| Convex-Concave-Loss | 0.7500±0.0210 | 0.0493±0.0271 | 0.1076±0.0601 |
| Fair-DRO | 0.7475±0.0214 | 0.0629±0.0248 | 0.1015±0.0432 |
| Fair-Mixup | 0.7280±0.0189 | 0.0693±0.0367 | 0.1907±0.0719 |
| FAAP | 0.7330±0.0147 | 0.0459±0.0438 | 0.1883±0.1198 |
| FDR | 0.7190±0.0020 | 0.1224±0.0027 | 0.3487±0.0080 |
| TabFairGAN | 0.7535±0.0256 | 0.0407±0.0247 | 0.2278±0.0527 |
| Fair-CDA | 0.7395±0.0119 | 0.0457±0.0288 | 0.1858±0.0958 |
| Ours (ALFA) | 0.7325±0.0186 | 0.0309±0.0147 | 0.0665±0.0228 |

Table 13: Experimental Results for Drug dataset with Logistic Regression

| Drug | Logistic Regression | | |
|---|---|---|---|
| | Accuracy | $\Delta DP$ | $\Delta EOd$ |
| Baseline | 0.6626±0.0135 | 0.2938±0.0761 | 0.5064±0.1616 |
| Influence-Reweight | 0.6446±0.0000 | 0.1245±0.0000 | 0.1694±0.0000 |
| Covariance-Loss | 0.6491±0.0078 | 0.0736±0.0436 | 0.2060±0.0549 |
| Convex-Concave-Loss | 0.6225±0.0138 | 0.0781±0.0316 | 0.1429±0.0317 |
| Fair-DRO | 0.6403±0.0091 | 0.0710±0.0441 | 0.1789±0.0515 |
| Fair-Mixup | 0.6533±0.0077 | 0.0979±0.0482 | 0.1787±0.0793 |
| FAAP | 0.6729±0.0117 | 0.3220±0.0486 | 0.5576±0.0998 |
| FDR | 0.6599±0.0011 | 0.3008±0.0128 | 0.5397±0.0268 |
| TabFairGAN | 0.6650±0.0113 | 0.2796±0.0527 | 0.4668±0.1101 |
| Fair-CDA | 0.6615±0.0021 | 0.3085±0.0078 | 0.5417±0.0156 |
| Ours (ALFA) | 0.6554±0.0067 | 0.0909±0.0261 | 0.1170±0.0255 |

Table 14: Experimental Results for Drug dataset with MLP

| Drug | MLP | | |
|---|---|---|---|
| | Accuracy | $\Delta DP$ | $\Delta EOd$ |
| Baseline | 0.6674±0.0096 | 0.2760±0.0415 | 0.4718±0.0838 |
| LAFTR | 0.6195±0.0352 | 0.1848±0.1035 | 0.3235±0.1715 |
| Influence-Reweight | 0.6525±0.0000 | 0.1610±0.0000 | 0.2362±0.0000 |
| Covariance-Loss | 0.6488±0.0099 | 0.0695±0.0361 | 0.1410±0.0354 |
| Convex-Concave-Loss | 0.6467±0.0074 | 0.0529±0.0294 | 0.1040±0.0251 |
| Fair-DRO | 0.6528±0.0112 | 0.0841±0.0158 | 0.1198±0.0216 |
| Fair-Mixup | 0.6499±0.0126 | 0.0723±0.0318 | 0.1168±0.0359 |
| FAAP | 0.6732±0.0095 | 0.2792±0.0410 | 0.4707±0.0860 |
| FDR | 0.6366±0.0000 | 0.1296±0.0000 | 0.1956±0.0000 |
| TabFairGAN | 0.6828±0.0122 | 0.2132±0.0245 | 0.3258±0.0421 |
| Fair-CDA | 0.5263±0.0479 | 0.1250±0.0950 | 0.2608±0.1844 |
| Ours (ALFA) | 0.6350±0.0082 | 0.0511±0.0356 | 0.0640±0.0528 |

Table 15: Experimental Results for Drug dataset with ResNet-like network

| Drug | ResNet-like | | |
|---|---|---|---|
| | Accuracy | $\Delta DP$ | $\Delta EOd$ |
| Baseline | 0.6541±0.0150 | 0.2556±0.0316 | 0.4366±0.0638 |
| Covariance-Loss | 0.6467±0.0189 | 0.0817±0.0474 | 0.1233±0.0847 |
| Convex-Concave-Loss | 0.6491±0.0124 | 0.0733±0.0584 | 0.1258±0.0808 |
| Fair-DRO | 0.6281±0.0074 | 0.0829±0.0228 | 0.1460±0.0267 |
| Fair-Mixup | 0.6363±0.0063 | 0.1906±0.0423 | 0.1785±0.0634 |
| FAAP | 0.6637±0.0124 | 0.2186±0.0397 | 0.3577±0.0792 |
| FDR | 0.6313±0.0000 | 0.2146±0.0000 | 0.3625±0.0000 |
| TabFairGAN | 0.6759±0.0120 | 0.2143±0.0401 | 0.3358±0.0646 |
| Fair-CDA | 0.6655±0.0106 | 0.2311±0.0261 | 0.3749±0.0479 |
| Ours (ALFA) | 0.6557±0.0132 | 0.0763±0.0419 | 0.0966±0.0553 |

## H.2 EXPERIMENTAL RESULTS OF IMAGE DATASET

Table 16: Experimental results for CelebA dataset with various backbones. The best results are marked in Blue, and the second best results are marked by Cyan.

| CelebA | ResNet-50 | | | ViT | | | Swin-Transformer | | |
|---|---|---|---|---|---|---|---|---|---|
| Attractive | Accuracy | $\Delta DP$ | $\Delta EOd$ | Accuracy | $\Delta DP$ | $\Delta EOd$ | Accuracy | $\Delta DP$ | $\Delta EOd$ |
| Baseline | 0.8196 | 0.4374 | 0.4457 | 0.8208 | 0.4675 | 0.4934 | 0.8301 | 0.4692 | 0.4817 |
| FairMixup | 0.8032 | 0.4035 | 0.4188 | 0.8192 | 0.4536 | 0.4822 | 0.8261 | 0.4526 | 0.4804 |
| FAAP | 0.8098 | 0.4146 | 0.4218 | 0.8210 | 0.4633 | 0.4865 | 0.8287 | 0.4616 | 0.4852 |
| FDR | 0.8142 | 0.4269 | 0.4382 | 0.8071 | 0.3983 | 0.3867 | 0.8249 | 0.4452 | 0.4666 |
| ALFA (Ours) | 0.8092 | 0.4137 | 0.4209 | 0.8051 | 0.3956 | 0.3777 | 0.8005 | 0.3796 | 0.3698 |
| CelebA | ResNet-50 | | | ViT | | | Swin-Transformer | | |
| Wavy Hair | Accuracy | $\Delta DP$ | $\Delta EOd$ | Accuracy | $\Delta DP$ | $\Delta EOd$ | Accuracy | $\Delta DP$ | $\Delta EOd$ |
| Baseline | 0.8339 | 0.3141 | 0.4112 | 0.8532 | 0.391 | 0.5579 | 0.8483 | 0.3391 | 0.4877 |
| FairMixup | 0.8052 | 0.2486 | 0.3432 | 0.8502 | 0.3521 | 0.4683 | 0.8581 | 0.3712 | 0.4935 |
| FAAP | 0.8352 | 0.3170 | 0.4126 | 0.8528 | 0.3931 | 0.5623 | 0.8478 | 0.3376 | 0.4864 |
| FDR | 0.8432 | 0.3353 | 0.3981 | 0.8490 | 0.3367 | 0.2969 | 0.8607 | 0.3779 | 0.4638 |
| ALFA (Ours) | 0.8391 | 0.3457 | 0.3419 | 0.8449 | 0.3425 | 0.2809 | 0.8600 | 0.3750 | 0.4022 |
| CelebA | ResNet-50 | | | ViT | | | Swin-Transformer | | |
| Smile | Accuracy | $\Delta DP$ | $\Delta EOd$ | Accuracy | $\Delta DP$ | $\Delta EOd$ | Accuracy | $\Delta DP$ | $\Delta EOd$ |
| Baseline | 0.9275 | 0.1566 | 0.0403 | 0.9239 | 0.1642 | 0.0587 | 0.9340 | 0.1629 | 0.0491 |
| FairMixup | 0.9281 | 0.1566 | 0.0384 | 0.9219 | 0.1592 | 0.0482 | 0.9338 | 0.1609 | 0.0453 |
| FAAP | 0.9273 | 0.1563 | 0.0400 | 0.9243 | 0.1647 | 0.0602 | 0.9326 | 0.1593 | 0.0427 |
| FDR | 0.9276 | 0.1558 | 0.0395 | 0.9234 | 0.1446 | 0.0361 | 0.9286 | 0.1399 | 0.0380 |
| ALFA (Ours) | 0.9284 | 0.1509 | 0.0271 | 0.9219 | 0.1552 | 0.0402 | 0.9269 | 0.1234 | 0.0344 |

## H.3 EXPERIMENTAL RESULTS OF TEXT DATASET

Table 17: Experimental Results for Wiki dataset with LSTM

| Wiki | LSTM | | |
|---|---|---|---|
| | Accuracy | $\Delta DP$ | $\Delta EOd$ |
| Base | 0.9384±0.0006 | 0.1764±0.0046 | 0.0779±0.0089 |
| Fair-Mixup | 0.8737±0.0290 | 0.1831±0.0618 | 0.0701±0.0347 |
| FAAP | 0.9395±0.0002 | 0.1801±0.0016 | 0.0807±0.0063 |
| FDR | 0.9256±0.0014 | 0.2110±0.0022 | 0.0715±0.0040 |
| Ours (ALFA) | 0.9360±0.0000 | 0.1974±0.0011 | 0.0548±0.0029 |

Table 18: Experimental Results for Wiki dataset with BERT

| Wiki | BERT | | |
|---|---|---|---|
| | Accuracy | $\Delta DP$ | $\Delta EOd$ |
| Base | 0.9384±0.0003 | 0.2042±0.0041 | 0.0742±0.0068 |
| Fair-Mixup | 0.9106±0.0041 | 0.1858±0.0164 | 0.0537±0.0249 |
| FAAP | 0.9512±0.0018 | 0.2128±0.0177 | 0.0639±0.0145 |
| FDR | 0.9142±0.0029 | 0.2427±0.0089 | 0.0680±0.0166 |
| Ours (ALFA) | 0.9145±0.0006 | 0.2214±0.0019 | 0.0321±0.0035 |

Table 19: Experimental Results for Wiki dataset with DistillBERT

| Wiki | DistillBERT | | |
|---|---|---|---|
| | **Accuracy** | $\Delta DP$ | $\Delta EOd$ |
| Base | 0.9562±0.0003 | 0.2103±0.0040 | 0.0856±0.0058 |
| Fair-Mixup | 0.9332±0.0067 | 0.1561±0.0131 | 0.0588±0.0273 |
| FAAP | 0.9547±0.0046 | 0.2240±0.0240 | 0.0977±0.0222 |
| FDR | 0.9113±0.0026 | 0.1922±0.0111 | 0.0592±0.0163 |
| Ours (ALFA) | 0.9242±0.0003 | 0.2278±0.0005 | 0.0397±0.0015 |

## I  DATASET DETAILS

We follow the existing data pre-processing, (Mroueh et al., 2021) for the Adult and CelebA dataset, and (Mehrabi et al., 2021) for other datasets.

Table 20: Features used from the Adult, COMPAS, German Credit, and Drug Consumption datasets.

**Adult**

| | | | |
|---|---|---|---|
| age | workclass | education-num | marital-status |
| occupation | relationship | race | sex |
| capital-gain | capital-loss | hours-per-week | |

**COMPAS**

| | | | |
|---|---|---|---|
| sex | age_cat | race | juv_fel_count |
| juv_misd_count | juv_other_count | priors_count | c_charge_degree |

**German**

| | | | |
|---|---|---|---|
| Checking Account | Duration | Credit history | Purpose |
| Credit amount | Savings account | Employment | Installment rate |
| Gender | Debtors/guarantors | Residence | Property |
| Age | Installment plans | Housing | Existing credits |
| Job | Liability | Telephone | Foreigner credits |

**Drug**

| | | | |
|---|---|---|---|
| Age | Gender | Education | Country |
| Ethnicity | Nscore | Escore | Oscore |
| Ascore | Cscore | Impulsive | SS |

**UCI Adult Dataset.** Adult dataset (Dua et al., 2017) contains 48,842 individuals' information about income obtained from the 1994 US Census database. The target label is binarized to determine whether the income exceeds $50K/yr. Similar to (Mroueh et al., 2021) and (Yurochkin et al., 2019), samples including missing values are dropped so that the number of available samples is 45,222. The sex feature is used as a sensitive attribute.

**COMPAS Dataset.** COMPAS dataset (Jeff Larson & Angwin, 2016) contains 7,214 samples about criminal defendants and risk of recidivism with 8 attributes. It aims to classify whether a person commits a crime in the two years after they were scored. The sex feature is used as a sensitive attribute.

**German Credit Dataset.** German dataset (Dua et al., 2017) contains the credit profiles for 1,000 individuals with 20 attributes such as accounts, income, properties, and gender. The prediction goal is to classify whether a person has good or bad credit risks. The gender feature is used as a sensitive attribute.

**Drug Consumption Dataset.** Drug Consumption dataset (Dua et al., 2017) contains records from 1,885 respondents about drug consumption. Each data point has 12 attributes including the level of

education, age, gender, and so on. The original task is multi-classification for 7 classes of whether and when respondents experienced drugs, but our prediction goal is abridged whether they consumed cocaine or not. The gender feature is used as a sensitive attribute.

**CelebA Dataset.** CelebA dataset (Liu et al., 2018) contains more than 200,000 celebrity face images, each coupled with 40 human-annotated binary characteristics such as gender. From these characteristics, we specifically choose attractive, smile and wavy hair, utilizing them to establish three binary classification assignments, with gender regarded as the sensitive attribute following (Zhang et al., 2017). We select these particular attributes as, in every task, a sensitive group is present which has a higher number of positive samples compared to the other.

**Wikipedia Talk Toxicity Dataset.** Moreover, we further explore the adaptability of the proposed method to the Natural Language Processing (NLP) dataset. We utilize Wikipedia Talk Toxicity Prediction (Thain et al., 2017) which is a comprehensive collection aimed at identifying toxic content within discussion comments posted on Wikipedia's talk pages, produced by the Conversation AI project. In this context, toxicity is defined as content that may be perceived as "rude, disrespectful, or unreasonable." It consists of over 100,000 comments from the English Wikipedia, each meticulously annotated by crowd workers, as delineated in their associated research paper. A challenge presented by this dataset is the underrepresentation of comments addressing sensitive subjects such as sexuality, religion, gender identity, and race. In this paper, the existence of sexuality terms such as 'gay', 'lesbian', 'bisexual', 'homosexual', 'straight', and 'heterosexual' is used as the sensitive attribute, 1 for existing, and 0 for absence.

# J  ADDITIONAL EXPERIMENTS

## J.1  MULTI-LABEL CLASSIFICATION SCENARIO

We clarify that ALFA can be applied to the multi-label classification with binary-protected features as it can be seen in multiple binary classification scenarios having individual decision boundaries. In this case, the fairness loss is newly defined as covariance between a sensitive attribute and the mean of the signed distances, $L_{fair} = Cov(a, \frac{1}{T}\sum_{t=1}^{T} g_t(z_t + \delta_t))$ where $T$ is the number of targeted prediction.

Luckily, one of our datasets, the Drug Consumption dataset (Dua et al., 2017) has multiple labels. To further investigate the feasibility of our framework for the multi-label classification, we conduct additional experiments on the Drug Consumption dataset choosing four prediction goals, Cocaine, Benzodiazepine, Ketamine, and Magic Mushrooms while only Cocaine is considered as a prediction goal in the manuscript. The experimental result shows that ALFA effectively mitigates biases in the multi-label classification.

Table 21: Experimental results for multi-label classification

| Accuracy | Cocaine | Benzos | Ketamine | Mushrooms |
|---|---|---|---|---|
| Logistic Regression | $0.7057 \pm 0.0099$ | $0.6689 \pm 0.0113$ | $0.6989 \pm 0.0267$ | $0.7223 \pm 0.0094$ |
| Logistic Regression + ALFA | $0.6816 \pm 0.0114$ | $0.6643 \pm 0.0122$ | $0.7505 \pm 0.0023$ | $0.7307 \pm 0.0082$ |
| MLP | $0.6802 \pm 0.0144$ | $0.6527 \pm 0.0138$ | $0.7551 \pm 0.0094$ | $0.7053 \pm 0.0114$ |
| MLP + ALFA | $0.6701 \pm 0.0057$ | $0.6138 \pm 0.0036$ | $0.7343 \pm 0.0031$ | $0.6587 \pm 0.0057$ |
| $\Delta DP$ | **Cocaine** | **Benzos** | **Ketamine** | **Mushrooms** |
| Logistic Regression | $0.2691 \pm 0.0232$ | $0.3597 \pm 0.0298$ | $0.2478 \pm 0.1140$ | $0.4151 \pm 0.0372$ |
| Logistic Regression + ALFA | $\mathbf{0.0986 \pm 0.0289}$ | $\mathbf{0.2666 \pm 0.0424}$ | $\mathbf{0.0248 \pm 0.0070}$ | $\mathbf{0.3993 \pm 0.0425}$ |
| MLP | $0.2183 \pm 0.0222$ | $0.3179 \pm 0.0278$ | $0.0903 \pm 0.1320$ | $0.4072 \pm 0.0206$ |
| MLP + ALFA | $\mathbf{0.0760 \pm 0.0114}$ | $\mathbf{0.1808 \pm 0.0137}$ | $\mathbf{0.0368 \pm 0.0103}$ | $\mathbf{0.2384 \pm 0.0099}$ |
| $\Delta EOd$ | **Cocaine** | **Benzos** | **Ketamine** | **Mushrooms** |
| Logistic Regression | $0.4411 \pm 0.0483$ | $0.6448 \pm 0.0635$ | $0.5184 \pm 0.2320$ | $0.7096 \pm 0.0732$ |
| Logistic Regression + ALFA | $\mathbf{0.1234 \pm 0.0471}$ | $\mathbf{0.4498 \pm 0.0858}$ | $\mathbf{0.0689 \pm 0.0158}$ | $\mathbf{0.6621 \pm 0.0911}$ |
| MLP | $0.3505 \pm 0.0449$ | $0.5601 \pm 0.0597$ | $0.2492 \pm 0.0385$ | $0.6912 \pm 0.0441$ |
| MLP + ALFA | $\mathbf{0.0963 \pm 0.0249}$ | $\mathbf{0.2971 \pm 0.0193}$ | $\mathbf{0.1215 \pm 0.0153}$ | $\mathbf{0.3628 \pm 0.0185}$ |

## J.2 MULTIPLE SENSITIVE ATTRIBUTE SCENARIO

In the binary classification with multi-protected features, the Differential Fairness (DF) is measured by binarization of each multi-protected features. For example, Foulds et al. (2020) defined DF

$$DF = \max_{i \in S} \max_{j \in S \setminus \{i\}} \left( \left| \log \frac{P(y = 1 \mid a = i)}{P(y = 1 \mid a = j)} \right| \right)$$

where $i, j \in S$, and $S$ denotes the set of multiple sensitive attributes. Therefore, in the multi-protected feature case, we can define 'unfair region' by finding a particular sensitive attribute provoking the maximum mistreatment and reducing the misclassification rate of the unfair region as well as the binary sensitive attribute case.

For the multiple sensitive attribute setting, we adopt COMPAS dataset and MEPS dataset. MEPS (Bellamy et al., 2018) data consists of 34,655 instances with 41 features(e.g. demographic information, health services records, costs, etc.) Among all the features, only 42 features are used. The sum of total medicare visiting is used as a binary target label. When the total number of visiting is greater or equal to 10, a patient is labeled as 1, otherwise 0. And 'race' is used as multiple sensitive attributes, 0 for White, 1 for Black, and 2 for others. The experimental result shows that ALFA is also applicable to the multiple sensitive attributes scenario.

Table 22: Experimental results for multiple sensitive attributes fairness

| COMPAS | Accuracy | DF |
|---|---|---|
| MLP | 0.6875±0.0048 | 1.7500±0.5794 |
| MLP + ALFA | **0.6895±0.0023** | **1.3960±0.0892** |
| **MEPS** | **Accuracy** | **DF** |
| MLP | 0.6208±0.0137 | 0.2900±0.0700 |
| MLP + ALFA | **0.6860±0.0024** | **0.1985±0.0226** |

## J.3 MULTI-CLASS CLASSIFICATION SCENARIO

For the multi-class classification, the decision boundaries are not linear, so our framework might not be directly applicable. However, multi-class classification can indeed be conceptualized as multiple binary classifications in a certain strategy called One-Vs-All. In this approach, for a problem with $N$ classes, we can create $N$ different binary classifiers. Each classifier is trained to distinguish between one of the classes and all other classes combined.

As each classifier can be seen as a binary classification task, we can utilize ALFA for the multi-class classification scenario by detecting unfair regions and covering the region by fairness attack. The evaluation metric for multi-class fairness takes maximum Demographic Parity across the classes (Denis et al., 2021). In details,

$$\Delta DP_{\text{multi}} = \max_{k \in [K]} \left| P(\hat{Y} = k | a = 1) - P(\hat{Y} = k | a = 0) \right|$$

where $\hat{Y}$ is the predicted class, and $k \in [K]$ denotes each class $k$ in the multi-class classification.

Among existing datasets for fairness research, Drug dataset can be used for multi-class classification. In fact, the original labels of the Drug dataset are multi-class settings, from 'CL0' to 'CL6' indicating the frequency of drug abuse. We have binarized them as 'never used' and 'ever used' regardless of the frequency in the main paper. However, for the multi-class classification setting, we adopt the original multi-class setting and report the mean accuracy and $\Delta DP_{\text{multi}}$ with MLP.

Table 23: Experimental results for multi-class classification

| Drug Multi-class | Accuracy | $\Delta DP_{\text{multi}}$ |
|---|---|---|
| MLP | 0.5196±0.0032 | 0.1930±0.0132 |
| MLP + ALFA | 0.4960±0.0219 | **0.1733±0.0287** |

## K  INTERPRETABILITY OF THE AUGMENTED FEATURE AND INPUT PERTURBATION

In this work, we can consider interpretability from two aspects: interpretability on decision boundary (latent space), and interpretability on original feature (input space). While we have focused on the first aspect, we argue that the proposed method can cover the second aspect as well.

At first, we are focusing on the interpretability of decision boundaries, which is a common approach to understand the classifier's behavior (Guidotti et al., 2020; Bodria et al., 2022). By manipulating features in the latent space by the fairness attack, we can interpret the decision boundary by discovering an unfair region and adjusting the decision boundary. In this case, it is true that it can't analyze how the changes in input features affect the decision boundary. On the other hand, the interpretability of the input feature might make it possible to analyze how the fairness attack perturbs input data. However, it may lose the interpretability of decision boundary, such as discovering unfair regions and understanding the last layer's behavior.

Fortunately, our framework is applicable to the input space by deploying the fairness attack and perturbation in the input space. In this case, the entire model will be fine-tuned, while offering input-level interpretability. We conducted additional experiments with MLP to show the validity of our framework on the input space in Table 24. Consequently, our method can offer either interpretability on latent space or input space. In both cases, we can maintain the accuracy level while mitigating the fairness issue. We opt to freeze the pretrained encoder and deploy perturbations in the latent space, as this approach generally leads to greater improvements in fairness compared to perturbation in input space in various datasets.

Table 24: Experimental results for input and latent perturbation with MLP.

| Adult | Accuracy | $\Delta DP$ | $\Delta EOd$ |
|---|---|---|---|
| MLP | $0.8525 \pm 0.0010$ | $0.1824 \pm 0.0114$ | $0.1768 \pm 0.0411$ |
| MLP + Latent Perturb. | $0.8380 \pm 0.0045$ | $0.1642 \pm 0.0261$ | $0.0971 \pm 0.0098$ |
| MLP + Input Perturb. | $0.8473 \pm 0.0016$ | $0.1588 \pm 0.0135$ | $0.1016 \pm 0.0394$ |
| **COMPAS** | **Accuracy** | $\Delta DP$ | $\Delta EOd$ |
| MLP | $0.6711 \pm 0.0049$ | $0.2059 \pm 0.0277$ | $0.3699 \pm 0.0597$ |
| MLP + Latent Perturb. | $0.6701 \pm 0.0020$ | $0.0207 \pm 0.0142$ | $0.0793 \pm 0.0418$ |
| MLP + Input Perturb. | $0.6629 \pm 0.0051$ | $0.0610 \pm 0.0389$ | $0.1086 \pm 0.0649$ |
| **German** | **Accuracy** | $\Delta DP$ | $\Delta EOd$ |
| MLP | $0.7800 \pm 0.0150$ | $0.0454 \pm 0.0282$ | $0.2096 \pm 0.0924$ |
| MLP + Latent Perturb. | $0.7570 \pm 0.0024$ | $0.0053 \pm 0.0064$ | $0.0813 \pm 0.0110$ |
| MLP + Input Perturb. | $0.7465 \pm 0.0067$ | $0.0188 \pm 0.0106$ | $0.1700 \pm 0.0400$ |
| **Drug** | **Accuracy** | $\Delta DP$ | $\Delta EOd$ |
| MLP | $0.6674 \pm 0.0096$ | $0.2760 \pm 0.0415$ | $0.4718 \pm 0.0838$ |
| MLP + Latent Perturb. | $0.6382 \pm 0.0061$ | $0.0820 \pm 0.0259$ | $0.1068 \pm 0.0476$ |
| MLP + Input Perturb. | $0.6188 \pm 0.0146$ | $0.0571 \pm 0.0365$ | $0.1893 \pm 0.0809$ |

## L    ANOTHER FAIRNESS CONSTRAINT

In this part, we show that ALFA can adopt any types of fairness constraint during the fairenss attack. As an alternative of (Zafar et al., 2017), we present (Wu et al., 2019) below.

Let's say $f(\mathbf{X})$ is a logit of binary classifier given data $\mathbf{X}$ and define indicator functions $\mathbb{1}(\cdot)$ where $\cdot$ denotes each condition for the indicator function.

The empirical DP Gap is

$$\Delta DP(f) = \frac{1}{|\mathbb{1}(a=1)|} \sum_{a=1} \mathbb{1}(f(\mathbf{X}) > 0) - \frac{1}{|\mathbb{1}(a=0)|} \sum_{a=0} \mathbb{1}(f(\mathbf{X}) > 0).$$

and can be rewritten in the expected form as

$$\Delta DP(f) = \mathbb{E}\Big[\frac{\mathbb{1}(a=1)}{p_1} \mathbb{1}(f(\mathbf{X}) > 0) - (1 - \frac{\mathbb{1}(a=0)}{1-p_1} \mathbb{1}(f(\mathbf{X}) < 0))\Big]$$

where $p_1 = p(a=1)$.

Moreover, the relaxed form replacing the indicator function to real-valued function is written as

$$\Delta DP(f) = \mathbb{E}\Big[\frac{\mathbb{1}(a=1)}{p_1} f(\mathbf{X}) - (1 - \frac{\mathbb{1}(a=0)}{1-p_1} f(\mathbf{X}))\Big].$$

In (Wu et al., 2019), $f(\mathbf{X})$ is replaced again to construct a convex form using two different surrogate functions to use $\Delta DP$ as a fairness constraint,

$$\Delta DP\kappa(f) = \mathbb{E}\Big[\frac{\mathbb{1}(a=1)}{p_1} \kappa(f(\mathbf{X})) - \big(1 - \frac{\mathbb{1}(a=0)}{1-p_1} \kappa(-f(\mathbf{X}))\big)\Big]$$

$$\Delta DP\delta(f) = \mathbb{E}\Big[\frac{\mathbb{1}(a=1)}{p_1} \delta(f(\mathbf{X})) - \big(1 - \frac{\mathbb{1}(a=0)}{1-p_1} \delta(-f(\mathbf{X}))\big)\Big]$$

where $\kappa$ is a convex surrogate function $\kappa(z) = \max(z+1, 0)$ and $\delta$ is a concave surrogate function $\delta(z) = \min(z, 1)$ as proposed in (Wu et al., 2019). If $\Delta DP(f) \geq 0$, we directly use $\Delta DP\kappa(f)$ as a fairness constraint, otherwise use $\Delta DP\delta(f)$,

$$L_{fair} = \begin{cases} \Delta DP\kappa(f) & \text{if } \Delta DP \geq 0 \\ \\ \Delta DP\delta(f) & \text{if } \Delta DP < 0. \end{cases}$$

Also, it can be extended to use $\Delta EOD$ directly as a fairness constraint, by conditioning $\Delta DP$ for each $y \in \{0,1\}$.

$$\Delta EOD = \Big[\frac{1}{|\mathbb{1}(a=1, y=1)|} \sum_{a=1, y=1} \mathbb{1}(f(x) > 0) - \frac{1}{|\mathbb{1}(a=0, y=1)|} \sum_{a=0, y=1} \mathbb{1}(f(x) > 0)\Big]$$

$$+\Big[\frac{1}{|\mathbb{1}(a=1, y=0)|} \sum_{a=1, y=0} \mathbb{1}(f(x) > 0) - \frac{1}{|\mathbb{1}(a=0, y=0)|} \sum_{a=0, y=0} \mathbb{1}(f(x) > 0)\Big],$$

and can be rewritten in the expected form as

$$\Delta EOD(f) = \mathbb{E}\Big[\frac{\mathbb{1}(a=1, y=1)}{p_{1,1}} \mathbb{1}(f(\mathbf{X}) > 0) - \big(1 - \frac{\mathbb{1}(a=0, y=1)}{\pi - p_{1,1}} \mathbb{1}(f(\mathbf{X}) < 0)\big)\Big]$$

$$+\mathbb{E}\Big[\frac{\mathbb{1}(a=1, y=0)}{p_{1,0}} \mathbb{1}(f(\mathbf{X}) > 0) - \big(1 - \frac{\mathbb{1}(a=0, y=0)}{1 - \pi - p_{1,0}} \mathbb{1}(f(\mathbf{X}) < 0)\big)\Big]$$

since $1 = \mathbb{E}[\frac{\mathbb{1}(a=0, y=1)}{p_{0,1}}] = \mathbb{E}[\frac{\mathbb{1}(a=0, y=1)}{\pi - p_{1,1}}] = \mathbb{E}[\frac{\mathbb{1}(a=0, y=1)}{\pi - p_{1,1}} \mathbb{1}(f(\mathbf{X}) < 0) + \frac{\mathbb{1}(a=0, y=1)}{\pi - p_{1,1}} \mathbb{1}(f(\mathbf{X}) > 0)]$ and $1 = \mathbb{E}[\frac{\mathbb{1}(a=0, y=0)}{p_{0,0}}] = \mathbb{E}[\frac{\mathbb{1}(a=0, y=0)}{1 - \pi - p_{1,0}}] = \mathbb{E}[\frac{\mathbb{1}(a=0, y=0)}{1 - \pi - p_{1,0}} \mathbb{1}(f(\mathbf{X}) < 0) + \frac{\mathbb{1}(a=0, y=0)}{1 - \pi - p_{1,0}} \mathbb{1}(f(\mathbf{X}) > 0)]$

0)], $\pi = p(y = 1)$ and $p(y = 0) = 1 - \pi$ where $p_{1,1} = P(a = 1, y = 1)$ and $p_{1,0} = P(a = 1, y = 0)$. $\Delta EOD$ can be expressed as a convex form,

$$\Delta EOD\kappa(f) = \mathbb{E}\Big[\frac{\mathbb{1}(a = 1, y = 1)}{p_{1,1}}\kappa(f(\mathbf{X})) - \Big(1 - \frac{\mathbb{1}(a = 0, y = 1)}{\pi - p_{1,1}}\kappa(-f(\mathbf{X}))\Big)\Big]$$

$$+\mathbb{E}\Big[\frac{\mathbb{1}(a = 1, y = 0)}{p_{1,0}}\kappa(f(\mathbf{X})) - \Big(1 - \frac{\mathbb{1}(a = 0, y = 0)}{1 - \pi - p_{1,0}}\kappa(-f(\mathbf{X}))\Big)\Big]$$

$$\Delta EOD\delta(f) = \mathbb{E}\Big[\frac{\mathbb{1}(a = 1, y = 1)}{p_{1,1}}\delta(f(\mathbf{X})) - \Big(1 - \frac{\mathbb{1}(a = 0, y = 1)}{\pi - p_{1,1}}\delta(-f(\mathbf{X}))\Big)\Big]$$

$$+\mathbb{E}\Big[\frac{\mathbb{1}(a = 1, y = 0)}{p_{1,0}}\delta(f(\mathbf{X})) - \Big(1 - \frac{\mathbb{1}(a = 0, y = 0)}{1 - \pi - p_{1,0}}\delta(-f(\mathbf{X}))\Big)\Big].$$

where

$$L_{fair} = \begin{cases} \Delta EOD\kappa(f) & \text{if } \Delta EOD \geq 0 \\ \\ \Delta EOD\delta(f) & \text{if } \Delta EOD < 0. \end{cases}$$

Therefore, different from the covariance (Zafar et al., 2017) between prediction and sensitive attribute, the convex fairness constraint takes into account the empirical outputs considering all potential dependencies, not focusing on a particular attribute.

We report the experimental results in the table below by comparing the baseline, the covariance-base fairness attack (suggested in the paper), and the convex fairness attack. The experiment shows that our method can adopt any type of fairness constraint during the attacking step, both showing improvement in fairness.

While our framework has wide adaptability in the choice of fairness constraint during the fairness attack, the reason we chose covariance instead of convex fairness constraint is it doesn't depend on the empirical outputs and offers clear proof illustrated in Proposition B.1 and Theorem B.2.

## M    ANALYSIS FOR THE COMPARISONS

We analyze how such approaches, FAAP, Fair-Mixup, and ALFA improve fairness on a synthetic dataset as shown in Figure 3. In FAAP, the author generates adversarial perturbation using GAN model towards the sensitive hyperplane to make the sensitive attributes not recognizable, while trying to maintain the accuracy. In the simplified form the objective function becomes,

$$\min_{\theta}\big(\mathcal{L}_{ce}(f_\theta, \boldsymbol{x} + \boldsymbol{\delta}, y) - \mathcal{L}_{ce}(f_\theta, \boldsymbol{x} + \boldsymbol{\delta}, a)\big).$$

However, in FAAP, the perturbations are not necessarily towards the sensitive hyperplane as shown in Figure 3 (b), especially in the tabular dataset. There could potentially be two reasons for the observed discrepancies: the variations in the population sizes of each demographic group and the possible unsuitability of GAN-based perturbation for tabular datasets. Moreover, although the perturbed samples are correctly projected to the sensitive hyperplane, it doesn't necessarily lead to the fairer classifier. In Fair-Mixup, the author uses an interpolation strategy to generate data in the manifold. However, the manifold assumptions could be too strict. Moreover, although the interpolated data may compensate for the imbalance in the dataset, it doesn't take into account the unfair regions, where the misclassification rates are disproportionately high, as shown in Figure 3 (c). On the other hand, as discussed in Section 3 and Figure 1, ALFA directly discovers and covers the unfair regions to correct the classifier to become fairer.

Table 25: Experimental results with different fairness attack objective function.

| Adult | Accuracy | $\Delta DP$ | $\Delta EOd$ |
|---|---|---|---|
| Logistic | $0.8470 \pm 0.0007$ | $0.1829 \pm 0.0020$ | $0.1982 \pm 0.0077$ |
| Logistic + ALFA (Covariance) | $0.8464 \pm 0.0004$ | $0.1555 \pm 0.0013$ | $0.0616 \pm 0.0022$ |
| Logistic + ALFA (Convex) | $0.8227 \pm 0.0026$ | $0.0852 \pm 0.0078$ | $0.1547 \pm 0.0133$ |
| MLP | $0.8525 \pm 0.0010$ | $0.1824 \pm 0.0114$ | $0.1768 \pm 0.0411$ |
| MLP + ALFA (Covariance) | $0.8380 \pm 0.0045$ | $0.1642 \pm 0.0261$ | $0.0971 \pm 0.0098$ |
| MLP + ALFA (Convex) | $0.8324 \pm 0.0031$ | $0.1400 \pm 0.0166$ | $0.0904 \pm 0.0184$ |
| **COMPAS** | **Accuracy** | $\Delta DP$ | $\Delta EOd$ |
| Logistic | $0.6578 \pm 0.0034$ | $0.2732 \pm 0.0129$ | $0.5319 \pm 0.0245$ |
| Logistic + ALFA (Covariance) | $0.6682 \pm 0.0040$ | $0.0210 \pm 0.0167$ | $0.0931 \pm 0.0323$ |
| Logistic + ALFA (Convex) | $0.6740 \pm 0.0034$ | $0.0470 \pm 0.0180$ | $0.1444 \pm 0.0379$ |
| MLP | $0.6711 \pm 0.0049$ | $0.2059 \pm 0.0277$ | $0.3699 \pm 0.0597$ |
| MLP + ALFA (Covariance) | $0.6701 \pm 0.0020$ | $0.0207 \pm 0.0142$ | $0.0793 \pm 0.0418$ |
| MLP + ALFA (Convex) | $0.6624 \pm 0.0010$ | $0.0130 \pm 0.0075$ | $0.0738 \pm 0.0150$ |
| **German** | **Accuracy** | $\Delta DP$ | $\Delta EOd$ |
| Logistic | $0.7220 \pm 0.0131$ | $0.1186 \pm 0.0642$ | $0.3382 \pm 0.1268$ |
| Logistic + ALFA (Covariance) | $0.7660 \pm 0.0189$ | $0.0397 \pm 0.0261$ | $0.1596 \pm 0.0354$ |
| Logistic + ALFA (Convex) | $0.7410 \pm 0.0130$ | $0.0240 \pm 0.0179$ | $0.1030 \pm 0.0360$ |
| MLP | $0.7800 \pm 0.0150$ | $0.0454 \pm 0.0282$ | $0.2096 \pm 0.0924$ |
| MLP + ALFA (Covariance) | $0.7570 \pm 0.0024$ | $0.0053 \pm 0.0064$ | $0.0813 \pm 0.0110$ |
| MLP + ALFA (Convex) | $0.7575 \pm 0.0087$ | $0.0181 \pm 0.0120$ | $0.1960 \pm 0.0079$ |
| **Drug** | **Accuracy** | $\Delta DP$ | $\Delta EOd$ |
| Logistic | $0.6626 \pm 0.0135$ | $0.2938 \pm 0.0761$ | $0.5064 \pm 0.1616$ |
| Logistic + ALFA (Covariance) | $0.6554 \pm 0.0067$ | $0.0909 \pm 0.0261$ | $0.1170 \pm 0.0255$ |
| Logistic + ALFA (Convex) | $0.6509 \pm 0.0072$ | $0.0596 \pm 0.0198$ | $0.1284 \pm 0.0286$ |
| MLP | $0.6674 \pm 0.0096$ | $0.2760 \pm 0.0415$ | $0.4718 \pm 0.0838$ |
| MLP + ALFA (Covariance) | $0.6382 \pm 0.0104$ | $0.0820 \pm 0.0259$ | $0.1068 \pm 0.0476$ |
| MLP + ALFA (Convex) | $0.6329 \pm 0.0173$ | $0.1002 \pm 0.0826$ | $0.1955 \pm 0.0956$ |

## N  BALANCING ACCURACY AND FAIRNESS WITH A NEW HYPERPARAMETER

In the paper, we intentionally designed the framework with only one hyperparameter, $\alpha$, to maintain simplicity. Under our cost-effective setup, where fine-tuning is applied only to the last layer in the latent space, we rely on grid search to find the optimal $\alpha$ to control accuracy and fairness. However, we consider adding more control factors to balance accuracy and fairness to enhance the framework's flexibility.

To address this, we introduce an additional hyperparameter, $\lambda$, to control accuracy in ALFA by modifying Eq.5 as follows:

$$\min_{\theta} \frac{1}{|X_c| + |Z_p|} \Big( (1 - \lambda) \sum_{x_i \in X_c} \mathcal{L}_{\text{ce}}\big(g(f(x_i)), y_i, \theta\big) + \lambda \sum_{z_j \in Z_p} \mathcal{L}_{\text{ce}}(g(z_j + \delta_j^*), y_j, \theta) \Big).$$

In this setting, a lower $\lambda$ reduces the contribution of perturbed samples, resulting in higher accuracy. In the original configuration, $\lambda = 0.5$ serves as the default value. $\lambda = 0.0$ corresponds to the baseline without any fairness constraints or data augmentation.

In the COMPAS dataset, accuracy remains stable across varying $\lambda$, aligning with baseline results. In contrast, for the German and Drug datasets, accuracy decreases with increasing $\lambda$, as expected, since perturbed samples contribute more to the training objective. On the other hand, small $\lambda$ is sufficient to improve fairness, with the improvement remaining consistent across $\lambda$. However, controlling fairness purely through $\lambda$ is challenging. As demonstrated in Figures 4, 5, and 6, varying $\alpha$ provides an alternative way to influence fairness.

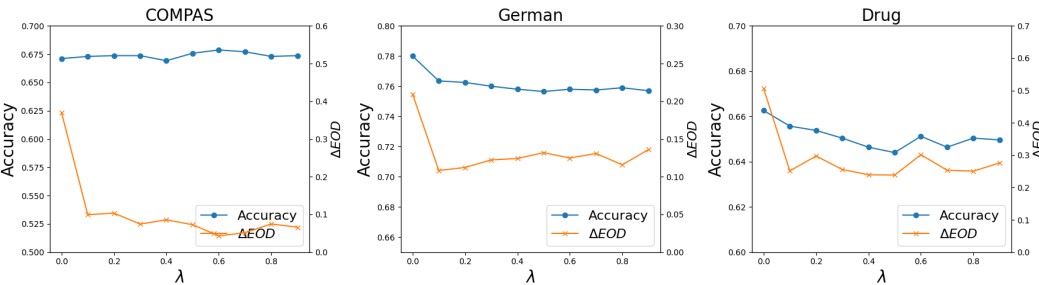

Figure 9: The impact of $\lambda$ on accuracy and fairness performance $\Delta EOd$ for the COMPAS, German, and Drug datasets.

## O   IMPACT OF REPRESENTATION SIZE

We conduct an additional ablation study to analyze the impact of representation dimensionality on fairness performance. In practice, the representation size is typically pre-defined in foundational models for computer vision and NLP tasks. However, for tabular datasets, it is feasible to train custom encoders with varying output dimensions. Specifically, for the Adult and COMPAS dataset, we vary the dimension size $d$ across $[32, 64, 128, 256, 512, 1024, 2048]$ both for the encoder's output $z \in \mathbb{R}^{n \times d}$ and perturbation $\delta \in \mathbb{R}^{n \times d}$.

The results in Figure 10 indicate that while accuracy remains consistent, larger dimensions result in improved fairness. We analyze this that larger dimensions allow for greater perturbation capacity, enabling richer representations that can more effectively attack the fairness constraint. Furthermore, richer representations provide the re-trained classifier with more detailed information, enhancing the overall fairness performance.

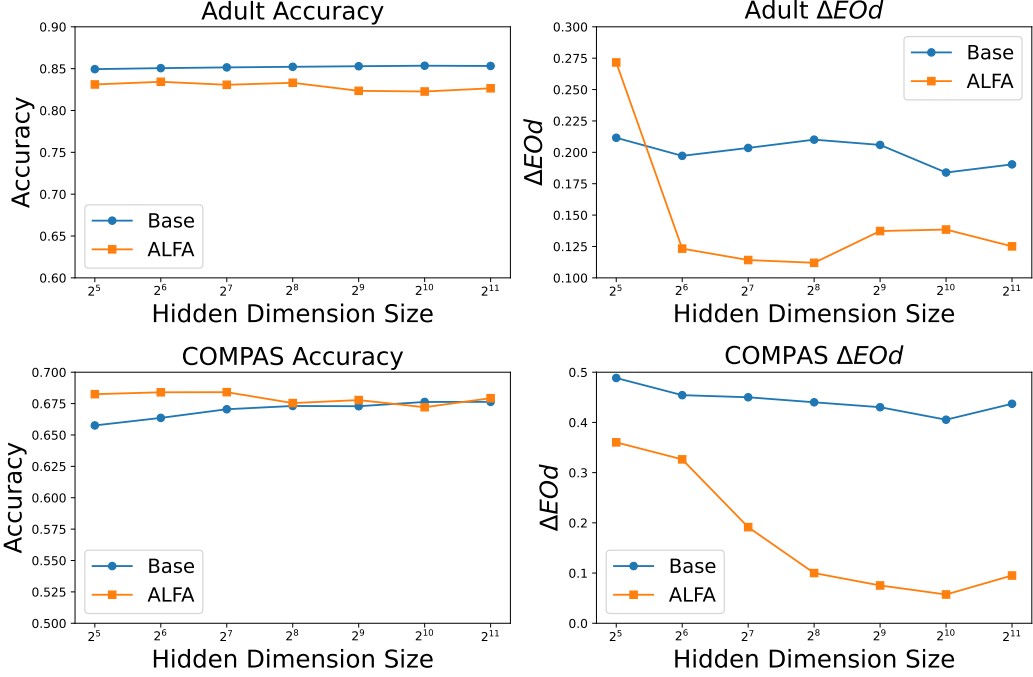

Figure 10: The impact of representation size on accuracy and fairness performance $\Delta EOd$ for the Adult and COMPAS datasets.

## P    COMPUTATIONAL RESOURCE

Table 26: Compute Resources Used for Experiments

| Component | Details |
|---|---|
| CPU | AMD Ryzen Threadripper 3960X 24-Core Processor |
| GPU | NVIDIA GeForce RTX 3090 |

