# OpenReview forum: "Adversarial Latent Feature Augmentation for Fairness"
_ICLR.cc/2025/Conference — ICLR 2025 Poster_

### Official Review · Reviewer_wUPu · 2024-10-30

**Soundness:** 3
**Presentation:** 3
**Contribution:** 3
**Rating:** 6
**Confidence:** 4

**Summary:**

The paper proposes a novel bias mitigation approach that leverages adversarial attacks on model fairness to augment samples in the latent space region where the model makes the most mistakes. The last layer of the model is then retrained with augmented latent features to correct the decision boundary for better, fairer predictions. The proposed method is evaluated on several benchmarks, showing the improvement over the baseline methods considered. Theoretical insights also back the proposed method.

**Strengths:**

1. The paper is well-written, well-organized, and relatively easy to follow.
2. The counterintuitive idea of using fairness attacks on a pretrained model to mitigate bias is interesting and novel.
3. The experiments are thorough and spanned across different types of datasets and model classes.
The proposed method is flexible and can be adapted to different types of fair constraints.
4. The results demonstrate the effectiveness of the adversarial data augmentation method in mitigating bias, and the theoretical results are intuitive.

**Weaknesses:**

1. While Figure 1 provides an intuitive comparison with related work, the related work section can be improved to amplify the paper's position. In particular, using adversarial learning to identify regions where the model makes the most mistakes is not novel and has been studied before by `Lahoti, Preethi, et al.` [1].  I recommend the authors discuss this work and consider it as a baseline for comparison.
2. The proof of theorem 3.2 seems to rely on the assumption that samples from high error regions mostly influence the gradient updates. This seems to depend on the success rate of the fairness attack, and the authors did not demonstrate that the attack can fully construct `R_unfair`.  I suggest to clarify the assumptions used in theorem 3.2.
3. It is suggested the author could perform additional experiments to show the success rate of the fairness attack on real datasets, i.e., the unfair region is actually the region targeted by the fairness attack. This will support the assumption that the attack can effectively identify `R_unfair` and oversample data points from that region. For example, predicting the accuracy of a sensitive attribute classifier in these regions and comparing it with the ground truth `R_unfair`. These experiments showing that the fairness attack effectively augments data points in the ground truth `R_unfair` would support the inherent assumption made in theorem 3.2.
4. Some parts of the paper can be improved for clarity; for example, in Figure 4-6, some baseline methods have fewer tradeoff points than others; the number of parameters controlling the tradeoff between $\Delta_{EOD}$ and Acc. should be provided. In Fig. 2(a), providing the decision boundary for different perturbation levels $\delta$ might provide better insight into the magnitude of the fairness attack over the corrected boundary.
5. Throughout the paper (e.g., Line 137), it’s stated that the method improves group fairness _without compromising accuracy_; as the results show the tradeoff between fairness and accuracy, the statement should be updated to reflect this fact. For example, the authors formulate the statement as "_improves group fairness while maintaining accuracy_ (as much as possible)". This would highlight the fact that there is a potential tradeoff.

[1] Lahoti, Preethi, et al. "Fairness without demographics through adversarially reweighted learning." Advances in neural information processing systems 33 (2020): 728-740.

**Questions:**

Please see the above Weaknesses section.

---

> ### Author Response · Authors · 2024-11-20
>
> ### W1: Comparison with ARL
> We appreciate the reviewer introducing an excellent reference! We agree that the concept of an "unfair region" in our method is analogous to the "computationally-identifiable regions with high loss" described in the ARL paper [1]. Below, we outline the key differences between our method and ARL, along with experimental comparisons on two tabular datasets.
>
> 1. **Handling Imbalanced Demographic Distributions:**
> ARL reweights samples with higher loss to improve performance for the worst-case sensitive groups. However, this strategy may struggle in scenarios with imbalanced demographic distributions. Specifically, if a small demographic group exhibits a significant false positive rate but contributes minimally to the overall loss, ARL may fail to sufficiently address the fairness issue. In contrast, our method is robust to such imbalances by employing an upsampling strategy during the fairness attack phase, ensuring that samples from unfair regions are generated stringently and effectively.
> 2. **Training Efficiency:**
> ARL requires training both the entire network and an adversarial model, which can be computationally expensive. While this is feasible for tabular datasets, it becomes prohibitively costly for larger networks like Vision Transformers (ViT) in computer vision or BERT in natural language processing. In contrast, our method leverages latent features from pre-trained models, focusing only on fine-tuning the last layer. This approach not only reduces computational cost but also increases versatility, making it applicable to a wide range of network architectures and data modalities, as demonstrated in Figures 5 and 6.
> 3. **Fairness Performance:**
> We provide a direct experimental comparison of ARL and our method (ALFA) in the table below. The results demonstrate that while ARL mitigates unfairness in classification models, its improvements are less significant compared to those achieved by our method. This highlights the distinctiveness and effectiveness of ALFA in addressing fairness concerns robustly and efficiently.
>
> **Experimental Results**
> The table below summarizes the performance comparison of ARL and ALFA on two tabular datasets. These results illustrate the superiority of our method in terms of fairness metrics and overall effectiveness:
>
> | COMPAS | Acc. | $\Delta DP$ | $\Delta EOd$|
> | -------- | -------- | -------- |-------- |
> | Logitic (Baseline) | 0.6578 $\pm$ 0.0034 | 0.2732 $\pm$ 0.0129 | 0.5319 $\pm$ 0.0245
> | Logitic (ARL) | 0.6315 $\pm$ 0.0113 | 0.2003 $\pm$ 0.0614 | 0.3951 $\pm$ 0.1299
> | Logitic (Ours) | 0.6694 $\pm$ 0.0036 | 0.0193 $\pm$ 0.0156 | 0.0876 $\pm$ 0.0354
> | MLP (Baseline) | 0.6711 $\pm$ 0.0049 | 0.2059 $\pm$ 0.0277 | 0.3699 $\pm$ 0.0597
> | MLP (ARL) | 0.6538 $\pm$ 0.0043 | 0.1556 $\pm$ 0.0208 | 0.2845 $\pm$ 0.0441
> | MLP (Ours) | 0.6702 $\pm$ 0.0021 | 0.0204 $\pm$ 0.0151 | 0.0410 $\pm$ 0.0188
>
>
> | German | Acc. | $\Delta DP$ | $\Delta EOd$|
> | -------- | -------- | -------- |-------- |
> | Logitic (Baseline) | 0.7220 $\pm$ 0.0131 | 0.1186 $\pm$ 0.0642 | 0.3382 $\pm$ 0.1268
> | Logitic (ARL) | 0.7335 $\pm$ 0.0158 | 0.0768 $\pm$ 0.0381 | 0.2726 $\pm$ 0.0837
> | Logitic (Ours) | 0.7940 $\pm$ 0.0058 | 0.0470 $\pm$ 0.0199 | 0.0469 $\pm$ 0.0276
> | MLP (Baseline) | 0.7800 $\pm$ 0.0150 | 0.0454 $\pm$ 0.0282 | 0.2096 $\pm$ 0.0924
> | MLP (ARL) | 0.7390 $\pm$ 0.0141 | 0.0447 $\pm$ 0.0192 | 0.1768 $\pm$ 0.0642
> | MLP (Ours) | 0.7570 $\pm$ 0.0024 | 0.0053 $\pm$ 0.0064 | 0.0813 $\pm$ 0.0110
>
> We will further enhance the manuscript by including ARL results in Figure 4 for a more comprehensive comparison.
>
> [1] Lahoti, Preethi, et al. "Fairness without demographics through adversarially reweighted learning." Advances in neural information processing systems 33 (2020): 728-740.

---

> > ### Author Response · Authors · 2024-11-20
> >
> > ### W2, W3: Success Rate of the Fairness Attack and Unfair Region
> > Thank you for raising this important point! The success rate of the fairness attack is indeed a critical factor for ensuring the effectiveness of the fine-tuning phase. To address this, we provide **Theorem B.2**, which demonstrates the success of our fairness attack by establishing an upper-bound relationship between the attacking objective and the empirical $\Delta \text{EOd}$ (briefly mentioned in Sec. 3.1). Specifically, Appendix B (Proposition B.1 and Theorem B.2) shows that maximizing $L_{\text{fair}}$ during the fairness attack leads to an increase in $\Delta \text{EOd}$. This ensures that the perturbations target unfair regions, allowing fine-tuning on these samples to effectively improve fairness, as validated in Theorem 3.2.
> >
> > We believe that demonstrating an increase in $\Delta \text{EOd}$ for the perturbed samples sufficiently validates the success of the attack. In response, we provide t-SNE plots for the COMPAS dataset **in Appendix O in the revised paper**, including the original dataset and the perturbed datasets (with $\alpha=0$ and $\alpha=1$, respectively) to show the success rate of the fairness attack, with the real-world dataset (COMPAS). In these visualizations, the black line represents the pre-trained decision boundary, while the red line represents the newly trained decision boundary on the combined dataset, where equal weighting is applied to the original and each perturbed dataset.
> >
> > The visualization reveals that under the pre-trained decision boundary, the perturbed samples exhibit extremely high $\Delta \text{EOd}$, indicating the success of our fairness attack. Fine-tuning on the concatenated dataset (represented by the red line) results in a corrected decision boundary that maintains accuracy while achieving significant improvements in fairness.
> >
> > Moreover, the effect of $\alpha$ aligns with our intuition. A higher $\alpha$ retains the original distribution more closely, resulting in a less corrected decision boundary and a less pronounced fairness attack. Nevertheless, both cases ($\alpha=0$ and $\alpha=1$) demonstrate significant improvements in fairness after fine-tuning.

---

> > > ### Author Response · Authors · 2024-11-20
> > >
> > > ### W4, W5: Clarification in Figure, Experiments, and Contents
> > > Thank you for pointing out the details regarding Figure 4 and the trade-off analysis. Regarding the differing number of data points for some methods in Figure 4, this is due to the omission of experimental results that did not converge well or exhibited significantly low accuracy. Despite providing detailed experimental settings for comparison methods in Appendix G, certain hyperparameter configurations resulted in non-convergent models, which were excluded from the results. For example, Fair-Mixup (Orange points) in Figure 4 consistently shows four points because the hyperparameter was varied across four values. In contrast, Fair-CDA (Purple points) often shows only two or three points, even though the hyperparameter was varied across five values, reflecting cases where some configurations failed to converge properly.
> > > Although Appendix G specifies the hyperparameters used in the experiments, we will add a clarification in the main text to explicitly state that non-convergent results have been omitted.
> > >
> > > For the impact of perturbation on decision boundary correction (regarding Figure 2 (a)), please see the previous response (W3) with the **Appendix O in the revised paper**.
> > >
> > > Regarding the statement about the fairness-accuracy trade-off, we will revise it according to the reviewer’s suggestion. We acknowledge that there may be a potential fairness-accuracy trade-off within our framework, and we will refine the wording to ensure this nuance is captured appropriately.

---

> > > > ### Comment · Reviewer_wUPu · 2024-11-21
> > > >
> > > > I would like to thank the authors, and I appreciate their efforts in addressing my concerns. Below are follow-up points for discussion;
> > > >
> > > > **Comparison with ARL**
> > > > - Handling Imbalanced Demographic Distributions. I do not understand the authors’ argument about the issue of handling imbalanced demographic distributions, which is the regime where methods such as ARL are designed to operate. How can _a small demographic group exhibiting a significant false positive rate contribute minimally to the overall loss_? If this is the case, ALFA would have the same issue targeting these groups since they would not influence the decision boundary of the pretrained model enough for the fairness attack to target them. On the other hand, performing upsampling in unfair regions can be similar to upweighting data points in these regions.
> > > >
> > > > - On the training efficiency. Please note that, as discussed in the ARL paper, a linear adversary performs well, and ALPHA involves fairness attacks and fine-tuning the last layer (which is also a linear model). ARL can also be used upon latent representations; therefore, the authors do not provide sufficient evidence to support the claim that ALPHA is more computationally efficient.
> > > >
> > > > - Regarding the fairness performance. Thank you for the additional results. They indeed show the efficiency of ALPHA in targeting the unfair region and improving the fairness of the downstream model, which does it better than ARL. However, for a fair comparison, the authors should clarify that ARL operates without demographic information, which makes it applicable in broader settings where demographic information is not available (for reasons such as privacy restrictions).  While I understand it is out of the scope of the paper, the contribution could be stronger if the authors discuss how they can make ALPHA rely less on demographic data. Without using demographic information for the fairness attack, a relationship likely exists between distance to the decision boundary (misclassification) and the unfair region.
> > > >
> > > > **Success Rate of the Fairness Attack and Unfair Region**
> > > >
> > > > I thank the authors for the additional experimental results and theorem showing how the attack can control unfairness. The proofs are sound and valuable.
> > > >
> > > > For the continued discussion, I would like the authors to provide some clarification;
> > > > 1. In the definitions of $\Delta_{dp}$  and $\Delta DP$ (similarly for EOd). Please clarify the relationship between $d_i$ and $\hat{y}_{i}$ in measuring the corresponding fairness metrics. In particular, how is $d_i$ restricted to positive outcomes as measured in  $\Delta DP$
> > > >
> > > > 2. Please discuss how strong the assumption of the piecewise linear function is for real-world data and how we can interpret the difference between the number of samples in each group within each segment measured by the constant value ($C, C_0, C_1$) in the upper bounds.
> > > > 3. I still have some concerns about the effect of the perturbation on the decision boundary, particularly regarding fairness. For the newly added Figure 8, can the authors clarify:
> > > > - why does the corrected model have better fairness for $\alpha=1$, meaning a less pronounced fairness attack, than for $\alpha=0$, where the attack is more pronounced, and the decision boundary is further corrected?
> > > > - what happens in between, i.e., $\alpha=0.5$, for example?
> > > > - what $\alpha$ controls in terms of fairness in the corrected model.
> > > >  I expected better fairness for more pronounced fairness attacks. Please clarify if I missed some parts of the method.

---

> > > > > ### Author Response · Authors · 2024-11-22
> > > > >
> > > > > ### Comparison with ARL
> > > > >
> > > > > #### Handling Imbalanced Demographic Distributions
> > > > >
> > > > > In ARL, the 'computationally identifiable region' is defined by high loss:
> > > > >
> > > > > $$
> > > > > J(\theta, \phi) = \min_{\theta} \max_{\phi} \sum_{i=1}^n \Bigl(1+n\cdot\frac{f_{\phi} (x_i, y_i)}{\sum_{1}^n f_{\phi} (x_i, y_i)} \Bigr)\cdot \ell_{ce} (h_\theta (x_i),y_i),
> > > > > $$
> > > > >
> > > > > where $h$ is the classifier and $f$ is the adversary. Assume $a=1$ is the privileged group, where $\text{Acc}\_{y=1}\^{a=0} < \text{Acc}\_{y=1}\^{a=1}$ and $\text{Acc}\_{y=0}\^{a=1} < \text{Acc}\_{y=0}\^{a=0}$, resulting in high misclassification rates for subgroups $S_{y=1}^{a=0}$ and $S_{y=0}^{a=1}$ compared to bias-aligned groups $S_{y=1}^{a=1}$ and $S_{y=0}^{a=0}$. Let's simplify the problem by considering only two subgroups with high misclassification rates: $S_{y=1}^{a=0}$ and $S_{y=0}^{a=1}$. The term $\ell_{ce} (h_\theta (x_i),y_i)$ can be expressed as:
> > > > >
> > > > > $$
> > > > > \sum_{S_{y=1}^{a=0}}\ell_{ce} (h_\theta (x_i),y_i)+\sum_{S_{y=0}^{a=1}}\ell_{ce} (h_\theta (x_i),y_i).
> > > > > $$
> > > > >
> > > > > If $|S_{y=0}^{a=1}| \gg |S_{y=1}^{a=0}|$, the entire term $\ell_{ce} (h_\theta (x_i),y_i)$ is dominated by the loss of the group $S_{y=0}^{a=1}$ during adversary training. Consequently, the adversary $f$ has less impact in mapping the minor group $S_{y=1}^{a=0}$ to the computationally identifiable region.
> > > > >
> > > > > On the other hand, in the proposed method, ALFA, during the attack step to perturb samples toward the unfair region, we employ an upsampling strategy that ensures all subgroups are equal in size, i.e.,
> > > > >
> > > > > $$
> > > > > |S_{y=0}^{a=0}| = |S_{y=0}^{a=1}| = |S_{y=1}^{a=0}| = |S_{y=1}^{a=1}|,
> > > > > $$
> > > > >
> > > > > thereby ensuring the attack is not influenced by data imbalance.
> > > > >
> > > > > Moreover, we would like to clarify the framework's workflow regarding upsampling. The upsampling strategy is designed to accurately identify the unfair region before fine-tuning the classifier on the data points within this region. While we understand the reviewer's concern that "performing upsampling in unfair regions can be similar to upweighting data points in these regions," this scenario would only apply if the unfair region were already known. In our method, however, the upsampling strategy is specifically utilized to detect the unfair region. Thus, the interpretation of "upsampling in unfair regions for fairness" does not fully align with the intended design of our approach.

---

> > > > > > ### Author Response · Authors · 2024-11-22
> > > > > >
> > > > > > #### Training Efficiency and Fairness Performance
> > > > > > Thank you for suggesting these valuable perspectives! We agree that ARL can be applied to latent representations while keeping the encoder fixed. In this scenario, while the training efficiency of ARL and our method (ALFA) might be comparable, ALFA still offers an advantage in terms of stability. Iteratively updating both the adversary and the model in ARL often leads to poor algorithmic stability, as noted in [2], making iterative adversarial training less appealing in scenarios where stability is critical.
> > > > > >
> > > > > >
> > > > > > We also acknowledge that ARL addresses broader scenarios where demographic data is unavailable, which differs from the focus of our approach. To explore this, we modified our method to operate without demographic information by incorporating pseudo-labeling through k-means clustering. We conducted a comparison between ARL and ALFA under more comparable settings. Specifically, ARL was run without demographic information with two configurations, training the entire network versus fine-tuning the last layer only. ALFA is conducted in two configurations with or without demographic information. Therefore, the bolded experiments are in the same experimental setting, last-layer fine-tuning without demographic information. To demonstrate applicability, we adopt MLP and ResNet-like architectures as backbones for our experiments.
> > > > > >
> > > > > > As a result, ALFA remains effective in achieving fairness even without demographic information and outperforms ARL's two configuration, the entire-network training and last-layer fine-tuning scenarios. This result showcases the robustness of ALFA's approach, particularly in its ability to maintain strong fairness performance under comparable settings by leveraging pseudo labeling.
> > > > > >
> > > > > > | COMPAS-MLP | Acc. | $\Delta DP$ | $\Delta EOd$|
> > > > > > | -------- | -------- | -------- |-------- |
> > > > > > | Baseline | 0.6711 $\pm$ 0.0049 | 0.2059 $\pm$ 0.0277 | 0.3699 $\pm$ 0.0597
> > > > > > | ARL, Entire | 0.6538 $\pm$ 0.0043 | 0.1556 $\pm$ 0.0208 | 0.2845 $\pm$ 0.0441
> > > > > > | **ARL, Last-Layer** |0.6504 $\pm$ 0.0030|0.1247 $\pm$ 0.0194|0.2220 $\pm$ 0.0425|
> > > > > > | Ours, w/ Demo. | 0.6702 $\pm$ 0.0021 | 0.0204 $\pm$ 0.0151 | 0.0410 $\pm$ 0.0188
> > > > > > | **Ours, w/o Demo.** |**0.6662 $\pm$ 0.0019**|**0.1051 $\pm$ 0.0125**|**0.1611 $\pm$ 0.0269**|
> > > > > >
> > > > > > | COMPAS-ResNet-like | Acc. | $\Delta DP$ | $\Delta EOd$|
> > > > > > | -------- | -------- | -------- |-------- |
> > > > > > | Baseline |0.6753 $\pm$ 0.0037|0.2055 $\pm$ 0.0307|0.3683 $\pm$ 0.0700|
> > > > > > | ARL, Entire |0.6541 $\pm$ 0.0046|0.2069 $\pm$ 0.0187|0.3944 $\pm$ 0.0418|
> > > > > > | **ARL, Last-Layer** |0.6489 $\pm$ 0.0017|0.1561 $\pm$ 0.0260|0.2894 $\pm$ 0.0583|
> > > > > > | Ours, w/ Demo. |0.6756 $\pm$ 0.0032|0.0124 $\pm$ 0.0137|0.0659 $\pm$ 0.0316|
> > > > > > | **Ours, w/o Demo.** |**0.6500 $\pm$ 0.0008**|**0.0984 $\pm$ 0.0027**|**0.1484 $\pm$ 0.0054**|
> > > > > >
> > > > > >
> > > > > > | German-MLP | Acc. | $\Delta DP$ | $\Delta EOd$|
> > > > > > | -------- | -------- | -------- |-------- |
> > > > > > | Baseline | 0.7800 $\pm$ 0.0150 | 0.0454 $\pm$ 0.0282 | 0.2096 $\pm$ 0.0924
> > > > > > | ARL, Entire | 0.7555 $\pm$ 0.0082 | 0.0257 $\pm$ 0.0156 | 0.1127 $\pm$ 0.0441
> > > > > > | **ARL, Last-Layer** |0.7425 $\pm$ 0.0081|0.0155 $\pm$ 0.0066|0.0947 $\pm$ 0.0309|
> > > > > > | Ours, w/ Demo. | 0.7570 $\pm$ 0.0024 | 0.0053 $\pm$ 0.0064 | 0.0813 $\pm$ 0.0110
> > > > > > | **Ours, w/o Demo.** |**0.7705 $\pm$ 0.0015**| **0.0043 $\pm$ 0.0005**|**0.0898 $\pm$ 0.0030**|
> > > > > >
> > > > > >
> > > > > >  German-ResNet-like | Acc. | $\Delta DP$ | $\Delta EOd$|
> > > > > > | -------- | -------- | -------- |-------- |
> > > > > > | Baseline |0.7595  $\pm$ 0.0224|0.0524 $\pm$ 0.0368|0.2193 $\pm$ 0.1224|
> > > > > > | ARL, Entire |0.7295 $\pm$ 0.0189 | 0.0325 $\pm$ 0.0177 | 0.1093 $\pm$ 0.0519|
> > > > > > | **ARL, Last-Layer** |**0.7120 $\pm$ 0.0040**|0.0610 $\pm$ 0.0084|0.0710 $\pm$ 0.0180|
> > > > > > | Ours, w/ Demo. |0.7325 $\pm$ 0.0186| 0.0309 $\pm$ 0.0147 | 0.0665 $\pm$ 0.0228|
> > > > > > | **Ours, w/o Demo.** |0.7065 $\pm$ 0.0023| **0.0583 $\pm$ 0.0060**| **0.0708  $\pm$  0.0075**|
> > > > > >
> > > > > >
> > > > > >
> > > > > >
> > > > > >
> > > > > > [2] Xing, Yue, Qifan Song, and Guang Cheng. "On the algorithmic stability of adversarial training." Advances in neural information processing systems 34 (2021): 26523-26535.

---

> > > > > > > ### Author Response · Authors · 2024-11-22
> > > > > > >
> > > > > > > ### Success Rate of the Fairness Attack and Unfair Region
> > > > > > > We sincerely appreciate the reviewer's detailed feedback and are happy to provide clarifications.
> > > > > > >
> > > > > > >
> > > > > > > #### Relationship between $d_i$ and $\hat{y}$
> > > > > > > The signed distance $d_i$ refers to the logit value after the linear classifier but before the sigmoid function. It is originally defined as the inverse sigmoid of the predicted probability $\hat{y}$. The signed distance explicitly quantifies how far a data point is from the decision boundary and on which side it lies. In our paper, we approximate the inverse sigmoid function using a piecewise linear approach to ensure the framework is provable.
> > > > > > >
> > > > > > > Importantly, $d_i$ is not restricted to positive values, as it is used to measure the distance gap in mean signed distances between two groups. Specifically,
> > > > > > > $$\Delta d_{dp}=\Bigl \vert \frac{1}{n_1} \sum_{i\in S_1} d_i  - \frac{1}{n_0} \sum_{j\in S_0} d_j \Bigr \vert,$$ where $S_1$ and $S_0$ represent two demographic groups. Since $\Delta d_{dp}$ does not require $d_i$ to be positive, the signed distance can represent disparities effectively.
> > > > > > >
> > > > > > > Moreover, $\Delta d_{dp}$ is proportional to the objective function $L_{fair}$ and serves as a lower bound for the empirical $\Delta DP$ in Eq.(15). Therefore, we claim that maximizing $L_{fair}$ induces unfairness, allowing the identification of samples within the unfair region.
> > > > > > >
> > > > > > >
> > > > > > > #### Regarding Piecewise Linear Assumption
> > > > > > > While it is mentioned in Line 821, our method indeed does not inherently **assume** that the inverse sigmoid function is approximated by a piecewise linear function for real-world data. Initially, the predicted probability $\hat{y}$ is produced by the standard sigmoid function. The piecewise linear approximation is **utilized** to effectively address the fairness constraint and to make the framework provable. We argue that the selection of the piecewise linear function is a methodological choice and does not depend on any underlying assumption about the data or model.
> > > > > > >
> > > > > > >  The constants $(C, C_0, C_1)$ are derived parameters appearing in theoretical fairness bounds, where they encapsulate the impact of group sample imbalances across segments. This $C$ is negative value, directly influences the increase in $\Delta DP$ to maintain the inequality in Eq.(16). Specifically, $C$ is defined as:
> > > > > > > $$
> > > > > > > C = \frac{2}{N} \sum_{k=1}^m (n_1^{(k)} - n_0^{(k)})b_k = \frac{2}{N} \sum_{k=1}^m C_k,
> > > > > > > $$
> > > > > > > where $C_k = r_k \cdot b_k$ and $r_k = (n_1^{(k)} - n_0^{(k)})$. As shown in Appendix B, under the assumption $\text{Cov}(a, y) > 0$ and using the piecewise linear approximation of the inverse sigmoid, $b_k$ is always negative, and $b_1 \geq b_m$. For example, when $m = 10$, the values of $b_k$ are given by:
> > > > > > > $$
> > > > > > > b_k = [-16.12, -3.01, -2.46, -2.17, -2.03, -2.03, -2.25, -2.93, -5.10, -123.09].
> > > > > > > $$
> > > > > > > Given that the pretrained classifier exhibits bias, $r_k$ is negative in earlier segments and positive in later segments. Considering $|b_m| \geq |b_1|, |b_{m-1}| \geq |b_2|, \dots, |b_{m/2+1}| \geq |b_{m/2}|$, it follows that $C$ is negative.
> > > > > > > The fairness bound is expressed as:
> > > > > > > $$
> > > > > > > \mathcal{L}\_{\text{fair}} = \frac{1}{4} \Delta d\_{dp} \leq \frac{1}{4} \bigl( a\_{\text{max}} \Delta DP + C \bigr).
> > > > > > > $$
> > > > > > > Since $C < 0$, $\Delta DP$ must increase to offset the negative contribution of $C$ and maintain the inequality while $\mathcal{L}\_{\text{fair}}$ is maximized. This demonstrates the role of $C$ in influencing $\Delta DP$ to ensure that the fairness attack objective $\mathcal{L}_\{\text{fair}}$ remains within the theoretical bounds.

---

> > > > > > > > ### Author Response · Authors · 2024-11-22
> > > > > > > >
> > > > > > > > #### Clarification on $\alpha$ in Figure 8
> > > > > > > >
> > > > > > > > As the reviewer correctly noted, a larger $\alpha$ indicates a less pronounced fairness attack, while $\alpha = 0$ represents the most pronounced attack. To provide greater clarity on our framework, **we have revised Figure 8 in the paper** by expanding the range of $\alpha$ values. The reported accuracy and fairness performance now reflect results on the original COMPAS dataset rather than those from dimensionality-reduced samples (e.g., those visualized using t-SNE).
> > > > > > > >
> > > > > > > > Our observations remain consistent: as $\alpha$ decreases, the EOd value on the pretrained decision boundary increases, indicating a more effective fairness attack. Visually, smaller $\alpha$ values result in greater correction meaning that they push more samples into the unfair region.
> > > > > > > >
> > > > > > > > However, a pronounced fairness attack does not necessarily lead to improved fairness in test data. For instance, if the perturbation introduced by the attack is too large, it may generate samples that extend beyond the unfair region. This phenomenon is further illustrated in Figure 2(b), where excessive perturbations can degrade fairness performance. Additionally, during fine-tuning, a successful fairness attack on the training set does not directly translate into fairness improvements on the test set, as the original dataset is also involved in fine-tuning.
> > > > > > > >
> > > > > > > > In our experimental results, while the success of fairness attacks is clearly observed by small $\alpha$, e.g. $\alpha=0$, the best test EOd is achieved when $\alpha = 1$. Currently, identifying the optimal $\alpha$ for the test set relies on grid search. We acknowledge this limitation and propose exploring alternative optimization strategies as future work.

---

> ### Comment · Reviewer_wUPu · 2024-11-25
>
> I thank the authors for their detailed response and the clarifications provided.
>
> The experiments with **ALPHA w/o Demo** show promising results and strengthen the paper's contribution.
> The use of pseudo-labels, however, has the risk of amplifying bias in some settings, especially for k-mean, see [1,2] for reference. The authors should discuss this risk in the description of _ALPHA w/o Demo_ in the revised paper.
>
> One main concern remaining, which was also raised in other reviews, is that the method fails to control the tradeoff between fairness and accuracy efficiently (Figure 8 shows that). This is a pretty important issue, as practitioners need actionable tools to satisfy certain business needs at a specific level of fairness.
>
> Overall, I remain positive about the work and intend to maintain my positive score.
>
>
> [1] Kenfack, P. J., et al. "Fairness under demographic scarce regime." Transactions on Machine Learning Research. 2024
>
> [2] Awasthi, Pranjal, et al. "Evaluating fairness of machine learning models under uncertain and incomplete information." Proceedings of the 2021 ACM conference on fairness, accountability, and transparency. 2021.

---

> > ### Author Response · Authors · 2024-11-27
> >
> > ### Regarding ALFA Without Demographic Information
> >
> > Thank you for raising this concern and providing valuable references! To summarize:
> > - [1] argues that incorporating uncertain sensitive attributes into fairness constraints can negatively affect the accuracy-fairness trade-off.
> > - [2] demonstrates that using pseudo-labels in the evaluation step is not effective.
> >
> > The reviewer's concern about our method using pseudo-labeling aligns more closely with the risks identified in [1].
> >
> > We now provide an analysis explaining how ALFA avoids these risks. Notably, the latent space distribution, as shown in Figure 1, remains consistent even without access to demographic information as the pretrained model is trained only with the class label. Below are the potential scenarios and their implications:
> >
> > 1. **Clustering Reflects Sensitive Attributes:**
> >    If k-means clustering divides the training data based on the sensitive attribute, i.e.,
> >    $$ \\mathcal{D} = \\bigl[(\\mathcal{S}\_{a=0}\^{y=0} \cup \\mathcal{S}\_{a=0}\^{y=1}); (\\mathcal{S}\_{a=1}\^{y=0} \cup \\mathcal{S}\_{a=1}\^{y=1})\bigr],$$
> >    this aligns with the intended behavior, posing no risk as identified in either [1] or [2].
> >
> > 2. **Clustering Reflects Class Labels:**
> >    If k-means clustering groups the data based on class labels, i.e.,
> >    $$
> >    \mathcal{D} = \bigl[(\mathcal{S}\_{a=1}\^{y=1} \cup \mathcal{S}\_{a=0}\^{y=1}); (\mathcal{S}\_{a=1}\^{y=0} \cup \mathcal{S}\_{a=0}\^{y=0})\bigr],
> >    $$
> >    ALFA might not detect unfair regions in this case. However, the perturbations introduced by ALFA push samples away from the decision boundary in the correct direction, making the latent space linearly separable. This improves both accuracy and fairness.
> >
> > 3. **Clustering Based on Biased Decision Boundaries:**
> >    If clustering is influenced by a biased decision boundary that correlates $a=1$ with $y=1$, the clusters may mix subgroups, e.g.,
> >    $$
> >    \mathcal{D}\_1 = \mathcal{S}\_{a=1}\^{y=1} \cup \mathcal{S}\_{a=0}\^{y=1} \cup \mathcal{S}\_{a=1}\^{y=0}, \quad
> >    \mathcal{D}\_0 = \mathcal{S}\_{a=0}\^{y=0} \cup \mathcal{S}\_{a=0}\^{y=1} \cup \mathcal{S}\_{a=1}\^{y=0}.
> >    $$
> >    In such cases, the mixed clusters contribute both to detecting unfair regions and improving separability. For example, in $\mathcal{D}\_1$, $\mathcal{S}\_{a=1}\^{y=0}$ detects unfair regions, $\mathcal{S}\_{a=0}\^{y=1}$ improves separability, and $\mathcal{S}\_{a=1}\^{y=1}$ contributes to both effects.
> >
> > 4. **Clusters with All Subgroups:**
> >    The only risk occurs if $\mathcal{D}\_1$ and $\mathcal{D}\_0$ contain samples from all subgroups. In this case, wrongly clustered samples, e.g., $\mathcal{S}\_{a=0}\^{y=0}$ in $\mathcal{D}\_1$, though less likely, may disturb both increasing separability and detecting unfair regions. This remains a limitation, while the correctly clustered samples continue to serve a dual role in improving separability and fairness.
> >
> >
> > These scenarios demonstrate the robustness of ALFA under different clustering outcomes if we utilize k-means clusering when the demographic information is not avaiable. While there are edge cases where clustering may not perfectly align with detecting unfair regions, ALFA's flexibility and effectiveness in addressing fairness concerns without demographic information remain significant strengths.
> >
> >
> > [1] Kenfack, P. J., et al. "Fairness under demographic scarce regime." Transactions on Machine Learning Research. 2024
> > [2] Awasthi, Pranjal, et al. "Evaluating fairness of machine learning models under uncertain and incomplete information." Proceedings of the 2021 ACM conference on fairness, accountability, and transparency. 2021.

---

> > > ### Author Response · Authors · 2024-11-27
> > >
> > > ### Controlling Accuracy-Fairness Trade-Off
> > >
> > > Thank you for highlighting the concern regarding performance control in ALFA.
> > >
> > > In our paper, we intentionally designed the framework with only one hyperparameter, $\alpha$, to maintain simplicity.
> > > - Under our cost-effective setup, where fine-tuning is applied only to the last layer in the latent space, we rely on grid search to find the optimal $\alpha$ to control accuracy and fairness.
> > > - However, we consider adding more control factors to balance accuracy and fairness to enhance the framework’s flexibility.
> > >
> > > To address this, we introduce an additional hyperparameter, $\lambda$, to control accuracy in ALFA by modifying Eq.(5) as follows:
> > > $$\\min\_{\\theta}\\frac{1}{\vert X\_c\vert + \vert Z\_p\vert} \Bigl (   (1-\\lambda)\\sum\_{{x}\_i \\in {X}\_c}\\mathcal{L}\_{\text{ce}} \\bigl(g(f({x}\_i)), y\_i ,\\theta\\bigr)  +  \\lambda\\sum\_{{z}\_j\in{Z}\_p} \\mathcal{L}\_{\text{ce}} (g({z}\_j + {\delta}\_j^*), y\_j ,\theta) \Bigr ).$$
> > > In this setting:
> > > - A lower $\lambda$ reduces the contribution of perturbed samples, resulting in higher accuracy.
> > > - In the original configuration, $\lambda=0.5$ serves as the default value.
> > > - $\lambda=0.0$ corresponds to the baseline without any fairness constraints or data augmentation.
> > >
> > > The experimental results are provided **Figure 10 in Appendix Q** of the revised paper.
> > >
> > > **Observations from Additional Experiments**
> > >
> > > - **Accuracy Trends**:
> > >   In the COMPAS dataset, accuracy remains stable across varying $\lambda$, aligning with baseline results (see Figure 4 and Table 8). In contrast, for the German and Drug datasets, accuracy decreases with increasing $\lambda$, as expected, since perturbed samples contribute more to the training objective.
> > > - **Fairness Trends**:
> > >   Small $\lambda$ is sufficient to improve fairness, with the improvement remaining consistent across $\lambda$. However, controlling fairness purely through $\lambda$ is challenging. As demonstrated in Figures 4, 5, and 6, varying $\alpha$ provides an alternative way to influence fairness.
> > >
> > >
> > > We acknowledge the reviewer's concern regarding the trade-off between fairness and accuracy. Our additional experiments demonstrate that $\lambda$ provides a clear mechanism to manage accuracy. However, controlling fairness remains a challenge, even though small values of $\lambda$ consistently lead to significant improvements in fairness.
> > >
> > > As future work, we aim to develop an analytic framework to better balance fairness and accuracy. Despite this limitation, $\lambda$ enhances ALFA's flexibility, providing practitioners with a practical tool to align performance with specific business objectives.

---

### Official Review · Reviewer_ZWM9 · 2024-11-02

**Soundness:** 2
**Presentation:** 3
**Contribution:** 2
**Rating:** 6
**Confidence:** 4

**Summary:**

This paper introduces ALFA (Adversarial Latent Feature Augmentation), an approach to enforce Equalized Odds fairness in binary classification through adversarial data augmentation. The method generates adversarial perturbations in the latent space by maximizing the covariance between sensitive attributes and logits of the predicted probability. The approach operates in two phases: first creating biased latent features through fairness attacks, then fine-tuning on both original and perturbed features using a dual cross-entropy loss to balance fairness and accuracy. Comprehensive experiments across multiple datasets (Adult, COMPAS, German, Drug, Wikipedia and CelebA) demonstrate ALFA's effectiveness in improving Equalized Odds while maintaining competitive predictive performance.

**Strengths:**

- The paper presents a novel integration of adversarial perturbations and data augmentation in latent space to address fairness constraints.
- The proposed method achieves competitive results

**Weaknesses:**

1) **Problem Formulation and Distribution Mismatch:**
The fairness enforcement is performed on artificially balanced demographic data (n00 = n01 = n10 = n11), while the test distribution could be imbalanced. This fundamental distribution mismatch is not adequately addressed.
The method achieves Demographic Parity proportional to Equalized Odds on this really specific case, which contradicts the known incompatibility between these metrics and loses the essential dependence between Y and A.

2) **Incomplete Baseline Comparisons:**
The experimental comparisons raise concerns about completeness and fair representation of SOTA methods:
On standard datasets (German/Adult and Wikipedia - Figure 6), the method achieves EO violations no better than 0.1, while existing approaches [1] show that Zafar et al. (2013) using covariance loss, typically achieve EO violations close to 0.03 under the same test conditions on adult uci(20% test split).
This performance gap suggests the experimental evaluation may not fully represent the capabilities of existing state-of-the-art methods

3) **Technical Gaps:**
The methodology lacks sufficient technical explanations and critical implementation details are missing, particularly regarding the generation of δ

[1] Andrew Lowy, Sina Baharlouei, Rakesh Pavan, Meisam Razaviyayn, Ahmad Beirami:
A Stochastic Optimization Framework for Fair Risk Minimization. Trans. Mach. Learn. Res. 2022 (2022)

**Questions:**

The authors' responses have addressed most of my concerns. Based on the current revision, I have updated my rating to 6.

##################################


Upsampling Strategy and Distribution Mismatch:

1) How is the upsampling implemented - are instances repeated multiple times to achieve Np = 4 · max(n00, n01, n10, n11)?
2) My main concern is regarding that the fairness improvement by reducing ∆EOd (i.e., ∆EOd(θp) ≤ ∆EOd(θ)) is demonstrated on this upsampled distribution rather than the original dataset. For a test set (20% of the original data) that maintains its original unbalanced distribution, how does your method guarantee fairness on the test data, particularly with highly unbalanced combinations of A and Y?  Being fair on the upsampled training data does not necessarily imply fairness on the original distribution present in your test set.

Training Procedure:

3) Why does the method use a single pass of the two-step process, unlike traditional adversarial learning approaches where the adversary and model are iteratively updated multiple times?

Technical Details of δ Implementation:

4) How exactly is δ maximized? Is it implemented as a neural network trained via gradient descent?
5) Does the maximization of δ take $X_p$ as input?
6) What mathematical constraints ensure proximity in latent space? Are there Lipschitz constraints on δ maximization?

Loss Function Balance:

7) Equation 5 uses equal weighting between the balanced and unbalanced distributions, which shifts half the focus to the upsampled distribution. Without a hyperparameter to control this split, how can the model maintain proper task accuracy? Even when fairness enforcement is not needed (i.e., when α approaches infinity), the attention remains partially shifted to the balanced distribution $D(X_p,Y_p,A_p)$, which could differs from the test distribution.

8) Could this equal weighting explain why your method shows lower accuracy than SOTA in the unfair regions of Figure 4? Additionally, the blue curve is largely absent in unfair regions (high α values) except for Adult UCI dataset. Did other datasets also show poor performance in these regions with higher values of α?"

---

> ### Author Response · Authors · 2024-11-20
>
> ### W1, Q1, Q2: Problem Formulation and Distribution Mismatch
> The reviewer's concern about distribution mismatch is not applicable to our method. While it is true that we utilize an upsampling strategy during the fairness attack to ensure effective adversarial performance, this strategy does not interfere with the test distribution's inherent imbalance. Specifically, the upsampling is applied as a data augmentation step during the fairness attack phase by random resampling to match $N_p = 4\cdot\max \bigl(n_{00}, n_{01}, n_{10}, n_{11} \bigr)$ for effective attacking proved by Proposition B.1 and Theorem B.2. However, our method still incorporates the original data distribution in training, as shown in Eq. (5). Specifically, the model is trained on a combination of the original data and adversarially perturbed data, ensuring that the test distribution's imbalance is appropriately addressed.
>
> Moreover, upsampling (or reweighting) strategies are well-established in fairness literature [1, 2, 3, 4, 5] as effective techniques for enhancing fairness by ensuring adequate representation of all demographic groups during training. These techniques are not intended to enforce a specific distribution on the test set but rather to provide a balanced view of the decision boundary during the training process.
>
> To summarize, our upsampling strategy during the fairness attack phase is purely a data augmentation technique. It enhances the attack's ability to identify and address unfair regions without altering the original distributional characteristics of the data used during training or evaluation.
>
> [1] Kabir, Md Alamgir, et al. "Balancing fairness: unveiling the potential of SMOTE-driven oversampling in AI model enhancement." Proceedings of the 2024 9th International Conference on Machine Learning Technologies. 2024
> [2] Yan, Bobby, Skyler Seto, and Nicholas Apostoloff. "Forml: Learning to reweight data for fairness." arXiv preprint arXiv:2202.01719 (2022).
> [3] Li, Peizhao, and Hongfu Liu. "Achieving fairness at no utility cost via data reweighing with influence." International Conference on Machine Learning. PMLR, 2022.
> [4] Dablain, Damien, Bartosz Krawczyk, and Nitesh Chawla. "Towards a holistic view of bias in machine learning: Bridging algorithmic fairness and imbalanced learning." arXiv preprint arXiv:2207.06084 (2022).
> [5] Chai, Junyi, and Xiaoqian Wang. "Fairness with adaptive weights." International Conference on Machine Learning. PMLR, 2022.

---

> > ### Author Response · Authors · 2024-11-20
> >
> > ### W2: Baseline Comparison
> > We appreciate the reviewer highlighting concerns about baseline comparisons. We would like to emphasize that our implementation and experimental settings are designed to be more stringent and comprehensive compared to typical practices in the literature reviwer provided [6]. Specifically:
> >
> > 1. **Random Splitting of Data:**
> >    In our experiments, datasets are randomly split into training and test sets for each run. This ensures that the label and sensitive attribute distributions in the test set are not biased or hand-tuned, but may differ from those reported in other studies. To account for these variations, we conducted our experiments across 10 independent runs and reported both the mean and standard deviation. This approach not only reflects the performance of the method under diverse conditions but also ensures that our evaluations are robust and statistically sound, providing a stricter benchmark.
> >
> > 2. **Stricter Fairness Metric Definitions:**
> > The code implementation used in the referenced paper differs from ours in how fairness metrics are computed. Specifically, in the implementation suggested by the reviewer [6], $\Delta \text{DP}$ is calculated as:
> >    $$
> >    \Delta \text{DP} = \left| \text{mean}(p(\hat{y}=1|a=0)) - \text{mean}(p(\hat{y}=1|a=1)) \right|.
> >    $$
> > While this definition captures overall differences in positive prediction probabilities, this **soft metric** may overlook significant unfairness in certain scenarios. For instance, if all samples are $p(y=1|a=1) \approx 0.51$ and $p(y=1|a=0) \approx 0.49$, the resulting metric $\Delta \text{DP} = 0.02$ might suggest fairness. However, under a classification threshold of 0.5, all predictions would yield $\\hat{y}\_{a=1} = 1$ and $\\hat{y}\_{a=0} = 0$, which reflects a starkly unfair outcome.
> > To avoid this potential inaccuracies or misinterpretations, we adopt stricter definitions of fairness metrics by using **hard metric** after binarization:
> >    $$
> >    \Delta \text{DP} = \left| \frac\{\text{sum}(\\mathbb{I}\_{a=1}\^{\hat{y}=1})\}\{n\_{a=1}\} - \frac\{\text{sum}(\mathbb{I}\_{a=0}\^{\hat{y}=1})\}\{n\_{a=0}\} \right|.
> >    $$
> >    $$
> >    \Delta \text{EOd} = \left| \frac\{\text{sum}(\mathbb{I}\_{a=1, y=1}\^{\hat{y}=1})\}\{n\_{a=1, y=1}\} - \frac\{\text{sum}(\mathbb{I}\_{a=0, y=1}\^{\hat{y}=1})\}\{n\_{a=0, y=1}\} \right| +
> > \left| \frac\{\text{sum}(\mathbb{I}\_{a=1, y=0}\^{\hat{y}=1})\}\{n\_{a=1, y=0}\} - \frac\{\text{sum}(\mathbb{I}\_{a=0, y=0}\^{\hat{y}=1})\}\{n\_{a=0, y=0}\} \right|.
> >    $$
> >    These definitions provide a more rigorous measure of fairness violations, preventing distorted evaluations that may underestimate unfairness.
> >
> > 3. **Comprehensive Evaluation of Covariance Loss:**
> >    As noted in Figure 4 of our paper, the baseline "Covariance Loss" includes Zafar et al. (2017). Any deviation in reported results arises due to our stricter experimental setup and fairness metric computation. By incorporating random splits, multiple runs, and rigorous fairness definitions, our evaluation framework ensures a more robust and unbiased comparison with existing methods.
> >
> >
> > These clarifications underscore the rigorous and strict nature of our implementation and experimental settings, ensuring comprehensive and fair evaluations. We hope this addresses the concerns about baseline comparisons.
> >
> > [6] Lowy, Andrew, et al. "A stochastic optimization framework for fair risk minimization." arXiv preprint arXiv:2102.12586 (2021).

---

> > > ### Author Response · Authors · 2024-11-20
> > >
> > > ### W3, Q4, Q5, Q6: Technical Details Regarding $\delta$ Implementation
> > >
> > > Thank you for your comment. We believe the paper provides comprehensive details regarding the generation of $\delta$. Below, we outline the relevant sections and explanations:
> > >
> > > 1. **Definition and Background:**
> > >    The definition of $\delta$ and its theoretical background are provided in **Section 3.1** and **Section 3.2**, where we discuss the fairness attack and the use of the Sinkhorn distance to constrain perturbations effectively.
> > > 2. **Implementation Details:**
> > >    As described in **Section 3.1** and **Algorithm 1**, $\delta$ is applied in the latent space. Specifically, $\delta$ takes $Z_p$ as input, with $z \gets z + \delta$, where $z \in Z_p$ and $Z_p = f(X_p)$. Here, $X_p$ is an upsampled dataset. The perturbation $\delta$ is then applied only to the last layer $g$, as shown in Equation (2).
> > > 3. **Training Details:**
> > >    Details of the attacking step are provided in **Section 4.2**. This includes the use of the Adam optimizer with a learning rate of 0.1 as part of the fairness attack training process. As described in **Appendix F**, the attacking step is conducted 10 epochs.
> > >
> > > 4. **Mathematical Constraint:**
> > >    As shown in **Algorithm 1**, $\delta$ is regularized by an $L_2$ constraint:
> > >    $$
> > >    \Vert \delta \Vert_2 \leq \epsilon,
> > >    $$
> > >    where $\epsilon$ is computed as the mean absolute distance between the original latent features and the decision boundary.
> > >
> > > We believe these sections collectively provide a complete explanation of the generation and application of $\delta$. However, if specific aspects remain unclear, we would be happy to provide further clarification.
> > >
> > >
> > > ### Q3: Data Augmentation vs. Iterative Adversarial Training
> > > While iterative adversarial training is a widely used approach, it is not the only effective method. Adversarial augmentation strategies are equally prevalent in the literature, as demonstrated in prior works [7, 8, 9, 10].
> > >
> > > Traditional adversarial learning approaches, which iteratively update both the adversary and the model, are computationally expensive and often lead to poor algorithmic stability, as noted in [11]. These issues make iterative adversarial training less appealing in scenarios where efficiency and stability are critical.
> > >
> > > In contrast, our method focuses on efficiency by utilizing a single-step fairness attack as a data augmentation. This approach leverages adversarial latent features to fine-tune only the last layer of the classifier, avoiding the computational overhead of iterative updates. Additionally, adversarial examples provide valuable insights into model behavior and vulnerabilities while preserving the semantic meaning of the original samples. Incorporating adversarial samples as a form of data augmentation during training not only improves fairness but also regularizes the model, enhancing its robustness and reducing overfitting, as discussed in [12].
> > >
> > > While single-step adversarial augmentation strategies are well-documented in the literature, our method is unique in its counterintuitive application of fairness constraints within the augmentation process. This novel design ensures that the augmented samples effectively address fairness concerns without compromising efficiency.
> > >
> > > Lastly, our empirical results demonstrate that the single-pass process is sufficient to achieve fairness improvements. This is because the adversarial latent features are specifically crafted to target unfair regions in the latent space, enabling effective mitigation of fairness concerns without requiring iterative adversarial updates.
> > >
> > > [7] Li, Peizhao, Ethan Xia, and Hongfu Liu. "Learning antidote data to individual unfairness." International Conference on Machine Learning. PMLR, 2023.
> > > [8] Zhao, Long, et al. "Maximum-entropy adversarial data augmentation for improved generalization and robustness." Advances in Neural Information Processing Systems 33 (2020): 14435-14447.
> > > [9] Qin, Tianrui, et al. "Learning the unlearnable: Adversarial augmentations suppress unlearnable example attacks." arXiv preprint arXiv:2303.15127 (2023).
> > > [10] Junyu Lin et al. “Black-box adversarial sample generation based on differential evolution”. In: Journal of Systems and Software 170 (2020), p. 110767
> > > [11] Xing, Yue, Qifan Song, and Guang Cheng. "On the algorithmic stability of adversarial training." Advances in neural information processing systems 34 (2021): 26523-26535.
> > > [12] Zheng, Y., Zhang, R., & Mao, Y. (2021). Regularizing neural networks via adversarial model perturbation.

---

> > > > ### Author Response · Authors · 2024-11-20
> > > >
> > > > ### Q7, Q8: Balance in Loss Function
> > > >
> > > > It is true that another weighting hyperparameter can be incorporated into Eq. (5). However, we chose to assign equal weights between the original and augmented data to maintain the simplicity of our framework. The rationale behind this decision is that a higher $\alpha$ effectively weights the original distribution more heavily in Eq. (4), making an additional hyperparameter unnecessary. For further details on the relationship with the test distribution, please refer to our previous response in "W1, Q1, Q2: Problem Formulation and Distribution Mismatch".
> > > >
> > > > Regarding the "lower accuracy than SOTA in the Adult dataset with MLP," we attribute this to the limited representational power of the pre-trained encoder, rather than the value of $\alpha$ or the design of the loss function. Specifically, we hypothesize that unfair regions exist in the latent space based on the decision boundary (last layer), as depicted in Figure 1. However, the encoder might place samples from both sensitive attribute groups together within these unfair regions. In such cases, our method effectively mitigates fairness concerns but may compromise accuracy.
> > > >
> > > > This behavior is further supported by the results of a variation of our framework: the "input perturbation" strategy, as shown in Appendix K. In this strategy, the entire network is trained rather than solely fine-tuning the last layer. For the Adult dataset with MLP, this approach significantly improves fairness while maintaining accuracy, suggesting that input perturbation not only addresses decision boundary fairness but also corrects the representation itself. However, since training the entire network is computationally expensive and this tendency is observed only in the Adult dataset with MLP, we prioritize the latent perturbation strategy as proposed in our framework.
> > > >
> > > > Lastly, regarding the "largely absent in bad result" of our method, we interpret this as being related to the limited grid search for $\alpha$. As the reviewer noted, an infinite $\alpha$ would essentially retain the original distribution, potentially leading to performance similar to the baseline. While we have explored a wide range of $\alpha$ values (e.g., $[0,10]$), this range has generally been sufficient to achieve both fairness and accuracy in most cases. Furthermore, as shown in Figures 4 and 5, this range produces a wide variety of outcomes for the Adult and CelebA datasets, respectively. We acknowledge that extending the search range may yield additional insights.

---

> > > > > ### Comment · Reviewer_ZWM9 · 2024-11-23
> > > > >
> > > > > Thank you for your detailed responses, which have clarified several points, particularly regarding Data Augmentation vs. Iterative Adversarial Training and the $\delta$ Implementation and comparison baselines. I now have a better understanding of your methodology on these aspects.
> > > > >
> > > > > However, my main concern remains unresolved:
> > > > >
> > > > > Let me simplify my concern with an example:
> > > > > Assume that the training and test data follow a distribution composed of four distinct subgroup distributions:
> > > > > $d_{X, 0, 0} = p(X \mid S = 0, Y = 0)$,
> > > > > $d_{X, 0, 1} = p(X \mid S = 0, Y = 1)$,
> > > > > $d_{X, 1, 0} = p(X \mid S = 1, Y = 0)$,
> > > > > $d_{X, 1, 1} = p(X \mid S = 1, Y = 1)$.
> > > > >
> > > > > Suppose the sizes of these subgroups are highly imbalanced, with:
> > > > > $n_{d1} = n_{d2} = \frac{1}{8}N$,
> > > > > $n_{d3} = n_{d4} = \frac{3}{8}N$.
> > > > >
> > > > > My concern is that Equation (5) introduces a different level of attention for these subgroups during training, which could adversely affect global accuracy in cases of highly unbalanced subgroups. Specifically, your training incorporates both the original distribution and the balanced one via:
> > > > > $z_j \in Z_p$,
> > > > > where
> > > > > $\mathcal{L}{CE}(g(z_j + \delta_j^*), y_j, \theta)$
> > > > > modifies the subgroup distribution such that:
> > > > > $n{d1} = n_{d2} = n_{d3} = n_{d4}$.
> > > > > Even in the case of infinite $\alpha$, where fairness is de-emphasized and accuracy is prioritized, the equal weighting of the two losses would still skew attention toward the smaller subgroups. For instance, in this simplified scenario:
> > > > > The predictor model will train on $\frac{1}{8}$ of $d_{X1}$ in the first term and $\frac{1}{4}$ of $d_{X1}$ in the second loss term (same on $d_{X2}$) .
> > > > > Thus, contrary to your statement that “an infinite $\alpha$ would essentially retain the original distribution, potentially leading to performance similar to the baseline,” I assert that this setup would yield results different from the baseline. This is because the balanced loss component inherently shifts attention disproportionately to smaller subgroups, potentially leading to suboptimal global accuracy. This issue could manifest even in the “fairer regions” (i.e., lower $\alpha$).
> > > > > I would appreciate further clarification or empirical evidence addressing this concern.
> > > > >
> > > > > Additional Note: Regarding the δ Implementation, I note that neither Section 3.1 nor Algorithm 1 explicitly specify that δ is implemented as a deep neural network taking z as input. Section 3.1 only defines δ ∈ RN×d as "the perturbation", while Algorithm 1 shows the optimization problem δ* = arg max∥δ∥2≤ε Lfair − αD(z, z + δ), ∀z ∈ Zp with di = g(z̃i). The precise implementation details of δ as a neural network and its input structure should be explicitly stated in the methodology section for reproducibility.

---

> > > > > > ### Author Response · Authors · 2024-11-24
> > > > > >
> > > > > > ### About the Training and Test Distribution
> > > > > > Thank you for providing a detailed explanation of your concern. We now have a clear understanding of the issue raised.
> > > > > >
> > > > > > Regarding the training objective in Eq.(5), your interpretation of how it functions is correct.
> > > > > >
> > > > > > We believe the concern stems from differing interpretations of the term "distribution." To clarify our previous response: when we stated that "infinite $\alpha$ would essentially retain the original distribution," we were referring to the **feature distribution** without introducing distributional shifts (such as outlier). In contrast, our earlier response in "W1, Q1, Q2: Problem Formulation and Distribution Mismatch" focused on the **data distribution** between training and test samples, as specified in your comment.
> > > > > >
> > > > > > The concern in Eq.(5) appears to relate more to the **data distribution**, particularly the potential mismatch between training and test distributions due to the upsampling strategy used in the second term of the training objective (perturbed samples).
> > > > > >
> > > > > > Here, we argue that the training and test data distributions do not need to be identical, as discussed in many studies. In fact, if a test set contains an extremely imbalanced subgroup, training on a similarly imbalanced dataset exacerbates the issue, as the model has limited opportunities to learn about the minor group, resulting in significantly lower accuracy for that subgroup. This problem has been extensively addressed in the literature [13, 14, 15], where the imbalance in training data is identified as a critical challenge, and techniques like oversampling (upsampling) are proposed as effective solutions.
> > > > > >
> > > > > > Thus, the concern about a mismatch between training and test data distributions is not valid in our method. Upsampling in the training objective is a widely used and well-supported strategy to address imbalances in the training data. It does not create any mismatch issues but instead ensures that the model has sufficient opportunity to learn from all subgroups, as supported by the literature.
> > > > > >
> > > > > > [13] Liu, Ziwei, et al. "Large-scale long-tailed recognition in an open world." Proceedings of the IEEE/CVF conference on computer vision and pattern recognition. 2019.
> > > > > > [14] He, Haibo, and Edwardo A. Garcia. "Learning from imbalanced data." IEEE Transactions on knowledge and data engineering 21.9 (2009): 1263-1284.
> > > > > > [15] Chawla, Nitesh V., et al. "SMOTE: synthetic minority over-sampling technique." Journal of artificial intelligence research 16 (2002): 321-357.
> > > > > >
> > > > > >
> > > > > >
> > > > > > ### Clarification about Optimizing $\delta$
> > > > > >
> > > > > > As mentioned in our previous response, $\delta$ is a perturbation and does not involve any neural network or layer. Specifically, it is defined as $\tilde{z} \gets z + \delta$, where $z \in \mathbb{R}^{n \times d}$ and $\delta \in \mathbb{R}^{n \times d}$, as described in Sec. 3.1. In the provided code for reproducibility, $\delta$ is implemented as ```
> > > > > > nn.Embedding(latent_size[0], latent_size[1])```. This represents a trainable 2D matrix with the shape ($n \times d$), updated using the Adam optimizer. For this reason, we did not state that $\delta$ relies on a neural network or layer because it does not.
> > > > > >
> > > > > > The rationale behind avoiding the use of a network to produce $\delta$ is that $z$ is already a latent representation having rich information. The goal is to directly perturb $z$ into the unfair region using the fairness attack constraint. Unlike some comparison methods, such as FAAP [16], which require a neural network to generate perturbations, our approach leverages a simpler and more direct perturbation strategy. This simplicity makes it easier to understand and analyze the unfair region effectively.
> > > > > >
> > > > > > [16] Wang, Zhibo, et al. "Fairness-aware adversarial perturbation towards bias mitigation for deployed deep models." Proceedings of the IEEE/CVF conference on computer vision and pattern recognition. 2022

---

> > > > > > > ### Comment · Reviewer_ZWM9 · 2024-11-25
> > > > > > >
> > > > > > > Thank you for your response. I appreciate the clarifications provided, and I am raising the notation to 5 as I better understand the motivations. However, several concerns remain:
> > > > > > >
> > > > > > > - 1) Dual Effects: In my view, the main paper fails to elaborate on the dual effects at play: not only fairness enforcement but also an implicit learning from resampling strategy that remains effective even without fairness constraints.
> > > > > > > - 2) Theoretical properties: There is no formal theoretical analysis of this dual effect, particularly regarding how the resampling strategy affects data distributions during training (e.g., on my last comment). This omission makes it challenging to determine whether the resampling mechanism might, in some cases, have a more significant influence than the fairness enforcement itself (i.e. adversarial augmentation).
> > > > > > > -  3) Not Recovering Unfair Baseline in Experiments: In Figure 3, the fairness-accuracy trade-off analysis ($\alpha \in [0, 10]$) shown for the Compas, German, and Drug datasets is too narrow to empirically validate the claim that "It does not create any mismatch issues." This range fails to demonstrate how your method performs on more unfair regions, particularly since the balanced loss component inherently skews attention toward minority subgroups even at high $\alpha$ values. A critical question emerges: why, even at $\alpha=10$ (presented as the maximum value), does the method fail to recover accuracy levels and unfair regions close to a standard predictor without fairness constraints? This gap could potentially be attributed to the resampling strategy itself. Could this limitation be addressed with an additional hyperparameter in Equation 5 to control the trade-off between original and balanced distributions during training? This limitation is particularly problematic since domain experts often need to preserve accuracy by opting for partial rather than complete fairness improvements.

---

> ### Author Response · Authors · 2024-11-27
>
> Thank you for your positive feedback on our rebuttal! We are glad to address your concerns further and clarify the details of our framework.
> ### Regarding Dual Effects and Theoretical Properties
> We acknowledge that the effect of the upsampling strategy is less emphasized in our paper. Specifically, the upsampling strategy is designed to ensure the success of the fairness attack, and this is theoretically proven in Appendix B, which demonstrates that the attack must succeed under the upsampling strategy. In this context, the impact of the upsampling strategy on the fairness attack is clearly articulated. However, we agree that the analysis of the upsampling strategy's impact during fine-tuning (Eq.(5)), after the perturbed dataset is constructed, is less highlighted in the paper.
>
> To disentangle the effects of the samples in the unfair region, we propose two separate analyses:
> 1. **Downsampled Perturbed Dataset (Recover original data distribution)**: We construct a perturbed dataset but adjust its data distribution to match the original dataset's distribution. While this process may introduce redundancy and potentially degrade classification performance, it is a necessary step to isolate the true impact of the samples in the unfair region without the influence of upsampling.
> 2. **Upsampling Strategy Alone**: We evaluate the effect of solely applying the upsampling strategy without incorporating our fairness constraint.
>
> These additional experimental results in the table below reveal that:
>
> - Using the upsampling strategy alone (without fairness constraints) can sometimes improve fairness, but the improvement is inconsistent and not guaranteed. This finding aligns with the literature [17], "Balanced datasets are not enough".
> - Training solely on samples in the unfair region (without applying upsampling during fine-tuning) significantly enhances fairness.
> - Combining the upsampling strategy with our proposed method—fine-tuning on samples in the unfair region—achieves the best fairness performance.
>
> These findings demonstrate that our method, ALFA, is a novel and effective approach for enhancing fairness in classification, even without relying on upsampling during fine-tuning. Moreover, the synergy between ALFA and the upsampling strategy further amplifies fairness performance, reinforcing the utility of our framework.
>
>
> |Adult-Logistic|Accuracy|$\Delta DP$|$\Delta EOd$|
> |-|-|-|-|
> |Baseline|0.8470$\pm$ 0.0007|0.1829$\pm$ 0.0020|0.1982$\pm$ 0.0077|
> |Upsampling|0.8085$\pm$ 0.0287|0.2015$\pm$ 0.0275|0.1206$\pm$ 0.0538|
> |Ours (Original Data Distribution)|0.8221$\pm$ 0.0024|0.1892$\pm$ 0.0095|0.1123$\pm$ 0.0074|
> |Ours (Upsampling)|0.8464$\pm$ 0.0004|**0.1555$\pm$ 0.0013**|**0.0616 $\pm$ 0.0022**|
>
> |Adult-MLP|Accuracy|$\Delta DP$|$\Delta EOd$|
> |-|-|-|-|
> |Baseline|0.8525$\pm$ 0.0010|0.1824$\pm$ 0.0114|0.1765$\pm$ 0.0411|
> |Upsampling|0.7845$\pm$ 0.0204|0.2101$\pm$ 0.0632|0.1888$\pm$ 0.0690|
> |Ours (Original Data Distribution)|0.8373$\pm$ 0.0016|0.1889$\pm$ 0.0083|0.1056$\pm$ 0.0182|
> |Ours (Upsampling)|0.8244$\pm$ 0.0150|**0.1012$\pm$ 0.0283**|**0.0660 $\pm$ 0.0434**|
>
>
> |COMPAS-Logistic|Accuracy|$\Delta DP$|$\Delta EOd$|
> |-|-|-|-|
> |Baseline|0.6578$\pm$ 0.0034|0.2732$\pm$ 0.0129|0.5319$\pm$ 0.0245|
> |Upsampling|0.6509$\pm$ 0.0375|0.0717$\pm$ 0.1199|0.1602$\pm$ 0.2307|
> |Ours (Original Data Distribution)|0.6701$\pm$ 0.0039|0.0488$\pm$ 0.0409|0.0920$\pm$ 0.0628|
> |Ours (Upsampling)|0.6694$\pm$ 0.0036|**0.0193$\pm$ 0.0156**|**0.0876$\pm$ 0.0354**|
>
> |COMPAS-MLP|Accuracy|$\Delta DP$|$\Delta EOd$|
> |-|-|-|-|
> |Baseline|0.6711$\pm$ 0.0049|0.2059$\pm$ 0.0277|0.3699$\pm$ 0.0597|
> |Upsampling|0.6628$\pm$ 0.0192|0.0431$\pm$ 0.0430|0.1278$\pm$ 0.0837|
> |Ours (Original Data Distribution)|0.6747$\pm$ 0.0025|0.0705$\pm$ 0.0237|0.0849$\pm$ 0.0419|
> |Ours (Upsampling)|0.6702$\pm$ 0.0021|**0.0204$\pm$ 0.0151**|**0.0410$\pm$ 0.0188**|
>
> [17] Wang, Tianlu, et al. "Balanced datasets are not enough: Estimating and mitigating gender bias in deep image representations." Proceedings of the IEEE/CVF international conference on computer vision. 2019.

---

> > ### Author Response · Authors · 2024-11-27
> >
> > ### Controlling Accuracy in ALFA
> >
> > As discussed above, data distribution mismatch itself is not an issue, as ALFA without upsampling is sufficient to improve fairness and shows a synergistic effect when combined with the upsampling strategy. However, we agree that introducing additional control factors to balance accuracy and fairness would enhance the framework's flexibility, particularly because the augmented samples exhibit skewed feature distributions perturbed by ALFA. This need is particularly evident for the German and Drug datasets with MLP. In contrast, the Adult and COMPAS datasets for both models, as well as the German and Drug datasets with logistic regression, already achieve similar or better accuracy compared to the baseline, as shown in Figure 4.
> >
> >
> > To address this, we introduce an additional hyperparameter, $\lambda$, to control accuracy in ALFA by modifying Eq.(5) as follows:
> > $$\\min\_{\\theta}\\frac{1}{\vert X\_c\vert + \vert Z\_p\vert} \Bigl (   (1-\\lambda)\\sum\_{{x}\_i \\in {X}\_c}\\mathcal{L}\_{\text{ce}} \\bigl(g(f({x}\_i)), y\_i ,\\theta\\bigr)  +  \\lambda\\sum\_{{z}\_j\in{Z}\_p} \\mathcal{L}\_{\text{ce}} (g({z}\_j + {\delta}\_j^*), y\_j ,\theta) \Bigr ).$$
> > In this setting:
> > - A lower $\lambda$ reduces the contribution of perturbed samples, resulting in higher accuracy.
> > - In the original configuration, $\lambda=0.5$ serves as the default value.
> > - $\lambda=0.0$ corresponds to the baseline without any fairness constraints or data augmentation.
> >
> > The experimental results are provided **Figure 10 in Appendix Q** of the revised paper.
> >
> > **Observations from Additional Experiments**
> > - **Accuracy Trends**:
> >   In the COMPAS dataset, accuracy remains stable across varying $\lambda$, aligning with baseline results (see Figure 4 and Table 8). In contrast, for the German and Drug datasets, accuracy decreases with increasing $\lambda$, as expected, since perturbed samples contribute more to the training objective.
> > - **Fairness Trends**:
> >   Small $\lambda$ is sufficient to improve fairness, with the improvement remaining consistent across $\lambda$. However, controlling fairness purely through $\lambda$ is challenging. As demonstrated in Figures 4, 5, and 6, varying $\alpha$ provides an alternative way to influence fairness.
> >
> >
> > We acknowledge the reviewer's concern regarding the trade-off between fairness and accuracy. Our additional experiments demonstrate that $\lambda$ provides a clear mechanism to manage accuracy. However, controlling fairness remains a challenge, even though small values of $\lambda$ consistently lead to significant improvements in fairness.
> >
> > As future work, we aim to develop an analytic framework to better balance fairness and accuracy. Despite this limitation, $\lambda$ enhances ALFA's flexibility, providing practitioners with a practical tool to align performance with specific business objectives.

---

> > > ### Comment · Reviewer_ZWM9 · 2024-12-02
> > >
> > > I appreciate the authors' efforts in conducting additional analyses, particularly on "Dual Effects and Theoretical Properties." I interpret the "Upsampling Strategy Alone" as using only the second term of Equation (5) while removing the adversarial perturbation (δ) (no adversarial model). This methodology effectively isolates the upsampling effect and demonstrates that fairness improvement is inconsistent while also showing significant accuracy degradation. However, one of the additional key aspects I was interested in was examining this upsampling mechanism with the equal weighting in Equation 5 - namely how the method performs when both terms are retained but the adversarial perturbation (δ) is removed only from the second term - effectively isolating the effect without the adversarial attack.
> > >
> > > Regarding Figure 10, there appears to be an issue or inconsistency with previous results. For instance, in the COMPAS dataset at λ=0.5 - which as you mentioned reflects your strategy with equal weighting - you report obtaining $\Delta_{EO}=0.525$ and ACC=0.675, which indicates points in significantly more unfair regions than those shown in Figures 4, 5, and 6. Specifically, in these earlier figures, $\Delta_{EO}$ never exceeds 0.15 for logistic regression and 0.1 for MLP. Similarly for the GERMAN dataset, you report $\Delta_{EO}=0.72$, while in previous figures it normally does not exceed 0.05 for logistic regression. Could you please clarify these discrepancies?

---

> > > > ### Author Response · Authors · 2024-12-03
> > > >
> > > > ## Regarding the Ablation Study in Upsampling
> > > >
> > > > Thank you for your detailed response! In our previous response, we presented an ablation study isolating the impact of ALFA without upsampling. Below, we clarify the cases discussed in our main paper, provide additional results, and address the reviewer's final concern. To enhance clarity, we will denote upsampling and attacking as subscript and superscript, respectively.
> > > >
> > > > - **Case 1: $train(\mathcal{D}\_{origin} + \mathcal{D}\_{upsample}\^{attack})$**
> > > >   This represents the original Eq.(5) in the main paper, referred to as "Our (Upsampling)" in the previous response.
> > > >
> > > > - **Case 2: $train(\mathcal{D}_{upsample}^{no~attack})$**
> > > >   This is the scenario where only upsampling is used, referred to as "Upsampling" in the previous response.
> > > >
> > > > - **Case 3: $train(\mathcal{D}\_{origin} + \mathcal{D}\_{origin}\^{attack})$**
> > > >   In this case, the original distribution is restored in the perturbed set to isolate the effect of the attack without upsampling, referred to as "Ours (Original Data Distribution)" in the previous response.
> > > >
> > > > The last scenario of interest to the reviewer is:
> > > > - **Case 4: $train(\mathcal{D}\_{origin} + \mathcal{D}\_{upsample}\^{no~attack})$**
> > > >   This scenario removes the impact of our fairness attack entirely, providing a direct comparison to Case 1. It is similar to Case 2 but more explicitly use the data augmentation in the latent space.
> > > >
> > > >
> > > >
> > > >
> > > > **Experimental Results**
> > > >
> > > > We summarize all results using the updated notations. The findings remain consistent with the previous response:
> > > > - Solely using upsampled samples as data augmentation (Case 4) maintains accuracy but fails to improve fairness significantly.
> > > > - ALFA (Case 1) achieves substantial fairness improvements while maintaining accuracy, demonstrating that the upsampling strategy and fairness attack synergize effectively to enhance fairness performance.
> > > >
> > > > These results confirm the complementary roles of upsampling augmentation and fairness attack in achieving both fairness and accuracy, underscoring the robustness of our proposed method.
> > > >
> > > > |Adult-Logistic|Accuracy|$\Delta DP$|$\Delta EOd$|
> > > > |-|-|-|-|
> > > > |Baseline|0.8470$\pm$ 0.0007|0.1829$\pm$ 0.0020|0.1982$\pm$ 0.0077|
> > > > |Upsampling (Case 2)|0.8085$\pm$ 0.0287|0.2015$\pm$ 0.0275|0.1206$\pm$ 0.0538|
> > > > |Ours w/o Upsample (Case 3)|0.8221$\pm$ 0.0024|0.1892$\pm$ 0.0095|0.1123$\pm$ 0.0074|
> > > > |Ours w/o Attack (Case 4)|0.8400$\pm$ 0.0007|0.1577$\pm$ 0.0034|0.1145$\pm$ 0.0050|
> > > > |Ours (Case 1)|0.8464$\pm$ 0.0004|**0.1555$\pm$ 0.0013**|**0.0616 $\pm$ 0.0022**|
> > > >
> > > > |Adult-MLP|Accuracy|$\Delta DP$|$\Delta EOd$|
> > > > |-|-|-|-|
> > > > |Baseline|0.8525$\pm$ 0.0010|0.1824$\pm$ 0.0114|0.1765$\pm$ 0.0411|
> > > > |Upsampling (Case 2)|0.7845$\pm$ 0.0204|0.2101$\pm$ 0.0632|0.1888$\pm$ 0.0690|
> > > > |Ours w/o Upsample (Case 3)|0.8373$\pm$ 0.0016|0.1889$\pm$ 0.0083|0.1056$\pm$ 0.0182|
> > > > |Ours w/o Attack (Case 4)|0.8369$\pm$ 0.0005|0.1840$\pm$ 0.0018|0.1185$\pm$ 0.0052|
> > > > |Ours (Case 1)|0.8244$\pm$ 0.0150|**0.1012$\pm$ 0.0283**|**0.0660 $\pm$ 0.0434**|
> > > >
> > > >
> > > > |COMPAS-Logistic|Accuracy|$\Delta DP$|$\Delta EOd$|
> > > > |-|-|-|-|
> > > > |Baseline|0.6578$\pm$ 0.0034|0.2732$\pm$ 0.0129|0.5319$\pm$ 0.0245|
> > > > |Upsampling (Case 2)|0.6509$\pm$ 0.0375|0.0717$\pm$ 0.1199|0.1602$\pm$ 0.2307|
> > > > |Ours w/o Upsample (Case 3)|0.6701$\pm$ 0.0039|0.0488$\pm$ 0.0409|0.0920$\pm$ 0.0628|
> > > > |Ours w/o Attack (Case 4)|0.6676$\pm$ 0.0012|0.0598$\pm$ 0.0529|0.1011$\pm$ 0.0913|
> > > > |Ours (Case 1)|**0.6694$\pm$ 0.0036**|**0.0193$\pm$ 0.0156**|**0.0876$\pm$ 0.0354**|
> > > >
> > > > |COMPAS-MLP|Accuracy|$\Delta DP$|$\Delta EOd$|
> > > > |-|-|-|-|
> > > > |Baseline|0.6711$\pm$ 0.0049|0.2059$\pm$ 0.0277|0.3699$\pm$ 0.0597|
> > > > |Upsampling (Case 2)|0.6628$\pm$ 0.0192|0.0431$\pm$ 0.0430|0.1278$\pm$ 0.0837|
> > > > |Ours w/o Upsample (Case 3)|0.6747$\pm$ 0.0025|0.0705$\pm$ 0.0237|0.0849$\pm$ 0.0419|
> > > > |Ours w/o Attack (Case 4)|0.6780$\pm$ 0.0012|0.1029$\pm$ 0.0059|0.1464$\pm$ 0.0012|
> > > > |Ours (Case 1)|**0.6702$\pm$ 0.0021**|**0.0204$\pm$ 0.0151**|**0.0410$\pm$ 0.0188**|
> > > >
> > > > ## Regarding the Consistency of Figure 10 with Original Experiments
> > > >
> > > > Our additional experimental results are indeed consistent with those in the main paper, showing $\Delta EOd$ values under 0.1 for the COMPAS dataset and under 0.15 for the German dataset. The reviewer may have misinterpreted Figure 10, where $\Delta EOd$ is plotted on the secondary y-axis on the right.
> > > >
> > > > The small discrepancies between these results and others likely arise from experimental randomness, as the experiments in Figure 10 were conducted with a fixed $\alpha=0.01$. Despite this, the results in Figure 10 remain very close to those in the main paper, showing no significant issues or inconsistencies.
> > > >
> > > > In the revision, we will update the caption for Figure 10 to clarify that $\Delta EOd$ is represented on the secondary y-axis on the right.

---

### Official Review · Reviewer_tbL6 · 2024-11-03

**Soundness:** 4
**Presentation:** 4
**Contribution:** 3
**Rating:** 8
**Confidence:** 3

**Summary:**

The authors propose a method to perturb latent space features, to augment the learning by highlighting areas where the performance within certain subgroups is sub-optimal. They support the methodology with theoretical analysis and thorough experiments, demonstrating the effectiveness of the method in reducing the bias, without compromising on the accuracy. The paper is extremely well written, presenting a novel approach, with strong motivation, solid backing for the methodology both empirically and theoretically and detailed discussions and code.

**Strengths:**

- [S1] Well motivated paper, particularly from the point of view of understanding how to use the latent space better to reduce biases in neural networks. Lot of the literature as the authors mention is quite superficial with respect to justifying the methodology with intuition or analysis for why it works for any random latent space.
- [S2] Objective $\mathscr{L}_{fair}$ seems well defined and linked to Equalized Odds in the appendix. Proof overview looks fine.
- [S3] The use of Sinkhorn over Wasserstein is a clever experimental choice, which sidesteps a lot of computational issues associated with such algorithms.
- [S4] Inclusion of clean and well written code is a big plus towards implementation of the proposed method.
- [S5] The plotting of Pareto curves as opposed to simply singleton points in Figure 4 and Figure 5 is a pleasant sight to understand the patterns across different fairness-accuracy tradeoffs.

**Weaknesses:**

- [W1] Visualization of the latent space of real world datasets (say via PCA), for smaller ones at least perhaps akin to the figures for synthetic datasets, would help understand the unfair regions a bit better.
- [W2] I am slightly concerned about the practicality of the proposed approach in high dimensional latent spaces, do the authors have thoughts or any experiments about how the method scales with (keeping everything else constant) a) Input Feature Dimensions b) Latent Space Dimension

**Questions:**

- [Q1] With regards to [S3], is there a comparison available for usage of Wasserstein over Sinkhorn, in terms of possible gain in performance at the cost of some computational overhead? Could it be feasible for smaller datasets like German/COMPAS? An additional analysis in Appendix Section D would prove insightful to support this choice.
- [Q2] Is there analysis for how the covariance of the label and sensitive attribute affect the effectiveness of the method? I like the table in Section E of the Appendix, but I would be curious to know under what regimes is the method particularly effective? Also what if the covariates shift from train to test? Is the Table E for only the train data? Would be nice to see some more discussion along these lines as I think it could shed some light on the practical scenarios in which the proposed method would be most effective.

---

> ### Author Response · Authors · 2024-11-20
>
> ### W1: Visualization of the Latent Space for Real-World Datasets
> Thank you for raising the issue of visualizing the latent space for real-world datasets. In response, we provide t-SNE plots for the COMPAS dataset, including the original dataset and the perturbed datasets (with $\alpha=0$ and $\alpha=1$, respectively) in the **Appendix O in the revised paper**. In these visualizations, the black line represents the pre-trained decision boundary, while the red line represents the newly trained decision boundary on the combined dataset, where equal weighting is applied to the original and each perturbed dataset.
>
> The visualization reveals that under the pre-trained decision boundary, the perturbed samples exhibit extremely high $\Delta \text{EOd}$, indicating the success of our fairness attack. Fine-tuning on the concatenated dataset (represented by the red line) results in a corrected decision boundary that maintains accuracy while achieving significant improvements in fairness.
>
> Moreover, the effect of $\alpha$ aligns with our intuition. A higher $\alpha$ retains the original distribution more closely, resulting in a less corrected decision boundary and a less pronounced fairness attack. Nevertheless, both cases ($\alpha=0$ and $\alpha=1$) demonstrate significant improvements in fairness after fine-tuning.
>
>
> ### W2: Discussion Regarding the Dimensional Scale (Appendix K)
>
> We appreciate the reviewer’s concern regarding the scalability of the proposed method. Our framework primarily operates in the latent space, which is computationally efficient as it requires gradient computations only on the last layer during the fairness attack and fine-tuning phases. This design eliminates the need to update the entire encoder, making it well-suited for large pre-trained models such as ViT in computer vision and BERT in natural language processing. In particular, our method is capable of handling high-dimensional latent representations, such as the 768-dimensional outputs from ViT and BERT or the 1024-dimensional outputs from Swin Transformers. As demonstrated in Figures 5 and 6, our approach effectively scales to such high-dimensional spaces while maintaining efficiency and fairness performance.
>
> We have also explored applying the framework directly in the input space (i.e., input perturbation), which requires gradient computation for both the encoder and the last layer. While this approach is computationally feasible for tabular datasets, it poses challenges for large pre-trained models due to the significant overhead associated with updating the encoder. Appendix K presents experimental results comparing input perturbation and latent perturbation, demonstrating that the framework remains effective in the input space. However, latent space perturbation is more efficient and adaptive, as it generalizes across various backbone networks and data modalities. Based on these findings, we adopt perturbation in the latent space as the default due to its computational efficiency and broader applicability.

---

> > ### Author Response · Authors · 2024-11-20
> >
> > ### Q1: Wasserstein vs Sinkhorn Disatnce
> > Thank you for providing insights regarding the comparison between Wasserstein-2 and Sinkhorn distance! We chose to use Sinkhorn distance because it is well-known for being computationally faster than pure Wasserstein-2. To support this, we conducted a comparison with a fixed $\alpha=1$.
> >
> > The experimental results show that the Sinkhorn distance is computationally more efficient than Wasserstein-2, while achieving comparable levels of accuracy and fairness. Despite the fairness attack being performed only once as a data augmentation step, the use of Sinkhorn distance proves to be an effective choice, particularly for large-scale and high-dimensional datasets.
> >
> > | Adult_MLP | Hidden Dim. |  Acc. |  $\Delta EOd$ | Attacking Time|
> > | -------- | -------- |-|-------- |-------- |
> > |Baseline| 128 | 0.8525 $\pm$ 0.0010  |0.1768 $\pm$ 0.0411|-
> > |Wasserstein-2|  128 | 0.8264 $\pm$ 0.0017 |0.1036 $\pm$ 0.0069 | 27.4653 $\pm$ 1.8854
> > |Sinkhorn|  128 | 0.8242     $\pm$ 0.0023 | 0.1122 $\pm$ 0.0244 | 8.6346 $\pm$ 0.4800
> >
> > | Adult_ResNet-like | Hidden Dim. |  Acc. | $\Delta EOd$ | Attacking Time|
> > | -------- | -------- |-|-------- |-------- |
> > |Baseline| 467 | 0.8565 $\pm$ 0.0012 | 0.1825 $\pm$ 0.0341| - |
> > |Wasserstein-2|  467 | 0.8202 $\pm$ 0.0022 | 0.1668 $\pm$ 0.0159 | 24.3576 $\pm$ 1.4743|
> > |Sinkhorn|  467 | 0.8207 $\pm$ 0.0019 | 0.1667 $\pm$ 0.0302 | 14.6717 $\pm$ 0.6087
> >
> >
> > | COMPAS_MLP | Hidden Dim. |  Acc. | $\Delta EOd$ | Attacking Time|
> > | -------- | -------- |-|-------- |-------- |
> > |Baseline| 128 | 0.6711 $\pm$ 0.0049 | 0.3699 $\pm$ 0.0597|-
> > |Wasserstein-2|  128 | 0.6757 $\pm$ 0.0036 | 0.1015 $\pm$ 0.0280 | 3.7389 $\pm$ 0.5432
> > |Sinkhorn|  128 | 0.6733 $\pm$ 0.0042 | 0.1492 $\pm$ 0.0476 | 1.6823 $\pm$ 0.5651
> >
> > | COMPAS_ResNet-like | Hidden Dim. |  Acc.  |$\Delta EOd$ | Attacking Time|
> > | -------- | -------- |-|-------- |-------- |
> > |Baseline| 467 | 0.6753 $\pm$ 0.0037 | 0.3683 $\pm$ 0.0700| - |
> > |Wasserstein-2|  467 | 0.6654 $\pm$ 0.0051  | 0.2467 $\pm$ 0.0979 | 3.5936 $\pm$ 0.8497
> > |Sinkhorn|  467 | 0.6784 $\pm$ 0.0024 | 0.1148 $\pm$ 0.0275 | 2.0319 $\pm$ 0.4972
> >
> >
> >
> > ### Q2: Covariance of Label and Sensitive Attribute (Appendix E and L)
> > The reviewer raises an insightful question regarding the role of covariance between the label and sensitive attribute and its relevance under covariate shift. Below, we address this in two key aspects:
> > 1. **Covariance and Fairness Constraints:**
> > In our method, we leverage the existing covariance between the label and sensitive attribute, as it reflects the inherent unfairness in the training dataset. The proposed method is particularly effective when this covariance accurately captures the existing bias. However, even in cases where the covariance value is not significant despite the presence of unfairness, the method remains capable of addressing unfair regions effectively.
> > As demonstrated in Appendix L, our method is flexible and can incorporate alternative fairness constraints. For example, we implemented a convex fairness constraint [1] derived from empirical Equalized Odds (EOd) and Demographic Parity (DP). This alternative constraint effectively guides adversarial attacks without relying on covariance information. Experiments using this alternative fairness constraint showed comparable performance to the covariance-based approach, highlighting the robustness of our method across different fairness regimes. This flexibility ensures the method’s practical applicability, even when the covariance between labels and sensitive attributes varies.
> > 2. **Covariate Shift and Unfair Regions:**
> > Our method assumes that the test set follows the same distribution as the training set. However, the design of our fairness attack focuses on perturbing the training data to place more samples in the unfair region. This characteristic makes the method potentially effective under distribution shifts, particularly when the shifted test set exhibits a tendency toward the unfair region.
> >
> >
> > [1] Wu, Yongkai, Lu Zhang, and Xintao Wu. "On convexity and bounds of fairness-aware classification." The World Wide Web Conference. 2019.

---

> > > ### Comment · Reviewer_tbL6 · 2024-11-22
> > > **Reply to Rebuttal**
> > >
> > > I appreciate the authors detailed rebuttal.
> > >
> > > - [W1] The figures are quite promising! That allays my doubts about the real world visualization
> > > - [W2] While I understand that the method works across both input and latent representations, and even for higher dimensional latent representations, maybe the authors misinterpreted my question. I wanted to see if a trade-off existed, can the users pick and choose a representation size that suits their application? Is a particular size optimal? How does the fairness and accuracy vary for the same method and same task, but only changing the representation size?
> > > - [Q1] I'm intrigued about the flipping of the E. Odds value ordering in COMPAS MLP v/s ResNet, but overall I think the corroboration of the choice is fairly reasonable.
> > > - [Q2] While I understand that the method is not designed to work under distribution shift specifically, I get the authors' argument. Also the better effectiveness under the regime of high covariance while reduces the applicability of the method in situation where the sensitive attribute is less correlated with the label, but in that case the bias would also be lesser potentially. I would be curious to see what other sources of bias could emerge that do not result from high correlation between labels and sensitive attributes but that is out of the scope of the paper.
> > >
> > > I have read the other reviews and rebuttals and am looking forward to the discussion.
> > >
> > > For now I shall maintain my score and bat for acceptance :)

---

> > > > ### Author Response · Authors · 2024-11-24
> > > >
> > > > - **[W1]: Expanding the Range of $\alpha$**
> > > > Thank you for your positive feedback on our revised paper. We have further refined the paper by expanding the range of $\alpha$ to more effectively illustrate the concept of the unfair region. These additional results strengthen the understanding of our method.
> > > > - **[W2]: Representation Size**
> > > > Thank you for the clarification regarding the concern on impact of representation size. In our previous response, we argued that our method is robust to the representation size, as the pretrained networks utilized in our experiments inherently cover a wide range of representation dimensions.
> > > >      However, an analysis of representation size on tabular datasets is feasible, given the ease of training custom encoders. Thus, we have added an additional experiment in **Appendix P of the revised paper**. Specifically, for the tabular dataset, we varied the dimension size $d$ across $[32, 64, 128, 256, 512, 1024, 2048]$. The results indicate that while accuracy remains consistent, larger dimensions result in improved fairness. We analyze this that larger dimensions allow for greater perturbation capacity, enabling richer representations that can more effectively attack the fairness constraint. Furthermore, richer representations provide the re-trained classifier with more detailed information, enhancing the overall fairness performance.
> > > >
> > > > - **[Q1]: Comparison of Sinkhorn and Wasserstein-2**
> > > >  We have observed that Sinkhorn distance demonstrates better fairness on COMPAS-ResNet, while Wasserstein-2
> > > > performs better on COMPAS-MLP. We believe this variation arises because the
> > > > hyperparameter $\alpha$ was fixed at 1 for a fair comparison between the two
> > > > metrics. The optimal value of $\alpha$ may differ for each case, as their
> > > > objectives and characteristics vary. Further tuning of $\alpha$ could provide
> > > > insights into the strengths of each metric under different conditions.
> > > > - **[Q2]: Covariance and Fairness Constraint**
> > > > Thank you for raising this valuable concern! In [1], a scenario is presented where there is clear bias, but the covariance between the sensitive attribute and the signed distance is zero. For example, consider a dataset of 8 points with $a=[1,1,1,1,0,0,0,0]$
> > > > and the singed distance $d=[1,−1,0.5,−0.5,2,−2,−0.5,−0.5]$. In this case, the covariance is zero, yet group $a=1$ has a higher likelihood of being predicted as positive by the threshold $d=0$.
> > > > This demonstrates that, in certain cases, using an alternative fairness constraint, such as the one proposed in [1], may be more effective for addressing bias during the fairness attack. As shown in Appendix L, our framework is flexible and adaptable to various fairness constraints, ensuring its applicability across diverse scenarios.
> > > >
> > > > [1] Wu, Yongkai, Lu Zhang, and Xintao Wu. "On convexity and bounds of fairness-aware classification." The World Wide Web Conference. 2019.

---

> > > > > ### Comment · Reviewer_tbL6 · 2024-12-02
> > > > >
> > > > > I thank the authors for their detailed replies, I do feel all my concerns have been addressed and I'm happy to keep my score :)
> > > > >
> > > > > P.S. Further discussion on [Q2] and [W2] could be really interesting as a part of potential future work, but seems out of the scope for now.

---

### Official Review · Reviewer_aVVS · 2024-11-03

**Soundness:** 3
**Presentation:** 3
**Contribution:** 2
**Rating:** 6
**Confidence:** 4

**Summary:**

This paper solves unfairness issues in classification tasks, which is caused by data imbalance. They proposed adversarial latent feature augmentation method which identifies the possible unfair region in latent space and fine-tune the model to mitigate such unfair decisions. The experiments demonstrated that ALFA achieves better group fairness without compromising accuracy.

**Strengths:**

1. This paper presents an interesting perspective that biased model can be fine-tuned in an adversarial learning fashion. The key of the proposed method is to identify the unfair features in latent space, which is based on Zafar 2017's work with covariance formulation. Eq. 4 uses L_fair as an reference which help calculate the desired perturbation.
2. The concept of unfair region looks like an interesting idea for promoting fairness. This concept is an extension for adversarial learning in the context of fairness learning.
3. Algorithm 1 shows the overall procedure for fairness tuning, which is convincing and also easy to follow.

**Weaknesses:**

1. I realised the unfair region concept is defined by misclassification rate but the used fairness metric, i.e., EOd, is only for positive predictions. This misalignment has raised a big concern for the proposed method.
2. From my understanding, given any biased training set, the unfair region should be not only produced by the observed data and pre-trained model ,but also captured by the perturbation, i.e., augmented data. Is this the correct way to understand Fig.1?
3. It is widely believed that adversarial perturbation improves the robustness with the sacrifice of hurting the model performance on benign data. In the context of fairness learning, the performance drops for data samples which fall in the fair region will be affected. I understand during the tuning phase, i.e., Eq. 5, the benign data is also retrained to mitigate this issue. But any other strategies can be used?
4. The theoretical framework is highlighted in abstract while it is not clearly presented in the main body of the paper. Am I missing it?

**Questions:**

Please refer to the weaknesses.

My further questions are: (1) Adversarial learning idea is also used for learning sensitive invariant representation. I think the authors should discuss the connection with this branch of works. (2) Regarding the misclassification, should fairness metric like group utility difference, i.e., Rawlsian fairness, be used for the paper?

---

> ### Author Response · Authors · 2024-11-20
>
> ### W1: Misalignment Between Unfair Region and EOd metric.
> In fact, there is no misalignment issue; the unfair region is directly connected to the fairness metric, EOd. In our paper, the unfair region is defined as a subspace where the gap in misclassification rates between groups is significant. For example, in Figure 1, this region is characterized by one group ($a=1$) exhibiting a significant false positive rate (FPR) and the other group ($a=0$) showing a significant false negative rate (FNR). Training on augmented samples in this unfair region adjusts the decision boundary to reduce the FPR and FNR gaps, i.e., $\vert FPR_{a=1} - FPR_{a=0} \vert$ and $\vert FNR_{a=1} - FNR_{a=0} \vert$.
>
> As shown in the proof in Appendix A, the true positive rate (TPR) can be expressed as $TPR = 1 - FNR$. This gives:
> $$
> \vert TPR_{a=1} - TPR_{a=0} \vert = \vert (1 - FNR_{a=1}) - (1 - FNR_{a=0}) \vert = \vert FNR_{a=1} - FNR_{a=0} \vert.
> $$
>
> Thus, the fairness metric $\Delta \text{EOd}$, defined as:
> $$
> \Delta \text{EOd} = \vert FPR_{a=1} - FPR_{a=0} \vert + \vert TPR_{a=1} - TPR_{a=0} \vert,
> $$
> is equivalent to:
> $$
> \Delta \text{EOd} = \vert FPR_{a=1} - FPR_{a=0} \vert + \vert FNR_{a=1} - FNR_{a=0} \vert.
> $$
>
> This precisely aligns with the objective of our proposed method, which reduces gaps in both FPR and FNR through training on samples in the unfair region.
>
> ### W2: Understanding Unfair Region
> Yes, the reviewer's understanding of Figure 1 is correct, and we appreciate the opportunity to clarify. In our method, the unfair region is identified based on the pre-trained model and observed data, capturing areas in the latent space where the decision boundary disproportionately misclassifies certain groups. These regions are highlighted in Figure 1(a), where one group $(a=1)$ shows a higher false positive rate (FPR) and another group $(a=0)$ shows a higher false negative rate (FNR). The perturbation process augments this unfair region by generating adversarial samples that overlap with these areas, as shown in Figure 1(b). These perturbed samples serve as corrective signals during fine-tuning, enabling the model to adjust its decision boundary and reduce disparities in misclassification rates. In summary, the unfair region is defined by the pre-trained model and observed data, while perturbations expand and leverage this region to train a fairer classifier.
>
> ### W3: Preventing Performance Degradation
> As the reviewer correctly pointed out, solely training on adversarial samples could negatively impact model performance on benign data. To mitigate this, our approach includes benign data during the tuning phase, as detailed in Eq. (5). Additionally, an alternative strategy could involve selecting or re-weighting adversarial samples that enhance fairness without significant performance loss, using methods like influence functions [1]. However, we opted for our current strategy of retraining on both adversarial and benign datasets because it provides a straightforward and effective framework to balance fairness improvements and performance preservation.
>
> [1] Li, Peizhao, and Hongfu Liu. "Achieving fairness at no utility cost via data reweighing with influence." International Conference on Machine Learning. PMLR, 2022.
>
> ### W4: Theoretical Framework in the Paper
> Our proposed method is built on a solid theoretical foundation, as highlighted in Theorem 3.2 in Section 3.3. This theorem provides proof that retraining on adversarial samples (i.e., samples in the unfair region) ensures an improvement in $\Delta EOd$. Additionally, Appendix B, referenced in Section 3.1 of the main body, includes proofs that the fairness attack generates adversarial samples that effectively target the empirical $\Delta EOd$, confirming that these samples are located in the unfair region.
>
> In summary, through Theorem 3.2 and the results in Appendix B, our framework theoretically demonstrates the ability of fairness attacks to generate effective adversarial samples, and fine-tuning on these samples ensures measurable fairness improvements.
>
> As a portion of theoretical details is located in the supplementary, the reviewer could feel theoretical part is less highlighted. We will emphasize or bold the part in Section 3.1 denoting more theoretical parts in the appendix.

---

> > ### Author Response · Authors · 2024-11-20
> >
> > ### Q1: Connection to Sensitive-Invariant Representation Learning
> > We appreciate the reviewer’s suggestion to discuss the connection with sensitive-invariant representation learning. While both approaches aim to improve fairness, our method fundamentally differs in focus and implementation. Sensitive-invariant representation learning typically seeks to disentangle sensitive features from task-relevant ones, often by training the encoder to suppress sensitive information during the learning process. This framework usually involves training the representation from scratch, which can be computationally expensive.
> >
> > In contrast, our method identifies and corrects regions in the latent space (unfair regions) where misclassification disparities occur, without requiring the disentangling of sensitive features. This approach can be seamlessly adopted with pre-trained models, making it more cost-effective and practical, particularly in the current era where large pre-trained networks are frequently used as foundational models. For example, as demonstrated in our experiments, we utilize pretrained ViT for computer vision tasks and BERT for natural language processing tasks to show the flexibility of our method.
> >
> > Thus, our method offers a cost-effective solution while addressing fairness directly in the latent space. Additionally, it has the potential to complement sensitive-invariant representation learning by mitigating residual biases even after sensitive features have been disentangled. We will add a discussion to highlight how our method fits into the broader fairness literature and its potential for future integration with other approaches
> >
> > ### Q2: Connection to Group Utility Difference
> > Our method indirectly addresses group utility differences, such as disparities in accuracy, by aiming to achieve Equalized Odds (EOd). EOd focuses on reducing gaps in True Positive Rate (TPR) and False Positive Rate (FPR) between groups. Since accuracy depends on the combination of TPR and True Negative Rate (TNR), and given the relationship $TNR = 1 - FPR$, reducing TPR and FPR gaps naturally contributes to narrowing accuracy disparities between groups. By ensuring EOd, our method mitigates misclassification disparities, thereby aligning accuracy across groups.
> >
> > Additionally, training on perturbed samples in our proposed method induces Rawlsian fairness. Specifically, the fine-tuning objective on perturbed samples can be expressed as $\max (\text{Acc} - \text{EOd})$. Given:
> > $$Acc = \sum_{y\in\\{0,1\\},a\in\\{0,1\\}}Acc_{y}^a, EOd = \vert TPR_{a=0} - TPR_{a=1} \vert+ \vert TNR_{a=0} - TNR_{a=1}\vert.$$
> > the objective can be rewritten as:
> > $$\small\max \Big( (ACC_{y=1}^{a=1}+ACC_{y=1}^{a=0}+ACC_{y=0}^{a=1}+ACC_{y=0}^{a=0})- (\vert ACC_{y=1}^{a=0} - ACC_{y=1}^{a=1} \vert +\vert ACC_{y=0}^{a=0} - ACC_{y=0}^{a=1} \vert)\Bigr).$$
> > Assuming $a=1$ is the privileged group, where $ACC_{y=1}^{a=0} < ACC_{y=1}^{a=1}$ and $ACC_{y=0}^{a=1} < ACC_{y=0}^{a=0}$, the objective simplifies to:
> > $$\max 2 (ACC_{y=1}^{a=0}+ACC_{y=0}^{a=1})$$
> > which corresponds to maximizing the accuracy of the two worst-case groups.
> >
> > Consequently, while accuracy disparity is not explicitly optimized in our approach, reducing TPR and FPR disparities inherently narrows accuracy differences. This connection demonstrates how achieving EOd fairness also contributes to parity in group utility, aligning with Rawlsian Max-Min fairness principles.

---

> > > ### Comment · Reviewer_aVVS · 2024-11-25
> > >
> > > You response has addressed my concerns. Based on the current version, I would like to raise my rating to 6.

---

### Meta-Review · Area_Chair_tv1N · 2024-12-13

**Metareview:**

I have read all the materials of this paper including the manuscript, appendix, comments, and response. Based on collected information from all reviewers and my personal judgment, I can make the recommendation on this paper, reject. No objection from reviewers who participated in the internal discussion was raised against the reject recommendation. But SAC recommended to accept this paper based on the consensus of the reviewers. Below is my original comments and the discussion between SAC and me.

**Research Question**

This paper considers the supervised fairness machine learning.

**Motivation**

The motivation is unclear and too general. The authors argue that current approaches that emphasize augmenting latent features, rather than input spaces, offer limited insights into their ability to detect and mitigate bias. What does this mean? It seems that the authors would like to sell the concept of “unfair region.”

**Philosophy**

Since the challenges are unclear, it is also unclear what philosophy is applied to tackle the challenges.

**Technique**

1.	The “unfair region” is disconnected from the proposed method. The authors claim that the proposed method can automatically identify the unfair region in Line 216; however, I did not find any clue in Section 3.1-3.3.
2.	Adversarial augmentation is widely used in attack-defense areas. The authors need to demonstrate the specific challenges in fairness learning. I did not see this part in this paper.

**Experiment**

1.	The experiments do not demonstrate the unfair region. In my eyes, it is of less value to identify unfair regions in the latent space, which lacks interpretation. Moreover, according to Eq.(1), it is difficult to calculate. Therefore, the authors do not verify their motivation. This point makes this paper not self-standing.

**Presentation**

The fonts in some figures are a little small.


---------------------------
**Discussion between AC and SAC**

Although the AC has significant concerns about the motivation of this paper, after discussion with the SAC, SAC recommended to accept this paper based on the consensus of the reviewers. However, the AC would like to strongly exhort the authors to improve the motivation in the final version, based on the critiques below. They are actually the extensions of the above points.

**Research question**

This paper considers supervised fairness machine learning, which aims to reduce misclassification rates among different groups. Every paper in this area does the same thing, reducing EOd. This is not motivation.

**Motivation**

Instead, motivation might come from two aspects according to the type of the targeted research question. If the research question is a newly defined one, the motivation should be the difficulty to tackle the research question; if the research question is a well-defined one, the motivation should talk about the drawbacks of existing solutions. For this paper, it belongs to the second category. The authors argue that "current approaches that emphasize augmenting latent features, rather than input spaces, offer limited insights into their ability to detect and mitigate bias" and provide an extended description in Line 48-53. However, the claim is too general, and it fails to demonstrate the value to address fairness issues that arise in the latent space. The latent space is determined by the algorithm and data, which lack interpretability. Thus, I am not convinced by this motivation.

**Unfair region**

3.1 has no relationship with unfair regions, which means this concept is never used in the proposed method and theoretical analysis.

**Experiments**

Only demonstrating superior performance on the research question is not enough. We all know that no algorithm can always win. Beyond this, it is essential to demonstrate whether the proposed method can tackle the targeted challenges. Combining this part makes a paper self-standing. Back to this paper, it is unclear on the motivation or the challenges.

**Suggestion to modify**

Basically, a paper without a clear and solid motivation is not self-standing, where self-standing is the basic criterion of accepting a paper. My personal suggestion is to remove "unfair region" from the paper and figure out another solid motivation.

**Additional Comments On Reviewer Discussion:**

No objection from reviewers who participated in the internal discussion was raised against the reject recommendation.

One reviewer participated into the internal discussion and supported AC initial rejection recommendation.

---

### Decision · Program_Chairs · 2025-01-22

Accept (Poster)